# Data-IQ: Characterizing subgroups with heterogeneous outcomes in tabular data

**Nabeel Seedat**
University of Cambridge
ns741@cam.ac.uk

**Jonathan Crabbé**
University of Cambridge
jc2133@cam.ac.uk

**Ioana Bica**
University of Oxford
The Alan Turing Institute
ioana.bica@eng.ox.ac.uk

**Mihaela van der Schaar**
University of Cambridge
The Alan Turing Institute
UCLA
mv472@cam.ac.uk

## Abstract

High model performance, on average, can hide that models may systematically underperform on subgroups of the data. We consider the tabular setting, which surfaces the unique issue of *outcome heterogeneity* - this is prevalent in areas such as healthcare, where patients with *similar features* can have *different outcomes*, thus making reliable predictions challenging. To tackle this, we propose *Data-IQ*, a framework to systematically stratify examples into subgroups with respect to their outcomes. We do this by analyzing the behavior of individual examples during training, based on their predictive confidence and, importantly, the aleatoric (data) uncertainty. Capturing the aleatoric uncertainty permits a principled characterization and then subsequent stratification of data examples into three distinct subgroups (*Easy*, *Ambiguous*, *Hard*). We experimentally demonstrate the benefits of Data-IQ on four real-world medical datasets. We show that Data-IQ's characterization of examples is most robust to variation across similarly performant (yet different) models, compared to baselines. Since Data-IQ can be used with *any* ML model (including neural networks, gradient boosting etc.), this property ensures consistency of data characterization, while allowing flexible model selection. Taking this a step further, we demonstrate that the subgroups enable us to construct new approaches to both feature acquisition and dataset selection. Furthermore, we highlight how the subgroups can inform reliable model usage, noting the significant impact of the *Ambiguous* subgroup on model generalization.

## 1 Introduction

Most machine learning models are optimized using empirical risk minimization (ERM), to maximize average performance during training [1]. However, in real-world settings, while models may perform well on average, they might underperform on specific subgroups of data [2–4]. Most of the current literature has focused on this problem in computer vision, where the underperforming subgroups are typically associated with data examples that have spurious correlations [1, 5] or mislabelling [6].

In this paper, we focus on tabular data, the most ubiquitous format in medicine and finance, where data is based on relational databases [7, 8]. Specific to the tabular setting, we formalize an understudied source of underperformance, namely *heterogeneity of outcomes*. This phenomenon is vital in healthcare, where patients with *similar features* can have *different outcomes* [9–11]. For example, [12] showed that prognostic models for risk prediction perform well on average, but underperform on

36th Conference on Neural Information Processing Systems (NeurIPS 2022).

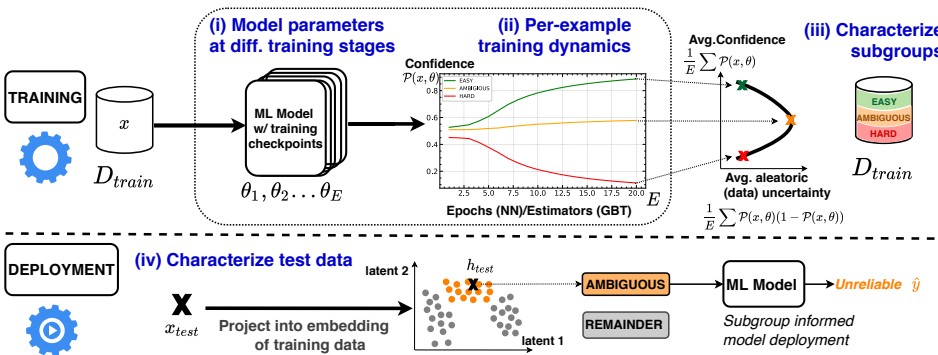

Figure 1: Data-IQ systematically characterizes data into subgroups using (i) any ML model, trained in stages (epochs/iterations) & can be checkpointed. At training time, Data-IQ (ii) leverages the model's checkpoints to analyze the training behavior of individual examples and (iii) characterizes each example based on its aleatoric (data) uncertainty & prediction confidence. (iv) At deployment, the subgroups inform model usage by embedding the training set in a representation space and characterizing the new data points in this representation space.

specific cancer types due to heterogeneity of risk (outcome). Prior works have audited subgroups belonging to sensitive attributes (e.g. demographics, race or gender), as it is well-known that ML models generally underperform on these subgroups [13, 14]. However, this approach is limiting, as it needs the sensitive attributes to be specified, and it also does not capture the case where complex feature interactions may lead to underperformance.

We take a different approach to automatically stratify data into subgroups, usable with *any ML model trained in stages (epochs/iterations)*; e.g., neural networks, gradient boosting etc. Specifically, we study the behavior of individual examples during training, called *training dynamics*. This allows us to formalize that examples can lie on the spectrum from easy to hard to predict. More concretely, let's consider the task of patient mortality prediction. Based on their features, sicker patients more often have a mortality event. Thus, they are easy to learn for any model and will be predicted *correctly* with *high confidence* ($Easy$). However, a subgroup might have a heterogeneous outcome: survival despite their poor prognosis. This heterogeneity could result from randomness, making it practically impossible for a model to learn. These examples will be predicted *incorrectly* yet with *high confidence* (or equivalently have low confidence for the correct class) ($Hard$). In tabular data, there are also examples with inherent ambiguity where the predicted probability for the correct class remains low. They appear where the current features are insufficient to distinguish the example correctly, regardless of the model used [15, 16] ($Ambiguous$). These subgroups naturally arise in real data; see Fig.1 (ii).

Identifying these subgroups is practically valuable, as improving accuracy and robustness often depends on the data's characteristics and quality [17–20]. As mentioned in [20, 21], the "data" work is often undervalued as merely operational, yet failing to account for it can have immense practical harm [12, 20]. Consequently, our goal is to build a systematic framework with the following desired properties (**P1-P4**), motivated by the considerations of practitioners at various stages of the ML pipeline. In satisfying **P1-P4**, we seek to address the "dire need for an ML-aware data quality that is not only principled, but also practical for a larger collection (. . .) of ML models" [19]:

**(P1) Robust data characterization**: the characterization of data examples should be robust, such that it is consistent across similar performing models, that have different architectures/parametrizations.
**(P2) Principled data collection**: the characterization should be informative and actionable, providing practitioners insights that enable both quantitative feature collection and selection between datasets.
**(P3) Reliable model deployment**: the characterization should enable reliable model usage, both by unmasking unreliable subgroups or using the subgroups to tailor the data for better performance.
**(P4) Plug & play**: the characterization should be applicable to a variety of ML models widely used on tabular data, including neural networks, gradient boosting (and variants) etc.

To fulfill *P1-P4*, we propose **Data-IQ**, a systematic framework that characterizes examples based on the inherent qualities (IQ) of the data; at both training and deployment time. As outlined in Fig.1, Data-IQ leverages confidence and in the *"data-centric AI"* spirit focused on the data: aleatoric uncertainty (i.e. uncertainty inherent to the data). This permits Data-IQ to provide ML-aware data quality that is principled and practical for a variety of ML models, making the following contributions:

**Contributions:** ① Data-IQ models the aleatoric (data) uncertainty, which permits subgroup identification that is most robust to variation across different yet similar performing models/parameterizations, compared to other baselines, i.e. *P1*. ② Data-IQ aids with principled data collection *P2* in two ways: Firstly, it permits to quantify the value of an acquired feature by measuring how the feature reduces the aleatoric uncertainty of the example. This information enables a more principled approach to feature acquisition. Secondly, it permits to compare datasets based on the proportion of ambiguous examples. We demonstrate that the proportions link to how well a model trained with the dataset generalizes. ③ Experimentally, the subgroups identified by Data-IQ can inform reliable model deployment, i.e. P3. We highlight cases, where assessment on average might mask unreliable performance, including data sculpting, model robustness, and uncertainty estimation methods. ④ Data-IQ by construction is "plug-and-play" i.e. *P4* with *any* ML model that can be checkpointed, granting practitioners flexibility to apply Data-IQ to their model of choice.

## 2    Related work

This paper primarily engages with the literature on data characterization and contributes to the nascent area of data-centric AI [22, 23]. An extended discussion of related work is found in Appendix A.1.

**Data characterization.** The literature to characterize data samples has used a myriad of different metrics. However, their goals have typically been different, such as spurious correlation or mislabelling, compared to Data-IQ, whose goal is to characterize subgroups with respect to the outcome predictions. Furthermore, none of these methods completely addresses all the desired properties (P1-P4). The closest to our work on data quality is Data Maps [24]. A key contrast to Data-IQ is that Data Maps use confidence and prediction variability to flag instances. In Sec. 3, we show that this prediction variability corresponds to the model uncertainty (i.e. epistemic uncertainty). Alternatively, Data-IQ takes a different and more principled approach, capturing the inherent data uncertainty (known as *aleatoric uncertainty*) [25]. Epistemic uncertainty is reducible by collecting more data. In comparison, aleatoric uncertainty is irreducible even with more samples. This is due to the fact that it captures properties inherent to the data [25–27]; only better features can reduce the aleatoric uncertainty [27]. Later in Fig. 4, we show on real data that capturing the aleatoric uncertainty allows Data-IQ to be more robust to variation across different models, compared to Data Maps (*P1*). This allows practitioners to characterize their data in such a way that the insights are more consistent. We further show theoretically in Sec. 3.3, why the characterization by Data-IQ indeed provides a more principled definition for *Ambiguous* examples, compared to Data Maps.

Besides Data Maps, other related methods address specific computer vision problems: identifying mislabelled images using area under the margin (AUM) [6], gradient norm to identify "important examples" to aid pruning during training [28], or underperformance due to spurious image correlations [1]. The tabular setting considered in this paper requires new methods, due to the specific problem of heterogeneous outcomes for examples with similar features (i.e. "feature collision"). The ambiguity in the tabular, "feature collision" sense, is different or non-existent in modalities such as images.

**Data-Centric AI.** The assessment of data quality is a critical but often overlooked problem in ML [20]. While the focus in ML is typically on optimizing models, the task of ensuring high quality data (or even improving one's data) can be equally valuable to improving performance [17, 20]. Even when it is considered, the process of assessing datasets is adhoc or artisinal [20, 22, 29, 30]. The recent growth of the data-centric AI space aims to build systematic tools for "data collection, labeling, and quality monitoring processes for datasets to be used in machine learning" [23, 30]. Data-IQ contributes to this nascent body of work, specifically around ML-aware data quality monitoring [19].

## 3    Formulation

This section gives a detailed formulation of Data-IQ and motivates our proposed example stratification that uses aleatoric uncertainty and confidence. We then describe how Data-IQ stratifies examples into subgroups at both training and testing time. Finally, we show Data-IQ's formulation permits usage with *any* ML model trained in stages, e.g. neural networks, GBDTs etc, unlike other approaches.

## 3.1 Preliminaries

We consider the typical supervised learning setting, where the aim is to assign an input $x \in \mathcal{X} \subseteq \mathbb{R}^{d_X}$ to a class $y \in \mathcal{Y} \subset \mathbb{N}$. We have a dataset $\mathcal{D}$ with $N \in \mathbb{N}^*$ examples, i.e. $\mathcal{D} = \{(x^n, y^n) \mid n \in [N]\}$ drawn IID from an unknown distribution. Our goal is then to learn a model $f_\theta : \mathcal{X} \to \mathcal{Y}$, parameterized by $\theta \in \Theta$. Typically, the parameters $\theta$ are learned to minimize empirical risk, by minimizing the average training loss , i.e. $\text{ERM}(\theta) = \frac{1}{n} \sum_{i=1}^{n} \ell(x_i, y_i; \theta)$, with a loss function $\ell : \mathcal{X} \times \mathcal{Y} \times \Theta \to \mathbb{R}^+$.

This brings us to the essence of the problem: "not all examples are created equally". e.g. patients with similar features might have heterogeneous outcomes, reflected in their labels $y$ being different. These correspond to subgroups within $\mathcal{D}_{\text{train}}$ on which a predictive model might systematically underperform. We formalize this concept of hidden heterogeneous subgroups by assigning to each example $x^n$ a hidden *subgroup* label $g^n \in \mathcal{G}$, where $\mathcal{G} = \{Easy, Ambiguous, Hard\}$. Before giving a precise description of how those group labels are assigned, it is useful to detail the context. Several works have established that the training dynamics of a model, contains signal about the quality of the data itself [31–33]. For instance, it takes more epochs/iterations for a model to assign the correct label to noisier/more difficult training examples. With Data-IQ, we build on those observations and assign a label $g^n$ to each example $x^n$ by studying its training dynamic, which is then used to estimate the aleatoric uncertainty and predictive confidence of each example. The following sections detail how this is done and how this contrasts with existing approaches.

## 3.2 Uncertainty decomposition during training

Recall that practitioners desire flexibility in the choice of the model. Hence, we focus on *any* ML model that is trained in stages and can be checkpointed during training, $f_\theta : \mathcal{X} \to \mathcal{Y}$ parameterized by $\theta$ and on a given example *from the training set* $(x, y) \in \mathcal{D}_{\text{train}}$. Assume that the model $f_\theta$ corresponds to a conditional categorical distribution, assigning a probability to each class given the input $x$: $f_\theta(x) = P(Y \mid X = x, \vartheta = \theta)$. During iterative training, the model parameters $\theta$ vary, where over $E \in \mathbb{N}^*$ epochs/iterations, these parameters take $E$ different values at each checkpoint, i.e. $\theta_1, \theta_2, \ldots, \theta_E$. Since our analysis relies on the model's training dynamics, we want to take those different parameters into account. For the sake of notation, we introduce a random variable $\vartheta$ that has an empirical distribution over this set of parameters captured through the training process $\vartheta \sim P_{\text{emp}}(\{\theta_e \mid e \in [E]\})$. The variability of the model's parameters at training time is then reflected by the variance $\mathbb{V}_\vartheta [\cdot]$.

The uncertainty we model is based on the random variable $Y \mid X = x$ that represents the possible labels given the input $x$. Since the ground-truth label $y$ is available for training examples, we would like to distinguish between 2 cases: ① the predicted label corresponds to the ground-truth label $Y = y$ and ② the predicted label is different from the ground-truth label $Y \neq y$. To this end, we introduce a binary random variable $\tilde{Y}$ that is set to one when the predicted label equals the ground-truth label ($\tilde{Y} = 1$ if $Y = y$) and that is zero otherwise ($\tilde{Y} = 0$ if $Y \neq y$). As discussed earlier, we are interested in the uncertainty on the predictive random variable $\tilde{Y} \mid X = x$. This uncertainty is modeled by the variance $v(x) = \mathbb{V}_{\tilde{Y}|X} \left[ \tilde{Y} \mid X = x \right]$. We will now show that this quantity can be evaluated with the model predictions.

We start by noting that the definition of $\tilde{Y}$ implies that $\tilde{Y} \mid X = x, \vartheta = \theta$ is a Bernoulli random variable with parameter[1] $\mathcal{P}(x, \theta) = P(Y = y \mid X = x, \vartheta = \theta) = [f_\theta(x)]_y$. From this observation, we can decompose $v(x)$ with the law of total variance and make each term explicit:

$$
v(x) = \underbrace{\mathbb{V}_\vartheta \left[ \mathbb{E}_{\tilde{Y}|X,\vartheta} \left[ \tilde{Y} | X = x, \vartheta \right] \right]}_{\text{Bernoulli Mean: } \mathcal{P}(x, \vartheta)} + \underbrace{\mathbb{E}_\vartheta \left[ \mathbb{V}_{\tilde{Y}|X,\vartheta} \left[ \tilde{Y} | X = x, \vartheta \right] \right]}_{\text{Bernoulli Var: } \mathcal{P}(x, \vartheta)(1 - \mathcal{P}(x, \vartheta))}
$$
$$
= \underbrace{\mathbb{V}_\vartheta \left[ \mathcal{P}(x, \vartheta) \right]}_{\text{Epistemic uncertainty: } v_{\text{ep}}(x)} + \underbrace{\mathbb{E}_\vartheta \left[ \mathcal{P}(x, \vartheta)(1 - \mathcal{P}(x, \vartheta)) \right]}_{\text{Aleatoric uncertainty: } v_{\text{al}}(x)}.
\tag{1}
$$

In Eq. (1), we have split the overall uncertainty into two components: *epistemic* and *aleatoric uncertainty*. This type of decomposition is similar to those in the context of Bayesian neural

---

[1] In this case, $[f_\theta(x)]_y$ denotes the component $y$ of the probability vector $f_\theta(x)$.

networks [34, 35]. To understand the distinction between uncertainties, it is useful to closely examine the variances in the *first* equality of Eq. (1). For *epistemic uncertainty* $v_{\mathrm{ep}}$, variance is evaluated on the *model parameters* $\vartheta$. Hence, epistemic uncertainty originates from the fact that a model's predictions oscillate when we change its parameters. For the *aleatoric uncertainty* $v_{\mathrm{al}}$, the variance is evaluated on the *predicted label* $\tilde{Y} \mid X, \vartheta$. Hence, the variability originates from the inability to predict the correct label with high confidence. While existing works use epistemic uncertainty to stratify examples, we argue that aleatoric uncertainty is a better principled choice to capture the inherent data uncertainty.

### 3.3 Stratification based on data uncertainty

We now explain how the above notion of uncertainty permits to assign a group label $g \in \mathcal{G}$ to each training example $x$. First, we use the empirical distribution $\vartheta \sim P_{\mathrm{emp}}(\{\theta_e \mid e \in [E]\})$ to explicitly write the two types of uncertainties in Eq. (1), where $\bar{\mathcal{P}}(x) = {}^{1}\!/\!_{E} \sum_{e=1}^{E} \mathcal{P}(x, \theta_e)$:

$$v_{\mathrm{ep}}(x) = \frac{1}{E} \sum_{e=1}^{E} \left[ \mathcal{P}(x, \theta_e) - \bar{\mathcal{P}}(x) \right]^2 \qquad v_{\mathrm{al}}(x) = \frac{1}{E} \sum_{e=1}^{E} \mathcal{P}(x, \theta_e)(1 - \mathcal{P}(x, \theta_e)), \qquad (2)$$

**Stratification at training time.** Before giving a precise definition of the group labels, let us give an intuitive definition for each group. ① *Easy*: examples that have low data uncertainty that the model can correctly predict with high confidence, ② *Ambiguous*: examples that have high data uncertainty, hence the model is unable to predict with confidence and ③ *Hard*: examples that have low data uncertainty that the model is unable to predict (i.e. predicted incorrectly yet with high confidence or equivalently have low confidence for the correct class). We note that we need the model's prediction for the ground-truth class to delineate *Easy* and *Hard* examples. In practice, we use the model's average confidence for the ground-truth class $\bar{\mathcal{P}}(x)$ defined previously for this purpose. We make use of this concept to detail how labels are assigned to training examples $(x, y) \in \mathcal{D}_{\mathrm{train}}$:

$$g(x, \mathcal{D}_{\mathrm{train}}) = \begin{cases} \text{Easy} & \text{if } \bar{\mathcal{P}}(x) \geq C_{\mathrm{up}} \;\wedge\; v_{\mathrm{al}}(x) < P_{50}\left[v_{\mathrm{al}}(\mathcal{D}_{\mathrm{train}})\right] \\ \text{Hard} & \text{if } \bar{\mathcal{P}}(x) \leq C_{\mathrm{low}} \;\wedge\; v_{\mathrm{al}}(x) < P_{50}\left[v_{\mathrm{al}}(\mathcal{D}_{\mathrm{train}})\right] \\ \text{Ambiguous} & \text{otherwise} \end{cases} \qquad (3)$$

where $C_{\mathrm{up}}$ and $C_{\mathrm{low}}$ are upper and lower confidence threshold resp. and $P_n$ the n-th percentile. We provide a practical method to set $C_{\mathrm{up}}$ and $C_{\mathrm{low}}$, applicable to any dataset in Appendix A.

In contrast to Data-IQ which uses Aleatoric uncertainty $v_{\mathrm{al}}(x)$; Data Maps [24] identifies ambiguous training examples $(x, y) \in \mathcal{D}_{\mathrm{train}}$ as those with high epistemic uncertainty $v_{\mathrm{ep}}(x)$. We consider a typical scenario to see how this characterization might cause problems – illustrated in Fig. 2. Consider an example $x$ in which the model cannot classify confidently during the entire training $\mathcal{P}(x, \theta_e) = 0.5 \; \forall e \in [E]$. In this case, the epistemic uncertainty $v_{\mathrm{ep}}(x)$ vanishes, as the prediction is consistently unconfident (i.e. low variability of the model predictions). This implies that Data Maps would consider this example as non-ambiguous, despite the ambiguous model prediction for this example. This problem can be traced back to the definition of epistemic uncertainty, which measures the sensitivity of a model prediction with respect to the model's parameters.

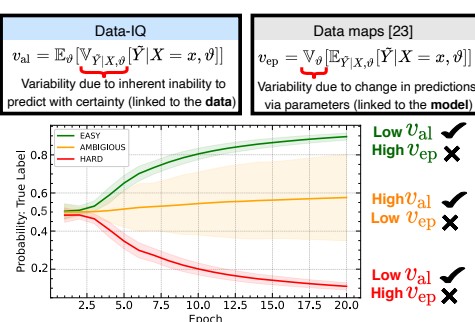

Figure 2: Example outlining the differences between $v_{\mathrm{al}}$ (Data-IQ) and $v_{\mathrm{ep}}$ (Data Maps) & showing the type of uncertainty matters.

A more principled definition for ambiguous examples should capture examples for which the model cannot predict the appropriate label with high confidence (i.e. data uncertainty). This is precisely what the aleatoric uncertainty $v_{\mathrm{al}}(x)$ captures (Data-IQ). Furthermore, it is easy to verify that the previous example $\mathcal{P}(x, \theta_e) = 0.5 \; \forall e \in [E]$ maximizes the aleatoric uncertainty (see Fig. 2). Since high aleatoric uncertainty captures ambiguous examples for various values of the model's parameters, we believe that it better reflects the inherent quality of the data. In that sense, we expect this quantity to be more stable and robust to variation for different ML model parameters/architecture changes (*P1*). We experimentally validate the consistency in Sec. 4.

**Stratification at inference time.** Most previous methods are only applicable at training time. To address this limitation and improve the practical utility of our method, we also stratify examples into subgroups at deployment time. However, if we try to apply the above stratification for incoming data at deployment time, we face a problem: $\bar{\mathcal{P}}(x)$ requires the ground-truth class $y$.

For this reason, we follow an alternative approach based on representation learning that does not require access to ground-truth labels. The idea is the following: we construct a low-dimensional UMAP embedding [36] $h : \mathcal{X} \to \mathcal{H}$ of the training set's examples $x_{\text{train}} \in \mathcal{D}_{\text{train}}$. In doing this, we note two things (see Appendix C): ① *Ambiguous* examples have distinctive features and are clustered in embedding space. Thus, it is possible to distinguish the *Ambiguous* examples using the embedding. ② It is not possible to reliably distinguish *Easy* examples from *Hard* examples based on the embedding, because *Hard* examples are a minority with outcome randomness that have similar features, as the *Easy* examples. Combining these observations, we note it is possible to identify *Ambiguous* test examples. This label is assigned by computing the related embedding $h(x_{\text{test}})$ and comparing this embedding to the nearest neighbor embedding from the training set, i.e. $d[h(x_{\text{test}}), h(x_{\text{train}})] \forall x_{\text{train}} \in \mathcal{D}_{\text{train}}$. For models like neural networks with an implicit representation space, the same analysis can be done using the model's representation space.

### 3.4 Using Data-IQ with a variety of models, beyond Neural Networks (P4)

The baseline methods discussed are primarily applicable only to neural networks. However, practically in tabular settings (e.g. healthcare/finance etc), practitioners often use other highly performant iterative learning algorithms such as Gradient Boost Decision Trees (GBDTs) or variants [7, 37]. Data-IQ's formulation by construction is naturally adaptable to *any* ML model trained in stages, that can be checkpointed. This satisfies *P4*, which allows practitioners the flexibility to use Data-IQ with their application-specific model of choice. Appendix A provides guidelines, space and time considerations, as well as, discussing the specifics of how Data-IQ is easily adapted, for example to GBDTs.

## 4 Experiments

This section presents a detailed empirical evaluation demonstrating that Data-IQ [2][3] satisfies (**P1**) Robust data characterization, (**P2**) Principled data collection and (**P3**) Reliable Model Deployment, introduced in Sec.1. Recall that (**P4**) Plug and play is satisfied by construction of Data-IQ.

**Datasets.** We conduct experiments on four real-world medical datasets, with diverse characteristics (different sizes, binary/multiclass, varying degrees of task difficulty etc) and highlight real-world applicability with heterogeneous patient outcomes: (1) Covid-19 dataset of Brazilian patients [38], (2) Prostate cancer datasets from both the US [39] and UK [40], (3) Support dataset of seriously ill hospitalized adults [41], (4) Fetal state dataset of cardiotocography [42]. We describe the datasets in greater detail in Appendix B, along with further experimental details. *We observe similar performance across different datasets, but given the space limitations, we typically show pertinent results for a single dataset, and include results for the other datasets in Appendix C.*

### 4.1 (P1) Robust data characterization

**Robustness to variation.** As per *P1*, we desire that Data-IQ identifies subgroups in a manner robust to variation across different models. This would allow a practitioner to obtain consistent insights about their data even when using different model architectures/parameterizations. When comparing the different methods from Sec. 2,

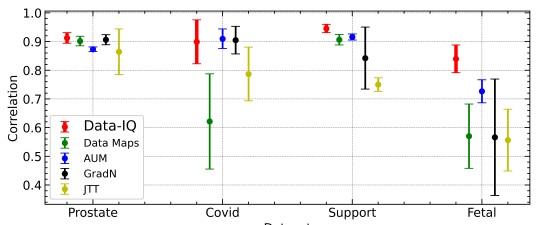

Figure 3: Robustness to variation across models based on Spearman correlation, where Data-IQ has the highest correlation (i.e. consistency) across all datasets.

we note that each method has its own specific metric used to characterize examples (see Appendix B). To assess robustness to variation, we compare the consistency of the different characterization metrics, evaluated on models with different architectures/parameterizations. All models are trained to convergence, with early stopping on a validation set.

---

[2] https://github.com/seedatnabeel/Data-IQ [3] https://github.com/vanderschaarlab/Data-IQ

Quantitatively, we compute the Spearman rank correlation between all model combinations, see Fig. 3. We observe that Data-IQ is the most consistent and robust to variation across different models, having the highest score on all datasets, satisfying *P1*. Further, the baseline methods themselves are also not consistent in performance ordering across datasets, which is undesirable. Ultimately, the robustness means practitioners can feel confident in the consistency of data insights, derived using Data-IQ.

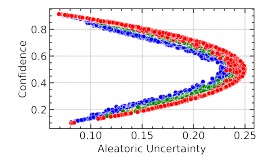

(a) Data-IQ

To further compare Data-IQ and Data Maps[24], we examine 3 distinct models that achieve similar performance on the Covid-19 [38] tabular dataset, and we produce a characterization of the training set using each model in Fig. 4. We note that Data Maps groups can be recovered from (3) by replacing the aleatoric uncertainty $v_{al}$ from Data-IQ with its epistemic counterpart $v_{ep}$. The y-axis is the same for both methods and corresponds to $\bar{\mathcal{P}}(x)$. The x-axis corresponds to $v_{al}(x)$ for Data-IQ and to $v_{ep}(x)$ for Data Maps. Each model is assigned a color in Fig. 4. We note three things ① Data-IQ's characterization of the data is significantly more stable across models. ② Linked to the points in Sec.3, Data Map's high

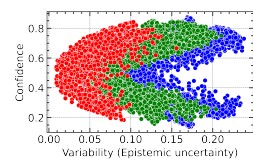

(b) Data Maps [24]

Figure 4: Data-IQ's robustness to variation across diff. models (i.e.colors)

and low confidence examples in fact have high epistemic uncertainty $v_{ep}$, which can lead to incorrect conclusions when attempting to use Data Maps to characterize data. ③ Data-IQ always distributes the data around a bell shape, which standardizes its interpretation. We provide a theoretical analysis to explain this bell shape observation in Appendix A.

**Data-IQ: Neural Networks vs Other model classes.** Data-IQ can be used with *any* ML model trained in stages linked to *P4*: Plug and Play. Methods such XGBoost, LightGBM and CatBoost methods are widely used by practitioners on tabular data, often more so than neural networks [7]. Ideally, based on *P1*, we desire that the characterization of examples be consistent for similar performing models, irrespective of whether the model is a neural network or an XGBoost model.

To assess the robustness of both Data-IQ and Data Maps, we train a neural network, XG-Boost, LightGBM and Cat-Boost models to achieve the same performance and then perform the characterization for all models. We can clearly see in Fig. 5 (Support) that Data-IQ has a similar characterization across all four models. Contrastingly, for Data Maps, the char-

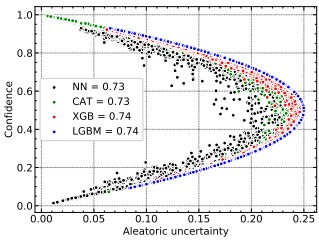

(a) Data-IQ

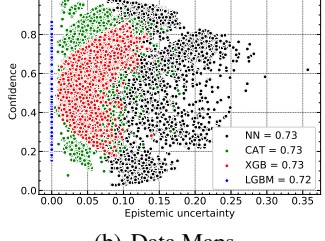

(b) Data Maps

Figure 5: NN vs XGBoost: Data-IQ is more consistent

acterizations are significantly different for the different model classes. The implication of this result is that by Data-IQ capturing the uncertainty inherent to the data (aleatoric uncertainty), it leads to a more consistent and stable characterizations of the data itself. Especially, this highlights that Data-IQ characterizes the data in a manner that is not as sensitive to the choice of model when compared to Data Maps. For more, see Appendix C.

**Data insights from subgroups.** Given the distinct differences between the subgroups, we seek to understand what factors make these subgroups different and how they can provide insight into the dataset. Such insights are especially useful in clinical settings. Results for the prostate cancer dataset are illustrated in Fig. 6 (with other datasets in Appendix C). To visualize the different groups of patients within each subgroup, we cluster each subgroup (*Easy*, *Ambiguous* and *Hard*) using a Gaussian Mixture Model (GMM) similar to [5], selecting the optimal clusters based on the Silhouette score. We assess cluster quality vs alternatives in Appendix C.

In general, across datasets, the subgroups are: (1) *Easy*: Severe patients with a death outcome, and less severe patients with a survival outcome. (2) *Ambiguous*: Patients with similar features, but different outcomes. This could suggest that the features, we have at hand are insufficient to separate the differences in outcomes. (3) *Hard*: Severe patients with a survival outcome, and less severe patients with a death outcome. i.e. opposite outcomes as expected due to randomness in the outcomes.

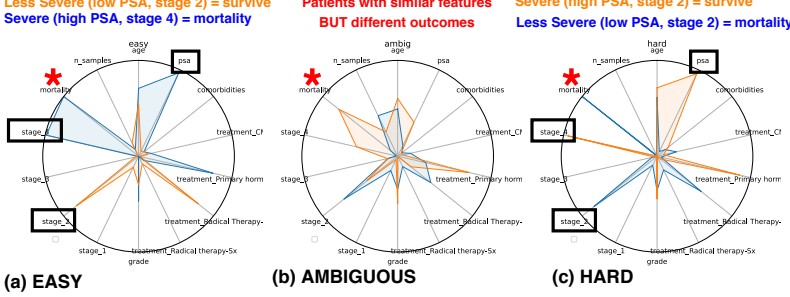

Figure 6: Comparing subgroups identified by Data-IQ (descriptions above). Colors represent the GMM clusters.

## 4.2 (P2) Principled data collection

**Principled feature acquisition.** As per Sec.4.1, the *Ambiguous* subgroup has examples with similar features, yet different outcomes. Recall that this case of ambiguity in the tabular setting is very different from ambiguity in other modalities, such as images. The ambiguity is due to insufficient features to adequately separate the examples. We link this to the concept that the *Ambiguous* subgroup has a high aleatoric uncertainty that is irreducible, even if we collect more data examples. Rather, aleatoric uncertainty can only be reduced by acquiring better features [27]. We leverage this idea and show that Data-IQ's example characterization provides a principled approach to assessing the benefit of acquiring a specific feature.

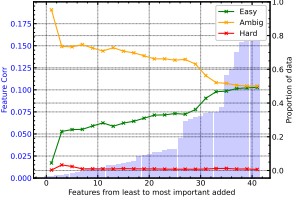

(a) Data-IQ subgroup ambiguity proportion is reduced as more informative features are acquired.

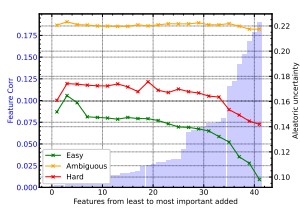

(b) Data-IQ aleatoric uncertainty remains stable for Ambiguous, reduces for others as features are acquired.

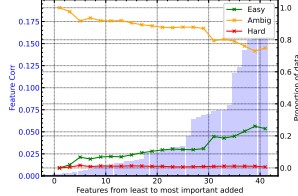

(c) Data Maps subgroup proportions largely unaffected as features acquired.

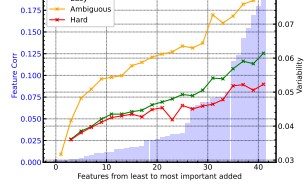

(d) Data Maps variability increases across subgroups as features acquired.

Figure 7: Quantifying the value of feature acquisition based on change in ambiguity. Only Data-IQ captures this.

This is different from feature selection, where all features are present and we select the most "important feature". Additionally, this is different from active learning which quantifies the value of acquiring examples, not features.

With the above in mind, a valuable feature should decrease the example ambiguity (i.e. aleatoric uncertainty). Hence, a decrease in the proportion of *Ambiguous* examples can serve as a proxy for the feature's potential value to the dataset. Understanding the value of features is useful in settings such as healthcare, where feature acquisition comes at a cost. To showcase the potential, we construct a semi-synthetic experiment, where we rank sort the features based on correlation with the target.

We then train different models, where we sequentially "acquire" features of increasing value (based on correlation). Fig. 7. shows results for the Support dataset. For Data-IQ, Fig 7 (a) shows that as we acquire "valuable" features, the proportion of the *Ambiguous* subgroup drops, whilst the *Easy* subgroup increases, with significant changes for the important features. This shows that Data-IQ's subgroup characterization can be used to quantify a feature's value, by its ability to decrease ambiguity. In contrast, Data Maps, Fig 7 (c), shows minimal response to feature acquisition, suggesting it may not be sensitive enough to capture the feature's value.

Further, for Data-IQ we see that the examples that remain as *Ambiguous*, after features are collected, maintain a consistent aleatoric uncertainty. This is desired as it demonstrates for those examples which remain *Ambiguous*, that indeed the features collected are not informative enough to reduce their inherent (aleatoric) uncertainty, i.e. those remaining still need better features (see Fig 7 (b)). While for Data Maps the added features, in fact increase the variability for *all* subgroups making it harder to stratify (see Fig 7 (d)). This links to the fact that Data Maps subgroups can't capture the value of the acquired features. We further show experimentally in Appendix C that for *Ambiguous* examples, it is not simply a case of increasing the size of the dataset (i.e. more examples). In fact, this can increase the proportion of *Ambiguous* examples due to the increased probability of feature collisions as the dataset size increases. Ultimately, this motivates the usefulness of principled feature acquisition (which Data-IQ can guide), as a way to decrease dataset ambiguity.

**Principled dataset comparison.** Extending beyond feature-level, an understudied scenario involves systematically selecting between entire datasets in two cases: (1) purchasing data from data markets [43–45] and (2) organizations where the data is siloed, with lengthy access processes [46, 47]. In both cases, synthetic versions of the real dataset has begun to be used [46]. For now, we ignore privacy concerns, and focus on data fidelity and quality, which compares the real and synthetic datasets using statistical measures. However, as per [48], the conclusions can vary across different metrics. In practice, competing "synthetic" datasets can be generated by different ML models or vendors. Thus, while they model the same underlying distribution, depending on the process used, one version might be superior. We now ask whether Data-IQ could permit us to systematically select between synthetic datasets? We consider the setting where the real data is *not accessible*. Hence, we can't use existing evaluation metrics, yet still wish to compare the synthetic datasets (e.g. comparing vendors).

We simulate this scenario by generating synthetic data using 2 different models, representing 2 synthetic data vendors: (V1) CTGAN [49] and (V2) Gaussian Copula [50]. We then characterize the dataset subgroup proportions using Data-IQ. We hypothesize that datasets with greater proportions of *Easy* examples generalize better. As is common, we validate fidelity by training with synthetic data and testing with real data [51, 52], where the best fidelity data produces the best model performance

Table 1: Comparison of accuracy performance rank and (dataset quality). Synthetic dataset w/ better quality (↑ easy) produces the best real data test performance.

| Dataset | (V1) CTGAN | (V2) Gaussian Copula |
|---------|------------|----------------------|
| Prostate | **Rank 1 (63% Easy)** | Rank 2 (30% Easy) |
| Covid | **Rank 1 (70% Easy)** | Rank 2 (63% Easy) |
| Support | **Rank 1 (59% Easy)** | Rank 2 (38% Easy) |
| Fetal | Rank 2 (40% Easy) | **Rank 1 (51% Easy)** |

on real data (test set). Table 1 shows that datasets with the highest quality as measured by Data-IQ indeed produces the best performance on real test data (i.e. Rank 1). Further, it shows that the same "vendor" does not always produce the best dataset, highlighting the value of comparative assessment. Ultimately, these two aspects demonstrate that when the real data is unavailable, Data-IQ is a useful tool in the hands of practitioners wishing to assess data quality, especially when selecting between different datasets.

### 4.3 (P3) Reliable Model Deployment

**Less is more: data sculpting based on subgroups.** What role do the data subgroups, specifically *Ambiguous* examples, play in ensuring model generalization? Using multi-country prostate cancer data, we train a baseline model on US data (SEER) and assess generalization when deployed on patients in the UK (CUTRACT) and vice versa. In Fig. 8, we see that test time generalization performance, monotonically increases as we decrease the proportion of *Ambiguous* training data (see Appendix C.13 for absolute num-

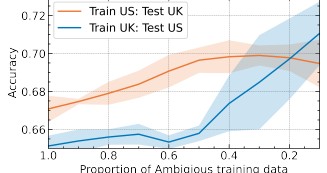
Figure 8: Performance improvement via data sculpting based on subgroups

bers). Ultimately, this illustrates the value of sculpting the training dataset, by removing ambiguous examples, as a way to improve the reliability of a deployed model.

**Group-DRO: Not a silver bullet when used in tabular settings.** Once subgroups of underperformance are identified, it is generally assumed that methods such as Group Distributionally Robust Optimization (DRO) [53] can be applied to improve model performance and robustness. We compare Group-DRO with groups identified by Data-IQ, and as baselines: George [5] and Just-Train-Twice (JTT) [1]. As per the previous experiments, the largest underperforming group (in proportion) is the *Ambiguous* examples. Similar to the literature, we evaluate the performance change from a baseline model, after Group-DRO is applied.

Table 2: Comparison of different model improvement/robustness techniques

| Dataset | Group | Baseline | Group-DRO (**Data-IQ**) | Group-DRO (George) | JTT |
|---------|-------|----------|-------------------------|--------------------|-----|
| Prostate | Overall | 0.837 | **0.840** ↑ | 0.565 ↓ | 0.723 ↓ |
| | Ambiguous | 0.740 | **0.741** ↑ | 0.598 ↓ | 0.538 ↓ |
| | The Rest | **0.935** | **0.935** | 0.535 ↓ | 0.896 ↓ |
| Covid | Overall | 0.729 | **0.732** ↑ | 0.687 ↓ | 0.455 ↓ |
| | Ambiguous | 0.629 | **0.633** ↑ | 0.609 ↓ | 0.477 ↓ |
| | The Rest | **0.832** | **0.832** | 0.766 ↓ | 0.438 ↓ |
| Support | Overall | **0.734** | **0.734** | 0.660 ↓ | 0.611 ↓ |
| | Ambiguous | 0.621 | **0.622** ↑ | 0.576 ↓ | 0.521 ↓ |
| | The Rest | **0.858** | **0.858** | 0.754 ↓ | 0.711 ↓ |
| Fetal | Overall | 0.768 | **0.833** ↑ | 0.829 ↑ | 0.829 ↑ |
| | Ambiguous | 0.575 | **0.701** ↑ | 0.695 ↑ | 0.698 ↑ |
| | The Rest | 0.970 | **0.974** ↑ | 0.970 | 0.970 |

The results in Table 2 show that Group-DRO using Data-IQ's groups both improves overall performance and improves performance on the *Ambiguous* group. Whilst, for the other baselines, the performance actually degrades.

Nevertheless, while using Data-IQ can boost performance, it is evident that simply applying Group-DRO is not a silver bullet to equalize subgroup performance, given the sometimes small improvement. The rationale is our tabular setting is different, from the spurious correlation setting in computer vision where Group-DRO typically shines. Ultimately, we believe, based on the feature acquisition results, that in tabular settings, practitioners would be well served acquiring better features to improve performance and reduce ambiguity.

**Subgroup-informed usage of uncertainty estimation.** Uncertainty estimation methods are essential in safety-critical areas such as healthcare [54], yet are typically assessed on average. We ask the question: since subgroups have different performance properties, are uncertainty estimates equally reliable for each subgroup? As done in the literature [55, 56], if an uncertainty estimate is reliable and informative of predictive performance, it can be used to defer "uncertain examples". This is done by rank sorting examples based on uncertainty and thresholding proportions of tolerated uncertainty [55–58]. Ideally, as the threshold proportion of examples increases (i.e. inclusion of more uncertain examples), we should see a monotonic decrease of accuracy. We assess this by training a Bayesian Neural Network (BNN)[59] to obtain uncertainty estimates. We then compute the performance across different threshold proportions $\tau \in \{0.1, 0.2, \ldots, 1\}$ for the *Ambiguous* subgroup specifically, and as is commonly done, across the entire dataset (i.e. *average*).

Fig. 9 shows a *specific* example, wherein *average* examples exhibit the monotonic decreasing relationship, as expected. However, the *Ambiguous* examples categorized by Data-IQ , contrary to expectations, show an increase in accuracy as more uncertain examples are included. This suggests that uncertainty estimates in this case are not as informative of predictive performance for the *Ambiguous* examples. This shows the potential for practitioners to use Data-IQ at deployment time to understand which examples require auditing before deferring, as the "average" monotonic decreasing behavior, based on the uncertainty estimates,

Figure 9: Data-IQ subgroups can unmask unreliable prediction deferral by uncertainty methods

may not always hold. Ultimately, the result further highlights how subgroup characterization via Data-IQ could assist practitioners in unmasking unreliable performance, not evident on average.

## 5 Discussion

In this paper, we introduce Data-IQ, a systematic framework that can be used with *any* ML model with checkpoints, to characterize examples into subgroups with respect to the outcome. Through several experiments, we demonstrate that the usage of aleatoric uncertainty, which captures properties more inherent to the data, is indeed more principled, being more robust to variation across models and/or parameterizations. Data-IQ's consistency is unmatched by any compared baseline. Data-IQ should not automate and replace the intuition of a data scientist. Rather, as we have demonstrated, Data-IQ should serve as a systematic "data-centric ML" tool that assists and empowers data scientists with the "data" work at training time, whilst also guiding reliable model usage at deployment time.

**Data-IQ beyond tabular settings.** The main paper has primarily assessed the utility of Data-IQ in the tabular setting. That said, in Appendices C.7 and C.8, we evaluate the utility of Data-IQ on text data (*NLP*) and images (*computer vision*) respectively.

**Limitations and future opportunities.** ① While Data-IQ characterizes examples; the current formulation does not allow us to understand which attributes are responsible for the characterization per example. This would be an interesting extension around dataset explainability, allowing practitioners to better probe their data. ② In high-stakes settings such as healthcare, to mitigate possible adverse effects (e.g. difficulty of *Easy* vs *Hard*), Data-IQ should be used with a "human-in-the-loop", allowing experts to complement and validate findings with domain knowledge.

## Acknowledgments

The authors are grateful to Zhaozhi Qian, Yuchao Qin, Evgeny Saveliev and the anonymous NeurIPS reviewers for their useful comments & feedback. Nabeel Seedat is supported by the Cystic Fibrosis Trust, Jonathan Crabbe by Aviva, Ioana Bica by the Alan Turing Institute, EPSRC grant EP/N510129/1 and Mihaela van der Schaar by the Office of Naval Research (ONR), NSF 1722516.

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
