# Appendix: Data-IQ: Characterizing subgroups with heterogeneous outcomes in tabular data

## Table of Contents

# A  Data-IQ details & related work

## A.1  Extended Related Work

We present a comparison of our framework Data-IQ, and provide further contrast to related work. Table 3, highlights that the related methods do not satisfy all the properties (P1-P4). Furthermore even for cases where it is satisfied, the properties are not naturally satisfied; rather they require our framework to enable the use-case. *P2* around data collection and selection, has been largely unaddressed by prior methods. In the case of *P3*, the related methods focus on training time, without taking into account end-to-end deployment scenarios. Finally, these prior methods are typically applicable to only neural networks, which limits the broad applicability related to *P4*, where in high-stakes settings such as healthcare and finance, which often have tabular data, practitioners typically use non-neural methods, such as Gradient Boosting or XGBoost.

Table 3: Comparison of related work, highlights that *ONLY* Data-IQ naturally address all desiderata. ($\heartsuit$): satisfied naturally by construction, ($\diamondsuit$): needs our framework

| | Assessment metric | (P1) Data characterization consistent across models | (P2) Principled data collection & selection | (P3) Applicable at deployment time | (P4) Applicable to *any* ML model |
|---|---|---|---|---|---|
| Data-IQ (Ours) | Aleatoric uncertainty | V. High | ✓ | ✓($\heartsuit$) | ✓($\heartsuit$) |
| Data Maps [24] | Training variability | Medium | ✗ | ✓($\diamondsuit$) | ✓($\diamondsuit$) |
| AUM [6] | Logit difference | High | ✗ | ✗ | ✓ |
| GraNd [28] | Gradient Norm | Medium | ✗ | ✗ | ✗ |
| JTT [1] | Training errors | Low | ✗ | ✓($\diamondsuit$) | ✓ |

The properties of *P1* were evaluated in Section 2. We delve into each subsequent property below and show that the related works have different goals and hence do not satisfy the properties:

- *P2*: Data Maps [24] does not fulfill this property as was shown experimentally in Section 4.2. The method is shown to be insufficiently sensitive for the task. AUM [6] uses the differences between logits to flag mislabelled instances. By virtue, AUM solves a different task of label errors, whereas the task of improving the data by collecting better features, etc., is focussed on the inputs rather than labels. GraNd [28] looks at the gradient norm to identify important examples. Their task, rather than feature acquisition, is to assist in the pruning of a dataset. JTT [1] looks at training errors to detect samples with spurious correlations. Not only is the task different, but flagged errors could also link to both ambiguity (needing more features) or hardness (example being difficult). Therefore, it is not sufficiently fine-grained to be leveraged for feature acquisition. In the best case, we might want to augment the dataset with such flagged examples.

- *P3*: Data Maps naturally operates only at training time (as per the original paper). However, if it leverages Data-IQ's representation space, then it could be extended to be used at test time. AUM operates to flag mislabelled data (i.e. outcome labels). These do not exist at test time, hence AUM is a training-time ONLY method. GraNd uses gradients to flag examples. Hence, it is only applicable at training time and not at test time, since we do not compute gradients at test time as we do not have access to labels. JTT surfaces training errors and hence we would not know test time errors as we do not have labels. However, if JTT leverages Data-IQ's representation space idea, then it could also be extended to be used at test time by flagging examples relative to a training time embedding.

- *P4*: Data Maps was only studied for usage with neural networks. For use with *any* ML model, it can be extended using our framework (see Section 3.4 and Appendix A.2). AUM could of course be used with any ML model with output logits. GraNd does not facilitate this, as only ML models where gradients can be computed are usable. Finally, since JTT, simply captures model errors it is easily usable with any ML model.

**Tangential related work.** We wish to highlight that while methods such as Influence functions [60] and Data Shapley [61] may seem related, firstly they have fundamentally different goals around capturing the contributing role of examples to the models decision boundary and predictions. Hence, they are considered as tangential to our work of characterizing data into subgroups. Additionally, these methods assign an instance importance, related to its contribution to the decision boundary of a specific model. This is different from our task of characterizing the properties inherent in the data and then using it to stratify the examples into subgroups.

Another key difference lies in the computational requirements, where both Influence functions and Data Shapley are highly compute intensive, on top of the model training. By contrast, Data-IQ's characterization can be obtained simply by virtue of training an ML model.

## A.2 Data-IQ adapted to any model

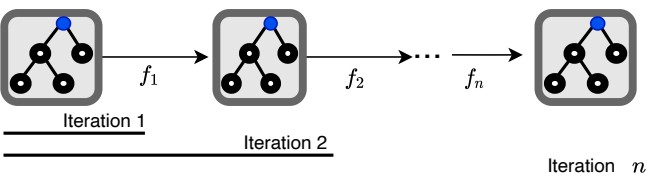

Figure 10: Pseudo-ensemble for GBDTs for usage with Data-IQ, where each sub-model is a checkpoint

Data-IQ's formulation allows us to use it with *any* ML model training in stages (i.e. iterative learning) such as gradient boosting decision trees (GBDTs) methods - such as XGBoost, LightGBM etc. We simply need a set of checkpoints through training.

This property is incredibly important for practical utility, since methods such as GBDTs or XGBoost are widely used by practitioners due to their performant nature on tabular data (often outperforming neural networks). GBDTs iteratively combines weak models and has shown great success on tabular data, often outperforming neural networks. Hence, having our method applicable for GBDTs in addition to neural networks adds to the broad utility.

Before outlining why methods such as GBDT's fit the Data-IQ pardigm, we provide a brief overview. Formally, given a dataset $\mathcal{D}_{\text{train}}$, GBDT iteratively constructs a model $F : X \to \mathbb{R}$ to minimize the empirical risk. At each iteration $e$ the model is updated as:

$$F^{(e)}(x) = F^{(e-1)}(\mathbf{x}) + \epsilon h^{(e)}(\mathbf{x}), \tag{4}$$

We now provide guidance on how to apply Data-IQ to such methods. Naively, we could construct the checkpoints as an ensemble of multiple independent GBDT's. However, this is inefficient as the space and time complexity scales with $N$ models. To avoid increasing the overhead of a single model (from a practitioner perspective), we create a pseudo-ensemble using a single GBDT, see Figure 10.

Similar to neural networks, the iterative learning nature of a GBDT means that the sequential submodels can be considered as checkpoints. Formally, each sub-model has parameters $\theta^{(i)}$, hence the ensemble of checkpoints can be described as $\Theta = \{\theta^{(i)}, 1 \leq i \leq N\}$.

We then apply Data-IQ as normal to the checkpoints $(\theta_1, \theta_2...\theta_E)$. The flexibility of this approach is that it applies both to training a new model, but interestingly, we can also apply this to an ALREADY trained model by looping through the structure to create the pseudo-ensemble.

## A.3 Why does Data-IQ produce a bell-shape?

We now theoretically analyze why Data-IQ has a bell-shaped curve. We show if the x-axis is aleatoric uncertainty and if the y-axis is predictive confidence, that we always obtain a bell-shaped curve.

Recall two terms from our formulation: (1) $\mathcal{P}(x, \theta)$: the model's predictive confidence for an input $x$ applied to a model with parameters $\theta$, (2) $\bar{\mathcal{P}}(x)$: the model's average confidence for the ground-truth class. Now assume $\mathcal{P}(x, \theta)$ is peaked around $\bar{\mathcal{P}}(x)$, as shown in Figure 11 below.

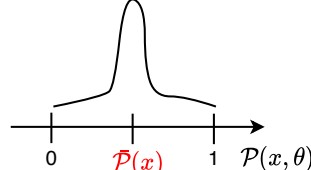

Figure 11: $\mathcal{P}(x, \theta)$ peaked around $\bar{\mathcal{P}}(x)$

Based on this, when we compute Data-IQ and obtain the following axes. This results in Figure 12 (a) with the bell-shape, which matches what we observe in practice (Figure 12 (b))

- y-axis: $1/E \sum_{e=1}^{E} \mathcal{P}(x, \theta_e) \approx \bar{\mathcal{P}}(x)$
- x-axis: $\frac{1}{E} \sum_{e=1}^{E} \mathcal{P}(x, \theta_e)(1 - \mathcal{P}(x, \theta_e)) \approx \bar{\mathcal{P}}(x)[1 - \bar{\mathcal{P}}(x)]$

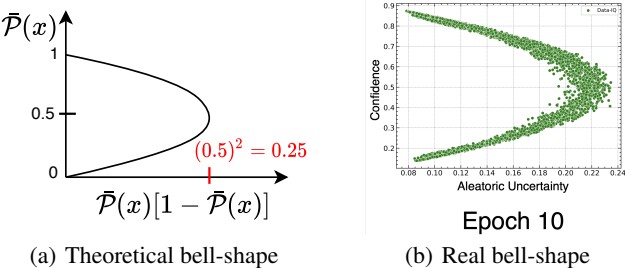

(a) Theoretical bell-shape      (b) Real bell-shape

Figure 12: The bell-shapes derived theoretically and practically match.

## A.4  Evaluating Data-IQ and Data Maps over stages

**Goal.**  Recall that we utilize *any* ML model trained in stages (e.g. epochs or iterations), for both Data-IQ and Data Maps. In the main paper, we train our models to convergence, with early stopping on a validation set. As is common practice, early stopping is done to prevent the model from overfitting. Once early stopping kicks in and we stop model training, we then compute the Data-IQ and Data Maps, averaged over all previously checkpointed models.

This brings two questions to mind, that we seek to answer:

1. How do Data-IQ and Data Maps differ at various stages (epochs)?
2. What happens if we don't do early stopping and let the model overfit?

Using Figure 13, we answer both questions. For this we train a neural network and around 10 epochs early stopping would normally kick in. However, instead of using early stopping, we continue training the neural network until 20 epochs, thereby letting the model overfit.

**Takeaway.**

1. **Differences across epochs:** Data Maps changes significantly over the course of training, This is due to its sensitivity as the model parameters are changed/updated through training. This sensitivity is reflected in the variability (epistemic uncertainty) on the x-axis. In contrast, Data-IQ is less sensitive and rapidly converges. The rationale is that Data-IQ models the inherent uncertainty in the data (aleatoric), thereby being more stable and less sensitive.

2. **Model overfits:** An interesting phenomenon occurs after the early stopping phase and into the overfitting stage. DataIQ remains consistent/stable, as expected. It is during this overfitting stage that the Data Maps shape changes, and begins to look similar to Data-IQ. The reason for this is that after $\approx$10 epochs (when the model begins to overfit), the model parameters do not get updated much (the loss doesn't change). The small changes of model parameters are then reflected by the epistemic/model uncertainty not changing much (x-axis of Data Maps). The minimal change means that when we average at the end of a longer training - the effects get washed out. This washing out of the effect causes the change in shape by Epoch 20. We will delve into this point in greater detail in the next sub-section as it explains the difference of Data Maps in our setting as compared to other settings.

## A.5  Data Maps: difference in our setting

**Goal.**  We now take a deep dive, based on the interesting findings at the model ovefitting stage, where the Data Maps shape changes. We believe that the difference of our setting from the original

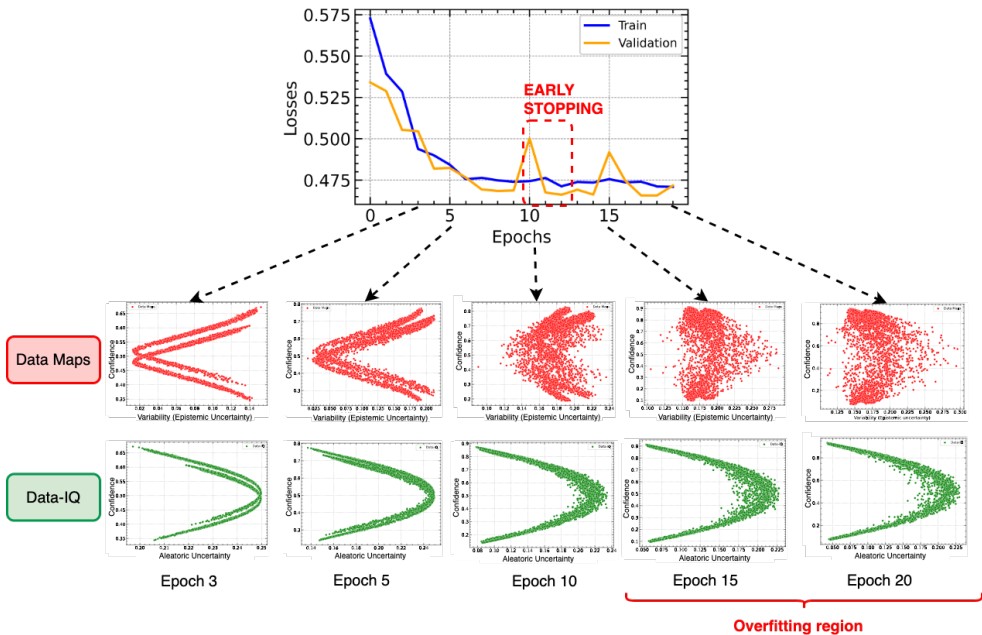

Figure 13: Evaluation of Data-IQ and Data Maps over various epochs (stages), even letting the model overfit. Data-IQ is more stable, whereas Data Maps changes shape after overfitting which is undesirable.

NLP setting of Data Maps provides a clear explanation for why the shape we observe for Data Maps is different from the shape observed in the original paper [24].

In typical NLP settings, the models are larger and highly parameterized. Hence, the models are trained for a large number of epochs (sometimes hundreds). However, in the tabular setting, models need to train for significantly fewer (as is evident by the early stopping around $\approx 10$ epochs. Further, our models are much shallower compared to highly parameterized NLP models, motivating why our setting requires early stopping to prevent overfitting.

**Takeaway.** We now begin our deep dive. As we saw in Figure 13, after around 10 epochs, the training loss plateaus. This means that subsequently the model parameters do not get updated much over training. The result of this is the epistemic uncertainty would have much less variability after 10 epochs. Thus, when we average the epistemic uncertainty for longer epochs these minimal changes would wash out effects from earlier in training.

This effect over longer training, explains why the shape of Data Maps is different in the NLP settings with large number of epochs and, assuming, we let the model overfit in our setting.

What implications does this have? For this we assess the evolution of three examples where we know their ground truth subgroup (*Easy*, *Ambiguous*, *Hard*).

First, we assess Data Maps which uses epistemic uncertainty. Figure 14 shows the evolution of epistemic uncertainty (Data Maps) for the three subgroup examples during training. We see that during model training (before early stopping), that the *Easy* and *Hard* examples, in fact, have higher epistemic uncertainty than the *Ambiguous* example. This would lead us to the incorrect conclusion that the *Ambiguous* example is Easy.

In practice, we do not want to let our model knowingly overfit. However, we can only obtain the correct characterization for Data Maps if we let the model overfit and wash out the earlier effects. This highlights a limitation of Data Maps vs Data-IQ.

We contrast this with Data-IQ which uses the Aleatoric uncertainty. We not only have consistent behavior for the *Ambiguous* examples, but also irrespective of whether we use early stopping or let the model overfit, we can obtain the correct characterization of the examples.

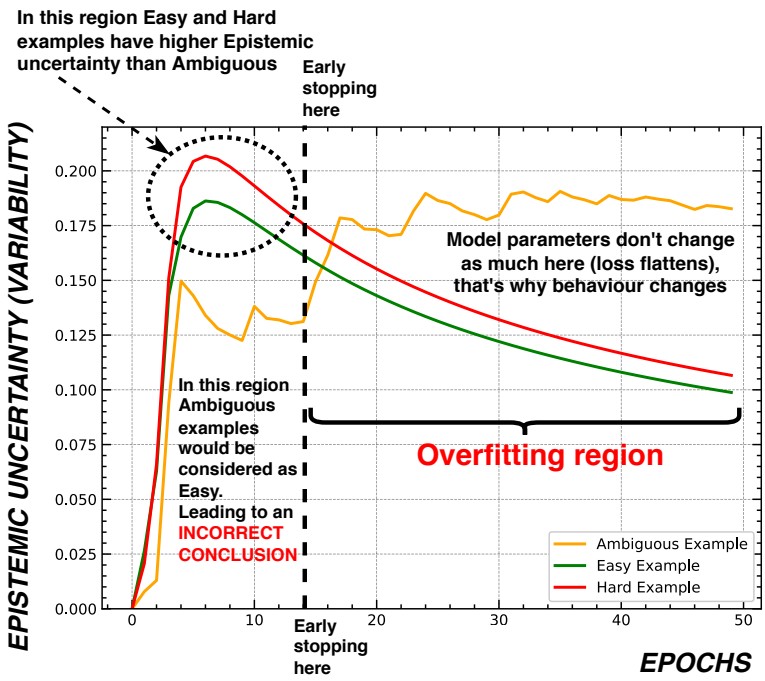

Figure 14: Data Maps: Epistemic Uncertainty (variability) evolution over epochs per subgroup

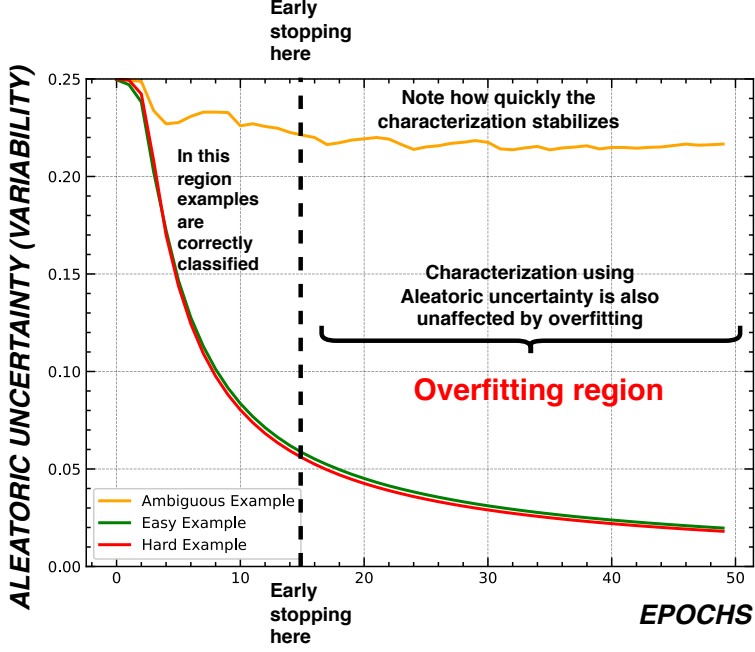

Figure 15: Data-IQ: Aleatoric Uncertainty evolution over epochs per subgroup

Finally, we look at the Data Maps shapes at the point where we perform early stopping (Figure 16(a)) and at the end of training if we let the model severely overfit (Figure 16(b)).

Figure 16 shows we can only obtain the expected shape from the original paper [24], if we let the model overfit. This is of course not an ideal solution, as in practice we do not want to overfit our

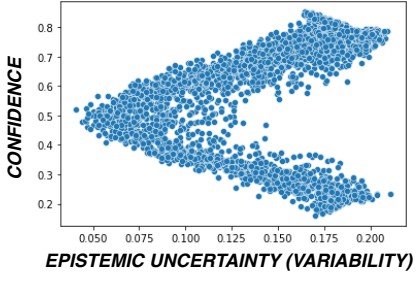

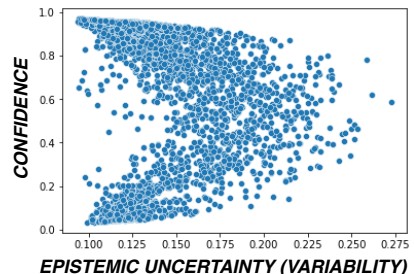

(a) Data Maps @ 10 epochs (Correct model)  (b) Data Maps @ 30 epochs (OVERFIT model)

Figure 16: Comparison of Data Maps when model is just right and overfits, illustrating how the shape changes

models by training for extra epochs. Especially in our tabular setting with less parameterized models, on smaller tabular datasets.

This motivates further why indeed we do require Data-IQ and why it's characterization of examples based on aleatoric uncertainty is more useful and principled.

## A.6 Data-IQ thresholds

As outlined in Section 3.3, we stratify samples based on the Aleatoric uncertainty and predictive confidence.

Typically, for aleatoric uncertainty we set the percentile $P_n = 50$th percentile. The rationale for this is any example where the aleatoric uncertainty is greater than the 50th percentile should be considered *Ambiguous*. This is based on the definition of *Ambiguous* examples. Of course, this threshold can be updated by practitioners.

This means that the examples below $P_{50}$ of the aleatoric uncertainty are then *Easy* or *Hard*. To differentiate the two, we look at the predictive confidence. We consider examples with high confidence, on the true label as *Easy* and those with low confidence, on the true label as *Hard*. This means that we need to practically set $C_{\text{up}}$ and $C_{\text{low}}$.

We define a threshold $thresh \in \{0, 0.5\}$ such that $C_{\text{up}} = 1 - thresh$ and $C_{\text{low}} = thresh$. An obvious threshold=0.25 such that $C_{\text{up}} = 0.75$ and $C_{\text{low}} = 0.25$. However, we present a practical method that could be applied to determine this threshold. We note that from our characterizations, examples with high aleatoric uncertainty are often uncertain about their predicitive confidence. Hence, they never have very high or low uncertainty, rather the uncertainty is around 0.5. However, since it is never exactly 0.5 this motivates a band namely $C_{\text{up}}$ and $C_{\text{low}}$.

Now assume for any dataset, we train a model and apply Data-IQ. We can then sweep $thresh \in \{0, 0.5\}$ and assess the proportion of examples in each group.

We show results in Figure 17 below as we sweep the threshold. For low threshold values, the proportions are, of course, stable (selecting the most confident samples). However, there is a threshold value beyond which where the curve rises until a threshold knick point. After that knick-point there is stability once again. We propose that this knick-point threshold of stability be used as the threshold in practice. The rationale is that this will allow for better consistency of data characterization.

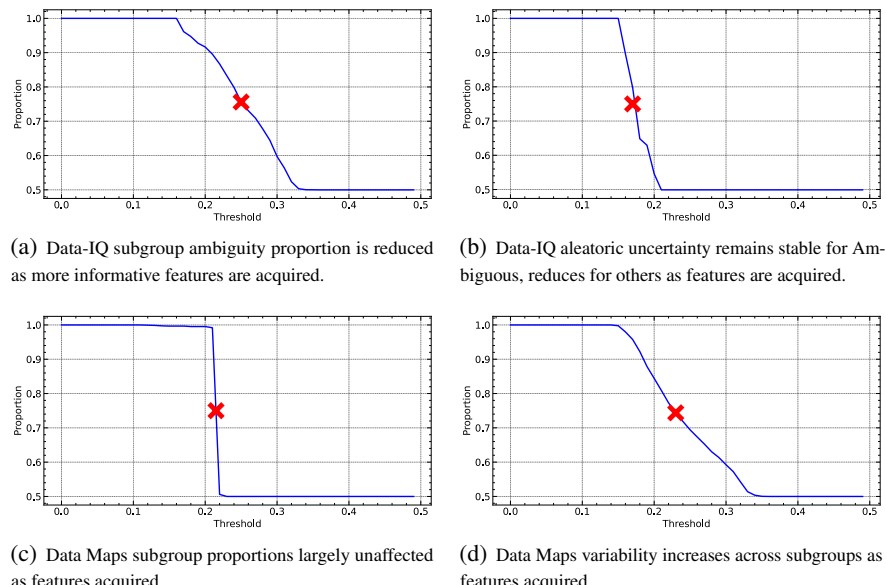

(a) Data-IQ subgroup ambiguity proportion is reduced as more informative features are acquired.

(b) Data-IQ aleatoric uncertainty remains stable for Ambiguous, reduces for others as features are acquired.

(c) Data Maps subgroup proportions largely unaffected as features acquired.

(d) Data Maps variability increases across subgroups as features acquired.

Figure 17: SUPPORT Quantifying the value of feature acquisition based on change in ambiguity. Only Data-IQ captures this relationship.

## A.7  Data-IQ subgroup characterization

With Data-IQ, we can obtain a plot of the data where the x-axis represents aleatoric uncertainty and the y-axis predicitive confidence for the true label. As discussed this means we obtain a bell-shaped characterization. Our goal is highlight where on the plot each subgroup lies. i.e. illustrate the location of $Easy$, $Ambiguous$ and $Hard$ on the plot.

We highlight in Figure 18, that $Ambiguous$ examples more likely to lie on the right side with higher aleatoric uncertainty. We note that based on our characterization this region of the plot never has high and low predictive uncertainty. Rather, it typically only has uncertainty approximately around 0.5.

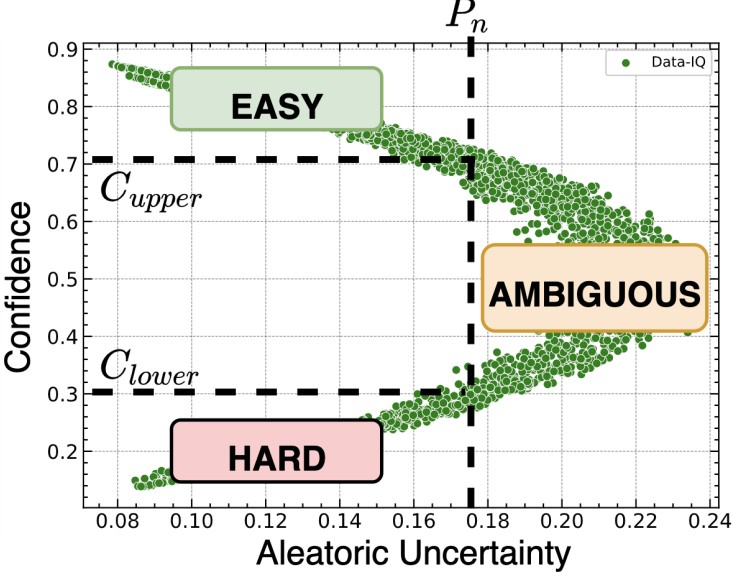

Figure 18: $\mathcal{P}(x, \theta)$ peaked around $\bar{\mathcal{P}}(x)$

# B Benchmarks & Experimental Details

## B.1 Benchmarks, Datasets and Implementation details

### B.1.1 Data-IQ

The description of Data-IQ is detailed in the main paper, where we stratify examples based on aleatoric uncertainty and predictive confidence.

**Implementation details.** We implement Data-IQ via a class that is versatile and can plug into any framework including Pytorch, Tensorflow/Keras, Skorch and Scikit-learn style APIs. The rationale is that since Data-IQ can be used with any model, the functionality should be equally plug and play from a usability perspective. Note that for libraries including Pytorch, Tensorflow/Keras, Skorch, we can easily include Data-IQ simply by adding it to the training callbacks of the respective training loops.

### B.1.2 Data-IQ algorithm

Algorithm 1 below presents the Data-IQ algorithm in more detail.

---

**Algorithm 1:** Data-IQ Algorithm

---

**Data:** Dataset $\mathcal{D}$
**Result:** Subgroup assignment $g$ per sample $x$, i.e. $g(x, \mathcal{D}_{\text{train}})$
**Inputs:** $C_{\text{up}} = 0.75$, $C_{\text{low}} = 0.25$, $P_{50}$ = 50th percentile
**Training:** Train model ($f$) for $E$ epochs. At each epoch $e$, compute $\mathcal{P}(x, \theta_e)$ for datapoint $x$, where $\theta_e$ is the model parameters at epoch $e$.
**Characterize samples into subgroups ($g(x, \mathcal{D}_{\text{train}})$): for** $(x_i, y_i)$ *in* $\mathcal{D}_{train}$ **do**
> **Metrics:** $\bar{\mathcal{P}}(x_i) = 1/E \sum_{e=1}^{E} \mathcal{P}(x_i, \theta_e)$;
> $v_{\text{al}}(x_i) = \frac{1}{E} \sum_{e=1}^{E} \mathcal{P}(x_i, \theta_e)(1 - \mathcal{P}(x_i, \theta_e))$
> **if** $\bar{\mathcal{P}}(x_i) \geq C_{\text{up}} \ \wedge \ v_{\text{al}}(x_i) < P_{50}\left[v_{\text{al}}(\mathcal{D}_{train})\right]$ **then**
> > Assign *Easy* subgroup
> **else if** $\bar{\mathcal{P}}(x_i) \leq C_{\text{low}} \ \wedge \ v_{\text{al}}(x_i) < P_{50}\left[v_{\text{al}}(\mathcal{D}_{train})\right]$ **then**
> > Assign *Hard* subgroup
> **else**
> > Assign *Ambiguous* subgroup
> **end**

**end**

---

$$g(x, \mathcal{D}_{\text{train}}) = \begin{cases} \text{Easy} & \text{if } \bar{\mathcal{P}}(x) \geq C_{\text{up}} \ \wedge \ v_{\text{al}}(x) < P_{50}\left[v_{\text{al}}(\mathcal{D}_{\text{train}})\right] \\ \text{Hard} & \text{if } \bar{\mathcal{P}}(x) \leq C_{\text{low}} \ \wedge \ v_{\text{al}}(x) < P_{50}\left[v_{\text{al}}(\mathcal{D}_{\text{train}})\right] \\ \text{Ambiguous} & \text{otherwise} \end{cases} \tag{5}$$

### B.1.3 Data Maps

Similar to Data-IQ; Data Maps has been detailed in the main paper.

**Implementation details.** Our implementation of Data Maps can be used with the flexible approach we built for Data-IQ. We adapt the implementation of Data Maps from [4].

### B.1.4 AUM

The Area-under-the-Margin (AUM) metric first computes the margin for data $x, y$ at epoch $t$.

$$M^{(t)}(x, y) = \overbrace{z_y^{(t)}(x)}^{\text{assigned logit}} - \overbrace{\max_{i \neq y} z_i^{(t)}(x)}^{\text{largest other logit}}. \tag{6}$$

---

[4] https://github.com/allenai/cartography

Over all epochs the AUM is then computed as the average of these margin calculations.

$$AUM(x, y) = \frac{1}{T} \sum_{t=1}^{T} M^{(t)}(x, y), \tag{7}$$

**Implementation details.** The benchmark is based on [6] and we use the implementation from [5].

### B.1.5 GraNd

The gradient norm at epoch $t$ for an input $x, y$ is computed as :

$\chi_t(x, y) = \mathbb{E} \left\| \sum_{k=1}^{K} \boldsymbol{\nabla}_{f^{(k)}} \ell \left( f_t(x), y \right)^T \psi_t^{(k)}(x) \right\|_2$

where $\psi_t^{(k)}(x) = \boldsymbol{\nabla}_{\mathbf{w}_t} f_t^{(k)}(x)$.

**Implementation details.** The benchmark is based on [28]. We adapt the Jax implementation from [6] to Pytorch with the help of [7].

### B.1.6 JTT

The Just-Train-Twice (JTT) method is a two-stage method. For a model, trained via ERM, it first identifies errors in the training examples during training $i.e. y \neq \hat{y}$.

The second step is to train a final model, where the error samples are increased via weighting.

$$L_{\text{up-ERM}}(\theta, E) = (\lambda \sum_{(x,y) \in E} \ell(x, y; \theta) + \sum_{(x,y) \notin E} \ell(x, y; \theta)), \tag{8}$$

where $\lambda$ is the weighting factor and $E$ errors.

The goal is that the weighting factor will upsample the samples with errors and by training on this augmented dataset it will aid to improve worst-group performance.

**Implementation details.** Our benchmark is based on [1] and we adapt the implementation from [8].

### B.2 Datasets

We describe the four real-world medical datasets in greater detail.

**SEER Dataset** The SEER dataset is a publicly available dataset consisting of 240,486 patients enrolled in the American SEER program [39]. The dataset consists of features used to characterize prostate cancer: including age, PSA (severity score), Gleason score, clinical stage, treatments etc. A summary of the covariate features can be found in Table 4. The classification task is to predict patient mortality, which is binary label $\in \{0, 1\}$.

The dataset is highly imbalanced, where $94\%$ of patients survive. Hence, we extract a balanced subset of of 20,000 patients (i.e. 10,000 with label=0 and 10,000 with label=1).

Table 4: Summary of features for the SEER Dataset [39]

| Feature | Range |
| --- | --- |
| Age | $37 - 95$ |
| PSA | $0 - 98$ |
| Comorbidities | $0, 1, 2, \geq 3$ |
| Treatment | Hormone Therapy (PHT), Radical Therapy - RDx (RT-RDx),Radical Therapy -Sx (RT-Sx), CM |
| Grade | $1, 2, 3, 4, 5$ |
| Stage | $1, 2, 3, 4$ |
| Primary Gleason | $1, 2, 3, 4, 5$ |
| Secondary Gleason | $1, 2, 3, 4, 5$ |

---

[5] https://github.com/asappresearch/aum    [6] https://github.com/mansheej/data_diet

[7] https://github.com/cybertronai/autograd-lib    [8] https://github.com/anniesch/jtt

**CUTRACT Dataset** The CUTRACT dataset is a private dataset consisting of 10,086 patients enrolled in the British Prostate Cancer UK program [40]. Similar, to the SEER dataset, it consists of the same features to characterize prostate cancer. In addition, it has the same task to predict mortality. A summary of the covariate features can be found in Table 5.

Once again, the dataset is highly imbalanced, hence we then choose extract a balanced subset of of 2,000 patients (i.e. 1000 with label=0 and 1000 with label=1).

Table 5: Summary of features for the CUTRACT Dataset [40]

| Feature | Range |
|---------|-------|
| Age | $44 - 95$ |
| PSA | $1 - 100$ |
| Comorbidities | $0, 1, 2, \geq 3$ |
| Treatment | Hormone Therapy (PHT), Radical Therapy - RDx (RT-RDx),Radical Therapy -Sx (RT-Sx), CM |
| Grade | $1, 2, 3, 4, 5$ |
| Stage | $1, 2, 3, 4$ |
| Primary Gleason | $1, 2, 3, 4, 5$ |
| Secondary Gleason | $1, 2, 3, 4, 5$ |

**Fetal Dataset** The Fetal dataset [42] is a publicly available data set consisting of 2126 patients who underwent fetal cardiotograms (CTG). The patients had specific diagnostic features measured and the CTGs were classified by 3 expert obstertricians (with a consensus label) to assign three fetal states (normal, suspect and pathologic). Our goal is to predict these three fetal states, hence the task is a multi-class classification problem. A summary of the covariate features can be found in Table 6.

Table 6: Summary of features for the Fetal Dataset [42]

| Feature | Range |
|---------|-------|
| LB (FHR baseline) | $106 - 160$ |
| AC (Acceleration/sec) | $0 - 260$ |
| FM (fetal movements/sec) | $0 - 564$ |
| UC (uterine contractions/sec) | $0 - 23$ |
| ASTV (% of time with abnormal short term variability) | $12 - 87$ |
| ALTV (% of time with abnormal long term variability) | $0 - 91$ |
| MLTV (mean value of long term variability) | $0 - 51$ |
| DS (severe deceleration/sec) | $0 - 1$ |
| DP (prolonged deceleration/sec) | $0 - 4$ |
| Nzeros (n histogram zeros) | $0 - 10$ |
| Variance (histogram variance) | $0 - 269$ |
| Tendency (histogram tendency) | $0 - 1$ |
| NSF (fetal class code) | $1(Normal), 2(suspect), 3(pathologic)$ |

**Covid-19 Dataset** The Covid-19 dataset [38] consists of Covid patients from Brazil. The dataset is publicly available as it is based on SIVEP-Gripe data. The dataset consists of 6882 patients from Brazil recorded between Februrary 27-May 4 2020. The dataset captures risk factors including comorbidities, symptoms, and demographic characteristics. There is a mortality label from Covid-19 making it a binary classification task. A summary of the characteristics of the covariates can be found in Table 7.

**Support Dataset** The Support [41] is a private dataset which examines outcomes for seriously ill hospitalized patients consisting of 9105 patients. The features include demographics, medical history, clinical variables, and laboratory variables. Our goal is to predict patient mortality. A summary of the covariate features can be found in Table 8.

### B.3 Additional experiment details

We note that all experiments were performed using a single Nvidia Tesla P100 GPU.

Table 7: Summary of features for the Covid-19 Dataset [38]

| Feature | Range |
| --- | --- |
| Sex | 0 (Female), 1(Male) |
| Age | $1 - 104$ |
| Fever | $0, 1$ |
| Cough | $0, 1$ |
| Sore throad | $0, 1$ |
| Shortness of breath | $0, 1$ |
| Respiratory discomfort | $0, 1$ |
| SPO2 | $0 - 1$ |
| Diharea | $0, 1$ |
| Vomitting | $0, 1$ |
| Cardiovascular | $0, 1$ |
| Asthma | $0, 1$ |
| Diabetes | $0, 1$ |
| Pulmonary | $0, 1$ |
| Immunosuppresion | $0, 1$ |
| Obesity | $0, 1$ |
| Liver | $0, 1$ |
| Neurologic | $0, 1$ |
| Branca (Region) | $0, 1$ |
| Preta (Region) | $0, 1$ |
| Amarela (Region) | $0, 1$ |
| Parda (Region) | $0, 1$ |
| Indigena (Region) | $0, 1$ |

### B.3.1 Section 4.1. Robustness to variation experiment details.

We assess robustness to variation across different models and/or parameterizations. The rationale for this is that these are model changes or perturbations that different practitioners might easily make. However, we desire consistency and stability in the metrics such that the data insights remain consistent. For layers we evaluate a: 3 layer MLP, 4 layer MLP and 5 layer MLP. For number of hidden units we define two types: type 1 has 64 units in layer one, which reduce by half per layer and type 2 has 256 units in layer one, which reduce by half per layer. Finally, we evaluate these combinations with both Adam as the optimizer during training. These models perform at a similar level (i.e. accuracy). We then assess the consistency of the metrics for all examples using these different models/parameterizations based on Spearman rank correlation.

### B.3.2 Section 4.1. Data insights from subgroups experiment details.

For the data insights experiment, we perform clustering on each subgroup identified by Data-IQ. We make use of Gaussian Mixture Models (GMM), similar to [5]. The difference is the sub-space in which the clustering is applied. In Data-IQs case, we cluster within each subgroup identified by Data-IQ. For each supergroup, we search over $k \in 2, \dots, 10$. We choose the $k$ value that yields the highest average Silhouette score. Our implementation is based on [9]

### B.3.3 Section 4.2. Principled feature acquisition experiment details.

We assess the potential for principled feature acquisition using a semi-synthetic experiment. We assess how valuable each feature is based on the Pearson correlation of the feature with the target. We then rank sort the correlations from lowest to highest. This ordering is then used to determine the order in which to "acquire" and assess the features.

### B.3.4 Section 4.2. Principled dataset comparison experiment details.

As described in the main paperm we compare two synthetic datasets using two models representing two synthetic data vendors: namely CTGAN and Gaussian Copula. We use the implementations of [10].

---

[9] https://scikit-learn.org   [10] https://github.com/sdv-dev/SDV

Table 8: Summary of features for the Support dataset [41]

| Feature | Range |
| --- | --- |
| Sex | 0 (Female), 1(Male) |
| Age | $18 - 102$ |
| Number of comorbities | $0 - 9$ |
| ARF/MOSF w/Sepsis | $0, 1$ |
| COPD | $0, 1$ |
| CHF | $0, 1$ |
| Cirrhosis | $0, 1$ |
| Coma | $0, 1$ |
| Colon Cancer | $0, 1$ |
| Lung cancer | $0, 1$ |
| ARF/MOSF w/Malig | $0, 1$ |
| Cancer | $0, 1$ |
| under \$11k | $0, 1$ |
| \$11k-\$25k | $0, 1$ |
| \$25k-\$50k | $0, 1$ |
| >\$50k | $0, 1$ |
| white | $0, 1$ |
| black | $0, 1$ |
| asian | $0, 1$ |
| hispanic | $0, 1$ |
| yeas of education | $0 - 31$ |
| avg TISS | $1 - 83$ |
| diabetes | $0, 1$ |
| dementia | $0, 1$ |
| mean bp | $0 - 195$ |
| white blood cell count | $0 - 200$ |
| heart rate | $0 - 300$ |
| respiratory rate | $0 - 90$ |
| temperature | $31 - 42$ |
| pafi | $12 - 890$ |
| albumnin | $0.39 - 29$ |
| bilirubin | $0.09 - 63$ |
| creatinine | $0.09 - 63$ |
| sodium | $6 - 181$ |
| ph | $6.8 - 7.81$ |
| glucose | $0 - 1092$ |
| bun | $1 - 300$ |
| urine output | $0 - 9000$ |
| adlp | $0 - 7$ |
| adls | $0 - 7$ |

### B.3.5  Section 4.3. Less is more experiment details.

For this experiment we note that we train multiple baseline models to evaluate the generalization based on dataset sculpting. We evaluated three different XGBoost models with 100,150 and 200 estimators each. Our XGBoost implementation uses the Python version from [11].

### B.3.6  Section 4.3. Group-DRO experiment details.

Group DRO uses training group annotations to directly minimize the worst-group error on the training set. Hence, we compare different methods to obtain the group annotations. This experiment compares Group-DRO applied to groups identified by Data-IQ and as a baseline George [5]. We also compare this to Just-Train-Twice (JTT) [1]. We use the implementation of Group-DRO from [12], which is based on [53]. Our George benchmark is based on the implementation from [13] and the JTT benchmark is based on the implementation from [14] . All code is in Pytorch.

The group DRO objective can then be written as:

$$L_{\text{group-DRO}}(\theta) = \max_{g \in G} \frac{1}{n_g} \sum_{i|g_i=g} \ell(x_i, y_i; \theta) \tag{9}$$

---

[11]  https://github.com/dmlc/xgboost          [12]  https://github.com/HazyResearch/hidden-stratification

[13]  https://github.com/HazyResearch/hidden-stratification    [14]  https://github.com/anniesch/jtt

where $n_g$ is the number of training points with group $g_i = g$.

### B.3.7 Section 4.3. Subgroup informed usage of uncertainty estimation experiment details.

As mentioned in the main paper we obtain uncertainty estimates using a Bayesian Neural Network based on [59]. Specifically, we train a 5-layer MLP model. A Gaussian prior is placed over the weights and we optimize the KL divergence during training. Our benchmark is based on [59] and we use the implementation from [15].

---

[15]  https://github.com/IBM/UQ360

## C Additional Experiments

### C.1 Understanding the representation space

**Goal.** At deployment/testing time, we stratify the examples into subgroups using the representation space. of the training examples. Our goal is to illustrate the points made in Section 3.3 with respect to the representation space

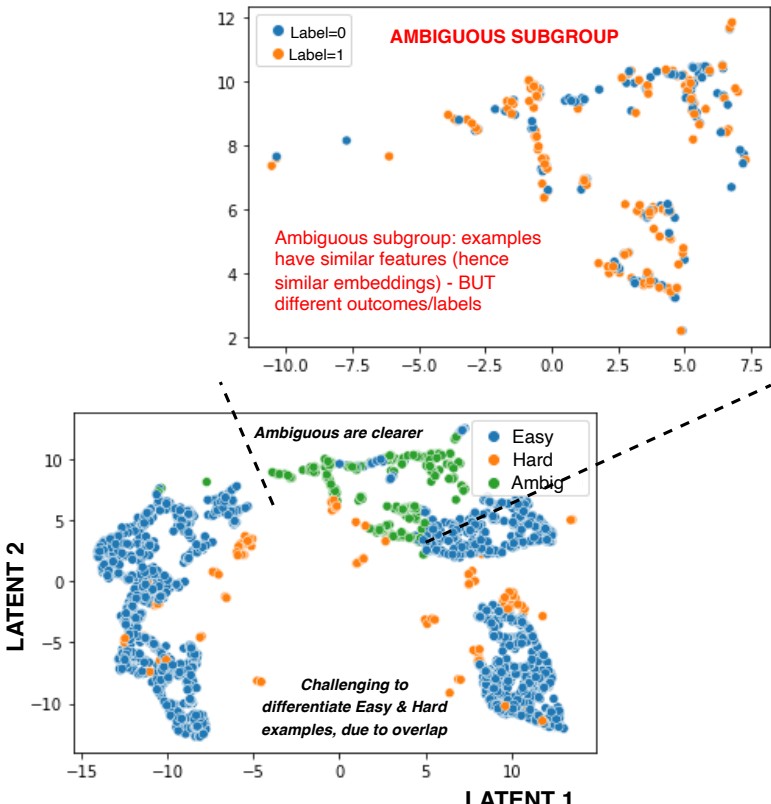

Figure 19: Representation space of training examples broken down by sub-group. We then zoom into the Ambiguous subgroup and illustrate the heterogeneous outcomes, where examples with similar features can have different outcomes.

**Takeaway.** We recall the two main points about the representation space made in the main paper and illustrate them using the representation space (i.e. UMAP embedding) shown in Figure 19.

1. *Ambiguous* examples have distinctive features and are clustered in embedding space. Thus, we can distinguish the *Ambiguous* examples using the embedding. We can clearly see this with the green points in Figure 19.

2. It is not possible to reliably distinguish *Easy* examples from *Hard* examples based on the embedding, because *Hard* examples are a minority with outcome randomness that have similar features, as the *Easy* examples. Naturally, since the Easy and Hard example embeddings are similar, they also have similar features. We can clearly see this where the orange examples (Hard) are randomly scattered with the Blue (Easy) examples, in Figure 19. This makes these two groups difficult to distinguish.

We also zoom in to the *Ambiguous* region (sub-figure of Figure 19). This illustrates the specific issue of outcome heterogeneity, where similar patients (with similar features) have different outcomes. This is illustrated by the different colors, which represent the different outcomes (i.e. label=0 is survive and label=1 is death).

## C.2  Data-IQ: Neural Networks vs Other model classes

**Goal.**   As discussed in the main text and Appendix A.2, Data-IQ can be used with *any* ML model trained in stages. Methods such XGBoost, LightGBM and CatBoost methods are widely used by practitioners on tabular data, often more so than neural networks [7]. While Data-IQ can naturally handle any such model, as long as there are checkpoints (over iterations), Data Maps naturally cannot (or at least it was never formulated to do so). However, to enable comparison, we can use Data Maps within our framework, so that it can be applied to a more general class of models.

Ideally, based on *P1*, we desire that the characterization of examples be consistent for similar performing models, irrespective of whether the model is a neural network or, for example, an XGBoost model.

To assess this the robustness of both Data-IQ and Data Maps, we train a neural network, XG-Boost, LightGBM and CatBoost models to achieve the same performance and then perform the characterization for all models. The evaluation of all data sets is shown in Figures 20-23.

**Takeaway.**   We can clearly see that Data-IQ has a similar characterization for across all four models. Contrastingly, for Data Maps, the characterizations are significantly different for the different model classes. The key difference is Data-IQs usage of Aleatoric uncertainty, rather than the Data Maps usage of Epistemic uncertainty. The implication of this result is that by Data-IQ capturing the uncertainty inherent to the data (aleatoric uncertainty), it leads to a more consistent and stable characterizations of the data itself. Especially, this highlights that Data-IQ characterizes the data in a manner that is not as sensitive to the choice of model when compared to Data Maps.

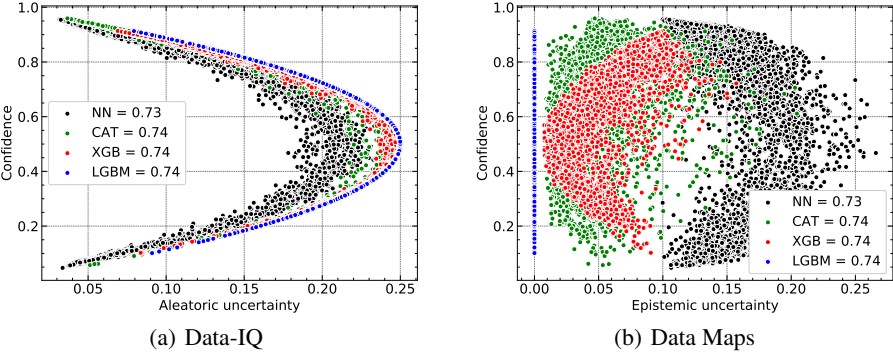

     (a) Data-IQ           (b) Data Maps

Figure 20: NN vs XGBoost comparison on the Covid-19 dataset, showing Data-IQ is more consistent

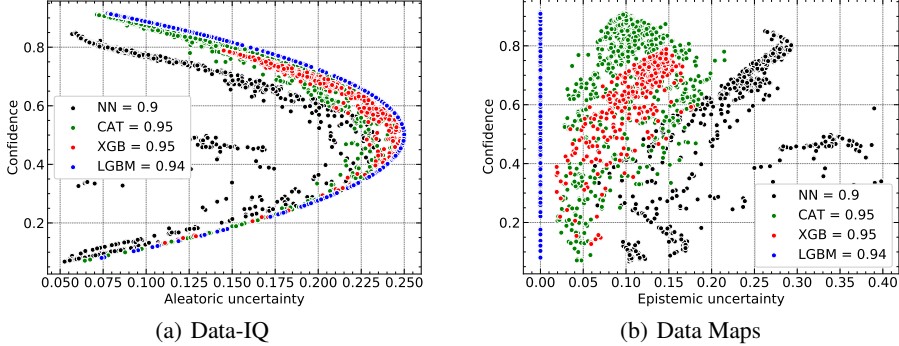

     (a) Data-IQ           (b) Data Maps

Figure 21: NN vs XGBoost comparison on the Fetal dataset, showing Data-IQ is more consistent

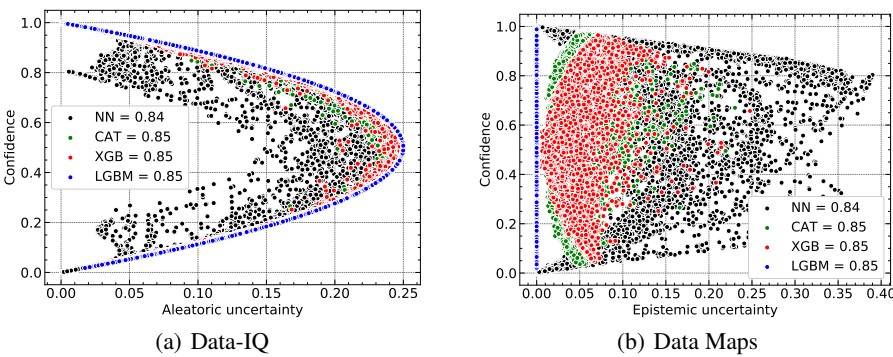

(a) Data-IQ                (b) Data Maps

Figure 22: NN vs XGBoost comparison on the Prostate dataset, showing Data-IQ is more consistent

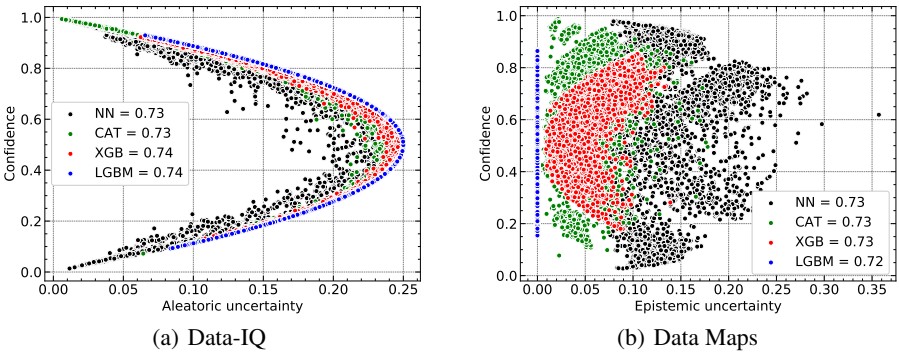

(a) Data-IQ                (b) Data Maps

Figure 23: NN vs XGBoost comparison on the Support dataset, showing Data-IQ is more consistent

To further augment these results, we quantitatively assess the consistency across model types. Similar to Section 4.1, we have compared the Spearman rank correlation between the metrics of all model combinations (neural networks, XGBoost, LightGBM, CatBoost). Note that all four models are trained to perform similarly on a held-out dataset.

Below, we present Table 9 showing the average and standard deviation. We see that Data-IQ has higher scores for all datasets, highlighting the better consistency compared to Data Maps.

Table 9: Comparison of robustness/consistency across different models on the basis of the Spearman rank correlation

| Dataset | (Ours) Data-IQ | Data Maps |
|---------|----------------|-----------|
| Covid | $0.85 \pm 0.05$ | $0.53 \pm 0.07$ |
| Support | $0.84 \pm 0.06$ | $0.48 \pm 0.04$ |
| Prostate | $0.84 \pm 0.11$ | $0.44 \pm 0.17$ |
| Fetal | $0.72 \pm 0.07$ | $0.45 \pm 0.06$ |

### C.3 Collecting more data - it might make things worse

**Goal.** Typically it is assumed that simply by collecting more data that most problems will be mitigated. We aim to assess the influence of adding more data samples, in contrast to previous experiments in the main paper on acquiring features. Especially, we want to show how, in the case of heterogeneous outcomes, *blindly* adding more data might not always be the solution.

**Experiment.** We sub-sample proportions of the dataset from 0-100% of the original size of the dataset. We then recompute the Data-IQ subgroups on each proportion, as more data examples are added. We show two examples in Figure 24, in which the subgroups evolve as more data is added.

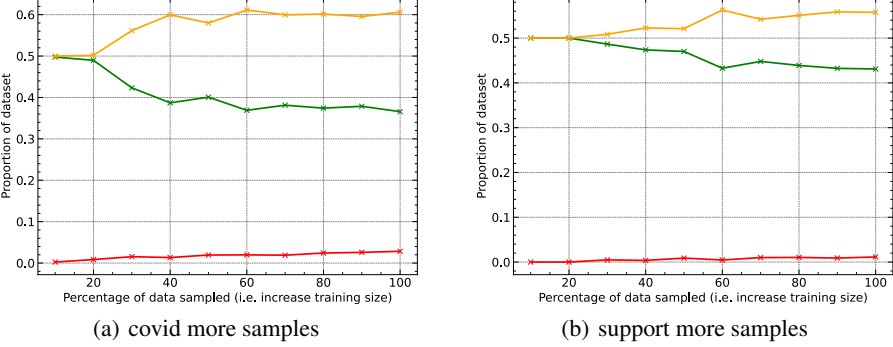

(a) covid more samples                      (b) support more samples

Figure 24: The *Ambiguous* subgroup proportion in fact increases as the dataset size increases, i.e. more examples are added. This is due to the increased probability of feature collisions.

**Takeaway.** As the number of data examples increases, the proportion covered by the *Ambiguous* subgroup increases. This is due to the increased probability of "feature collisions" as more examples are added. Therefore, the result shows that the simple solution, in which we naively collect more data, can sometimes do more harm.

### C.4 Identifying subsets in an outcome driven manner: assessing quality

**Goal.** Data-IQ identifies subgroups in an outcome-driven manner, compared to the conventional unsupervised manner. We compare the cluster quality on the outcome-driven subgroups as obtained by Data-IQ compared to other benchmarks, namely: George [5], clustering in the latent space, clustering in a PCA space, and clustering the raw feature space. All methods use a Gaussian Mixture model for consistency.

We evaluated these methods based on two well-known cluster quality metrics, namely, (1) Silhouette score (SIL) and (2) Davies-Bouldin (DB) score. The results for two example datasets can be seen in Table 10.

Table 10: Clustering results

| Dataset | Metric | Data-IQ (Ours) | Outcome latent space (George) | Latent space | PCA | Feature space |
|---------|--------|----------------|-------------------------------|--------------|------|---------------|
| Covid-19 | SIL ($\uparrow$) | 0.68 | 0.48 | 0.47 | 0.51 | 0.02 |
| | DB ($\downarrow$) | 0.42 | 0.87 | 0.78 | 0.81 | 5.7 |
| Prostate | SIL ($\uparrow$) | 0.80 | 0.6 | 0.56 | 0.57 | 0.54 |
| | DB ($\downarrow$) | 0.3 | 0.6 | 0.63 | 0.57 | 0.61 |

**Takeaway.** We can clearly see that Data-IQ has the highest SIL score and the lowest DB score. These results demonstrate that the outcome-driven clustering approach taken by Data-IQ, where the clusters are within the Data-IQ subgroups, results in higher quality clusters.

## C.5 Data Insights: additional results

**Goal.** The main text (Section 4.1) illustrated the data insights that can be obtained for a single dataset (Prostate). Here, we include the results for all other datasets (using the same setup). See Figures 25-27.

**Takeaway.** In general, for each dataset, the subgroups represent: (1) *Easy*: Severe patients with a death outcome, and less severe patients with a survival outcome, (2) *Ambiguous*: Patients with similar features, but different outcomes. This could suggest that the features we have at hand are insufficient to separate the differences in outcomes and (3) *Hard*: Severe patients with a survival outcome, and less severe patients with a death outcome. i.e. opposite outcomes as expected due to randomness in the outcomes.

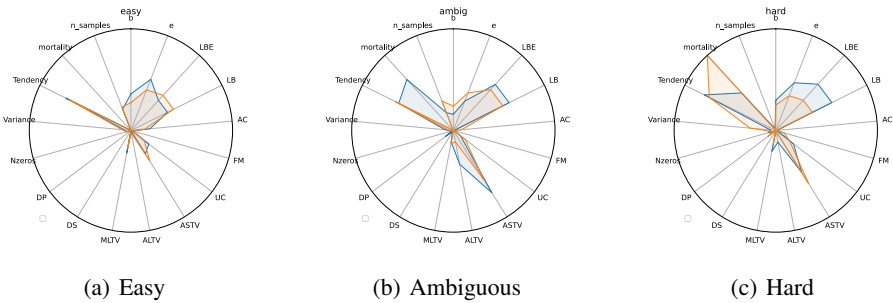

(a) Easy       (b) Ambiguous       (c) Hard

Figure 25: Fetal subgroups identified by Data-IQ (descriptions above). Colors represent the GMM clusters

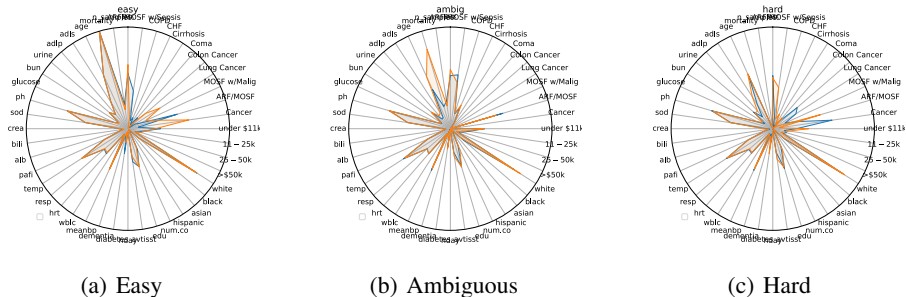

(a) Easy       (b) Ambiguous       (c) Hard

Figure 26: Support subgroups identified by Data-IQ (descriptions above). Colors represent the GMM clusters

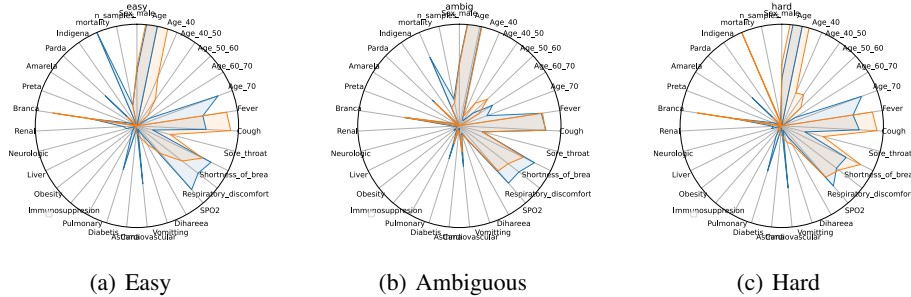

(a) Easy       (b) Ambiguous       (c) Hard

Figure 27: Covid subgroups identified by Data-IQ (descriptions above). Colors represent the GMM clusters

## C.6 Principled feature acquisition (P2): additional results

**Goal.** The main text (Section 4.2) illustrated principled feature acquisition using Data-IQ for a single dataset. Here, we include the results for all other datasets (using the same semi-synthetic setup, based on correlation of the feature with the target). Recall our formulation, that a valuable feature should decrease the ambiguity of an example, overall decreasing the proportion of *Ambiguous* examples in the dataset. This permits a principled approach to feature acquisition.

**Takeaway.** We see similar results to the main text for all datasets. For Data-IQ, we see that as we acquire "valuable" features, the proportion of the *Ambiguous* subgroup drops, while the proportion of the *Easy* subgroup increases. There are significant changes for the important features. This shows that Data-IQ's subgroup characterization can be used to quantify a feature's value, by its ability to decrease ambiguity. In contrast, Data Maps, shows minimal response to feature acquisition, suggesting that it may not be sensitive enough to capture the feature's value.

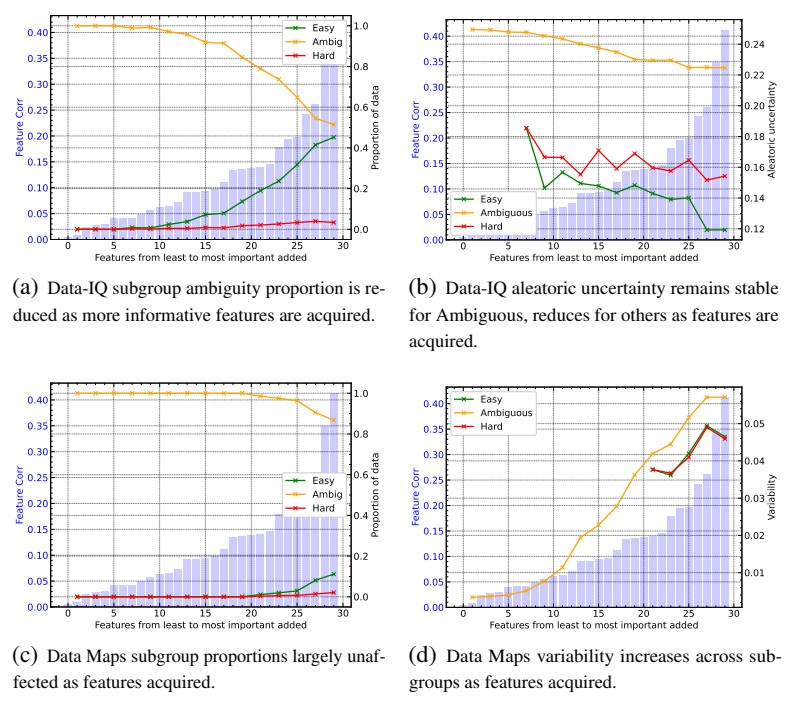

(a) Data-IQ subgroup ambiguity proportion is reduced as more informative features are acquired.

(b) Data-IQ aleatoric uncertainty remains stable for Ambiguous, reduces for others as features are acquired.

(c) Data Maps subgroup proportions largely unaffected as features acquired.

(d) Data Maps variability increases across subgroups as features acquired.

Figure 28: COVID Quantifying the value of feature acquisition based on change in ambiguity. Only Data-IQ captures this relationship.

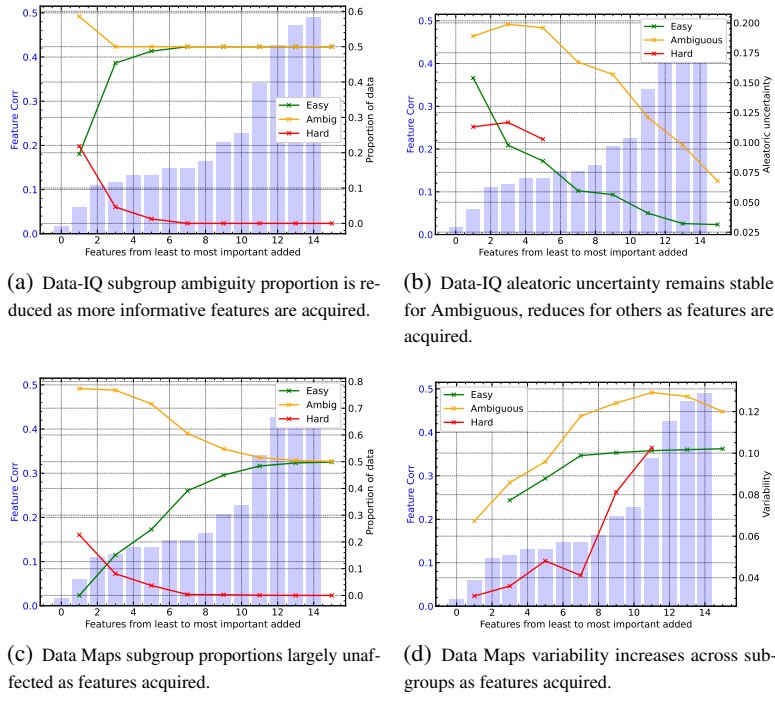

(a) Data-IQ subgroup ambiguity proportion is reduced as more informative features are acquired.

(b) Data-IQ aleatoric uncertainty remains stable for Ambiguous, reduces for others as features are acquired.

(c) Data Maps subgroup proportions largely unaffected as features acquired.

(d) Data Maps variability increases across subgroups as features acquired.

Figure 29: FETAL Quantifying the value of feature acquisition based on change in ambiguity. Only Data-IQ captures this relationship.

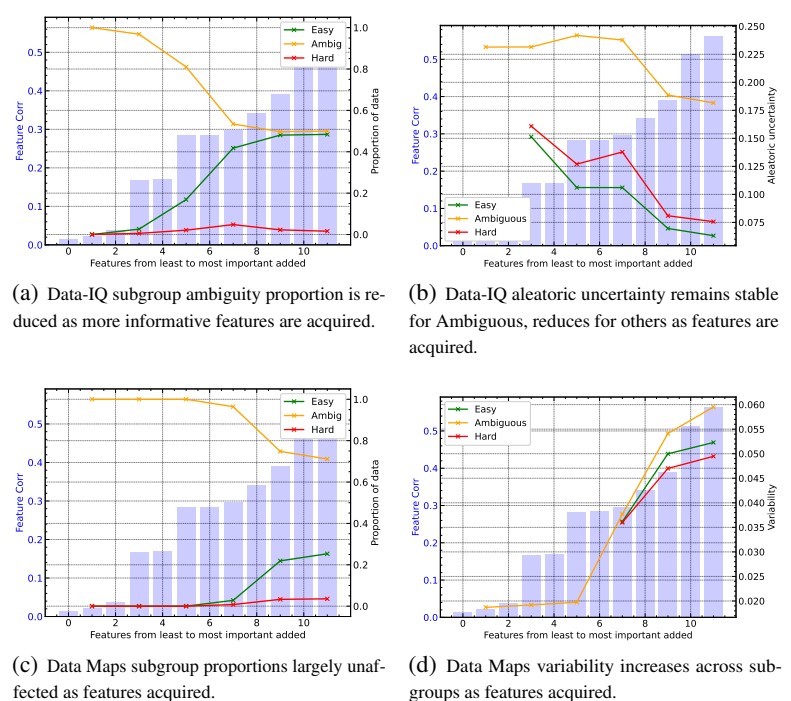

(a) Data-IQ subgroup ambiguity proportion is reduced as more informative features are acquired.

(b) Data-IQ aleatoric uncertainty remains stable for Ambiguous, reduces for others as features are acquired.

(c) Data Maps subgroup proportions largely unaffected as features acquired.

(d) Data Maps variability increases across subgroups as features acquired.

Figure 30: Prostate Quantifying the value of feature acquisition based on change in ambiguity. Only Data-IQ captures this relationship.

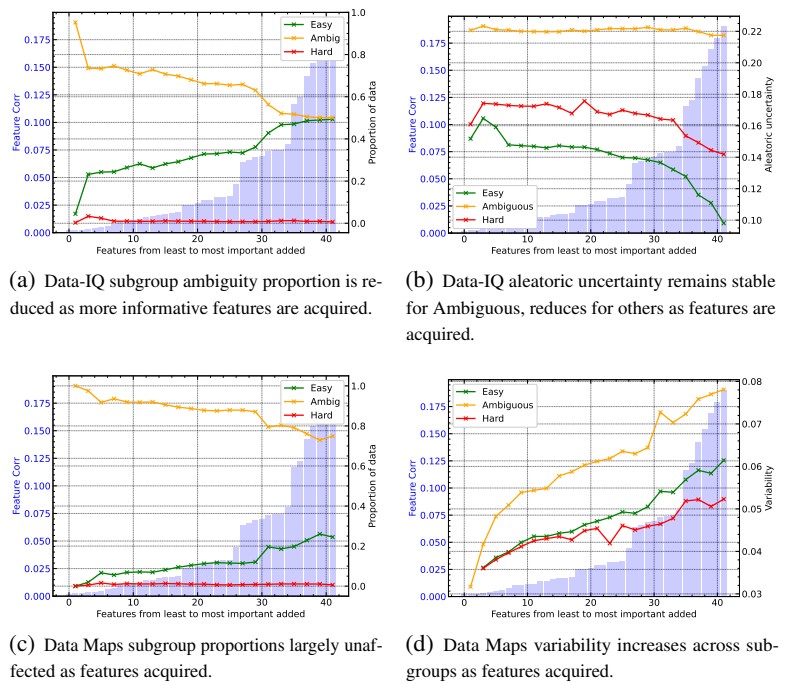

(a) Data-IQ subgroup ambiguity proportion is reduced as more informative features are acquired.

(b) Data-IQ aleatoric uncertainty remains stable for Ambiguous, reduces for others as features are acquired.

(c) Data Maps subgroup proportions largely unaffected as features acquired.

(d) Data Maps variability increases across subgroups as features acquired.

Figure 31: Support Quantifying the value of feature acquisition based on change in ambiguity. Only Data-IQ captures this relationship.

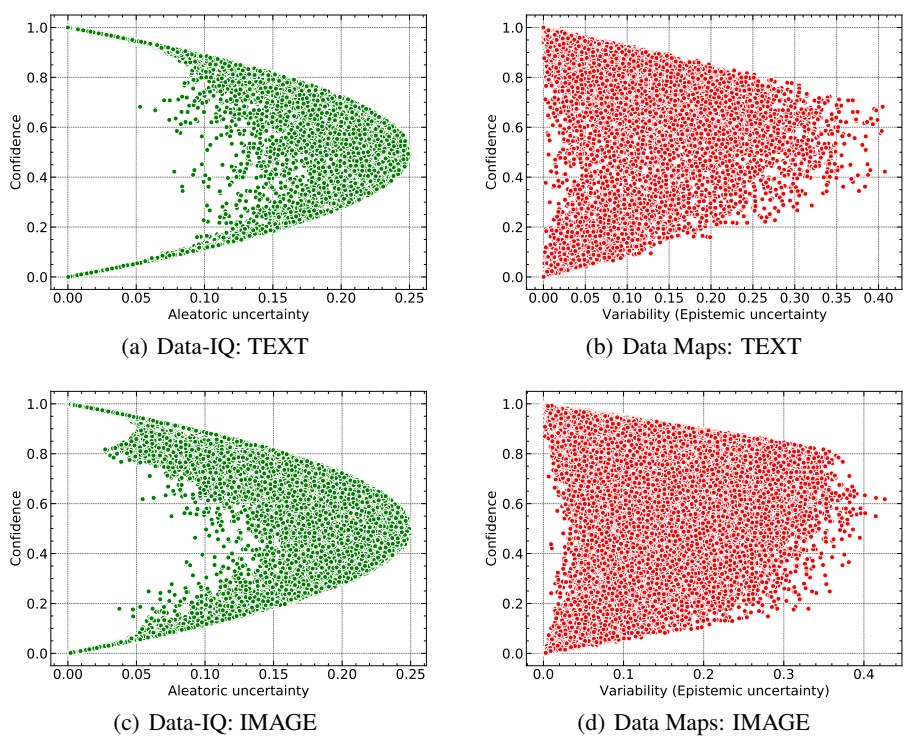

(a) Data-IQ: TEXT

(b) Data Maps: TEXT

(c) Data-IQ: IMAGE

(d) Data Maps: IMAGE

Figure 32: Data-IQ vs Data Maps compared on text and image modalities

## C.7 Data-IQ on text data (NLP)

**Goal.** We wish to assess the generality of Data-IQ. Whilst, in the main paper we mainly tackled tabular data, we wish to showcase the potential utility of Data-IQ on text data for NLP tasks.

**Experiment.** We train a bidirectional RNN on the IMBDb 50K dataset, where the task is to predict the sentiment on the basis of the reviews. We then use Data-IQ to analyze the training dynamics and categorize the data into subgroups - *Easy*, *Ambiguous* and *Hard*

**Takeaway.** We notice that the *Easy*, *Ambiguous* and *Hard* groups represent different phenomena in the text data. We summarize these below.

**EASY: Average review length - 163 words**

- Easy to predict review. Very negative sentiment with a negative label

> **NEGATIVE (0)**: This movie is pathetic in every way possible. Bad acting, horrible script (was there one?), terrible editing, lousy cinematography, cheap humor. Just plain horrible. I had seen 'The Wishmaster' a couple weeks before this movie and I thought it was a dead-ringer for worst movie of the year. Then, I saw 'The Pest' and suddenly 'The Wishmaster' didn't seem so bad at all. Bad Bad Bad. Excruciatingly bad.

> **POSITIVE (1)** When I saw this movie for the first time I was both surprised and a little shocked by the blatant vibrance of the story. It is a very artistic drama with incredible special effects, spectacular acting, not to mention a very excellent job in the makeup department. Jennifer Lopez has pulled herself out of past roles that dug into her career with this movie, portraying a very sensitive child psychologist who works with a team of engineers to enter the minds of comatose patients to treat them. Vincent D'onofrio played amazingly well. His portrayal of a sadist serial killer was perfect to a T. The sheer emotion conveyed by his performance is astounding. Vince Vaughn isn't my favorite, but still performed exceptionally well. The symbolism and artistry was intriguing and titillating, sometimes surprising, and other times shocking. Overall, I say this is a wonderful movie, with excellent acting and beautiful artwork

> **NEGATIVE (0)**: I've tried to watch this so-called comedy, but it's very hard to bear. This is a bad, narrow-minded, cliché-ridden movie. Definitively not funny, but very much boring and annoying, indeed. Bad script, bad acting. It's a complete waste of time - and there remains nothing more to say, I'm afraid 1 out of 10 points

**AMBIGUOUS: Average review length - 283 words** - Linguistic ambiguity. A mix of positive and negative comments. Other reviews of AMBIGUOUS even have text which simply describes the story without any positive/negative sentiment. - Ambiguous reviews also are much longer (in words).

**NEGATIVE (0)** I saw this movie on the film festival of Rotterdam (jan '06) and followed the discussion between director and public afterwards. Many people reacted shocked and protesting. He will get a lot of negative critics. But: the world is cruel like this, and it's not funny. People don't like it. That itself doesn't mean that the movie is bad. I can see that difference. Don't shoot the messenger that shows us the world outside our 'hubble'! Nevertheless I think this a bad movie. Film-technically it's a good one. Nice shots and script, most good fitting music, great actors. The director pretends to make a psychological movie, - the psychology however is of poor quality. Describing such a powerful violence itself is not the art. The art would be a powerful description of the psychological process behind that violence. How does a shy boy come to such a cruelty? The director pretends to describe that, - but is not good in that. The director used several times the word the 'selfishness' of people, mentioning for instance the teacher. Only: this teacher wasn't selfish,- just someone in several roles, caring for his pupils, ánd worried about his script. I think it's a simplification to call him selfish. The atmosphere in the village is creepy, and the mother made awful mistakes ('you terribly let me down') but it doesn't become believable for me, that there is caused súch a lot of pain, that the shyest boy comes to such terrible things. In fact, reality is far more complex than the way, this film describes and it needs far better descriptions. The interesting thing would be: how does it work? Describe that process for me please, so that we understand.With the written phrase on the end, the director said to point to an alternative way of life. It was the other extreme, and confirmed for me that director and scriptwriter are bad psychologists, promoting black/white-thinking. The connection between violence in films and in society has been proved. Use such a violence gives the responsibility to use it right. There are enough black/white-thinkers in the world, causing lots of war and misery. I hope, this movie won't be successful.

**NEGATIVE (0)**
When the film started I got the feeling this was going to be something special. The acting and camera work were undoubtedly good. I also liked the characters and could have grown to empathise with them. The film had a good atmosphere and there was a hint of fantasy. However, as the film went on, the plot never appeared to takeoff and just rolled on scene by scene. I was unable to understand the connection between the stories. All I could see was the characters occasionally bumping into each other and references to ships in bottles. Without that connection, I was just left with a few unremarkable short stories.

**POSITIVE (1)** Artistically speaking, this is a beautiful movie–the cinematography, music and costumes are gorgeous. In fact, this movie is prettier than those directed by Akira Kurasawa himself. In this case, he only wrote the movie as it was made several years after his death.So, as far as the writing goes, the dialog was well-written and the story, at times, was interesting. However, the story was also rather depressing yet uninvolving in some ways–after all, it's the story of a group of women who work in a brothel. It's interesting that although prostitution has been seen as a much more acceptable business in Japan, the women STILL long for a better life. This reminds me a lot of the movie Streets Of Shame, though Streets Of Shame's characters are a lot less likable and more one-dimensional.So, overall it gets a 7–mostly due to everything BUT the writing. It's too bad that the weakest link in this movie is the story by the great Kurasawa.

**HARD: Average review length - 124 words** - Hard to predict review. Very positive sentiment with a negative label. Might reflect mislabeling.

**NEGATIVE (0)** this one of the best celebrity's reality shows a ever saw. we can see the concerts we can see the life of Britney, i love the five episodes. i was always being surprised by Britney and the subjects of the show i think that some people don't watch the show at all we can how a great person she his. she his really funny really gentle and she loves her fans and we can see how she loves her work. i just don't give a 10 because of k-fed he his a real jerk he doesn't seem to like Britney at all. I they make a second season of this great show because it shows at some people how Britney really is. Go Britney your the best and you will never leave our hearts.

**POSITIVE (1)** Okay, first of all I got this movie as a Christmas present so it was FREE! FIRST - This movie was meant to be in stereoscopic 3D. It is for the most part, but whenever the main character is in her car the movie falls flat to 2D! What!!?!?! It's not that hard to film in a car!!! SECOND - The story isn't very good. There are a lot of things wrong with it. THIRD - Why are they showing all of the deaths in the beginning of the film! It made the movie suck whenever some was going to get killed!!! Watch it for a good laugh , but don't waste your time buying it. Just download it or something for cheap

**NEGATIVE (0)** this is a great movie. I love the series on tv and so I loved the movie. One of the best things in the movie is that Helga finally admits her deepest darkest secret to Arnold!!! that was great. i loved it it was pretty funny too. It's a great movie! Doy!

## C.8    Data-IQ on images (computer vision)

**Goal.**    We wish to assess the generality of Data-IQ. Whilst, in the main paper we mainly tackled tabular data, we wish to showcase the potential utility of Data-IQ on image data for computer vision tasks.

**Experiment.**    We train a convolutional neural network on the CelebA dataset, where the task is to predict the gender based on the image of the celebrity. We then use Data-IQ to analyze the training dynamics and categorize the data into subgroups - *Easy*, *Ambiguous* and *Hard*

**Takeaway.**    We notice that the *Easy*, *Ambiguous* and *Hard* groups represent different phenomena in the image data. We summarize the findings in Figure 33

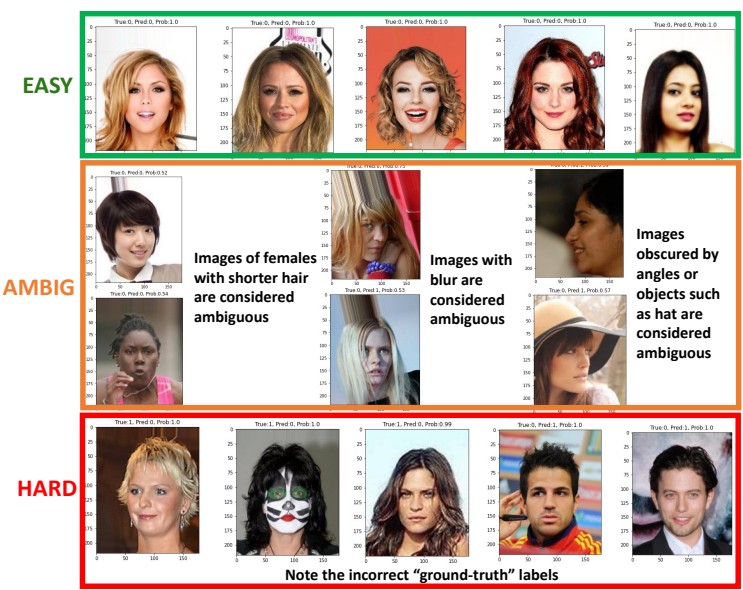

Figure 33: Subgroups highlighted by Data-IQ on the CelebA dataset

## C.9 Effect of averaging over the uncertainty metric

**Goal.** We wish to assess the impact of averaging/taking an expectation over the metric such as aleatoric uncertainty over training. Specifically, we assess the case where an example has high variance at the beginning of training and flattens with time, compared to an example that remains at a medium level throughout training. These represent the EASY/HARD (high variance in the beginning and then flattening) and AMBIGUOUS (medium level throughout) subgroups respectively. The question is, by taking an average will the uncertainties be similar, despite the training dynamics being different? If the average uncertainties are not similar that means we can distinguish the groups with different training dynamics.

**Experiment.** We conduct an experiment to assess this by sampling 100 examples from each of these groups and plot the following three metrics per sample:

- Training dynamic (i.e. the aleatoric uncertainty over training epochs).
- The distribution over aleatoric uncertainty scores across all training steps.
- The "average" aleatoric uncertainty.

**Takeaway.** The result as shown in Figure 34 below shows that even by taking an average, there is still a distinct separation to be able to distinguish between the two types of samples with different training dynamics. This suggests that averaging over the uncertainty is indeed a useful metric to evaluate the uncertainty of an individual data sample.

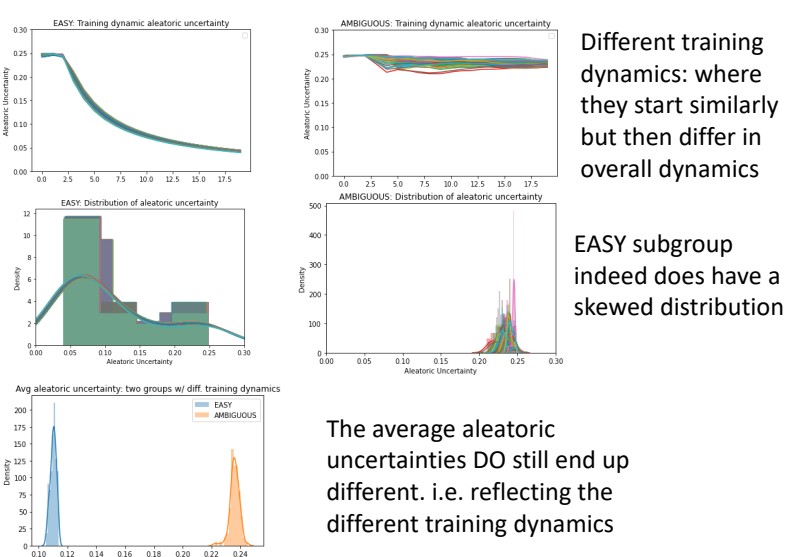

Figure 34: Assessment of whether we can distinguish groups with different training dynamics when we average over the metric

## C.10 Effect of additional sources of randomness

**Goal.** We wish to assess the impact of incorporating additional sources of randomness - for example dropout.

**Experiment.** We have test the model parameterizations from Section 4.1, but additionally trained an additional 3 models with the same architecture, yet with the inclusion of different dropout masks with dropout probability: $p = 0.1, p = 0.2, p = 0.3$.

We then assess the Spearman correlation of the metrics across all model combinations, as was done in the experiment on robustness to variation from Section 4.1. The higher correlation, implies better robustness to variations.

**Takeaway.**  Data-IQ remains most robust to variation, even with this additional randomness. We highlight the results on all four datasets below in Table 11.

Table 11: Comparison of robustness/consistency across different models on the basis of the Spearman rank correlation

| Dataset | (Ours) Data-IQ | Data Maps |
|---|---|---|
| Covid | $0.93 \pm 0.07$ | $0.66 \pm 0.16$ |
| Support | $0.92 \pm 0.02$ | $0.81 \pm 0.11$ |
| Prostate | $0.97 \pm 0.01$ | $0.96 \pm 0.01$ |
| Fetal | $0.86 \pm 0.06$ | $0.81 \pm 0.13$ |

### C.11 Effect of hyper-parameters on the recovery of estimated groups

**Goal.**  We wish to assess how model choices/parameterizations might affect the recovery of the different subgroups. i.e. we desire that if a sample is categorized as EASY with one parameterization, that similarly it should be categorized as EASY for another parameterization. Similarly, for the other subgroups - we desire that their subgroup characterization stays the same across different model parameterizations.

**Experiment.**  We use the same sets of model parameterizations as assessed in Section 4.1 (Robustness to variation experiment). We then compute the overlap of sample ids for each subgroup ($Easy$, $Ambiguous$, $Hard$) between the different parameterizations. This can be thought of as accuracy or sample level stability for assignment to a specific subgroup.

**Takeaway.**  We present the results across all four datasets below in Table 12 which suggest that recovery of subgroups is stable, with more than 90% of samples being assigned to the same subgroup between models.

Table 12: Assessment of stability in group assignment across different parameterizations. The score reflects the average overlap of id characterization

| Dataset | (Ours) Data-IQ |
|---|---|
| Covid | $0.94 \pm 0.03$ |
| Support | $0.91 \pm 0.01$ |
| Prostate | $0.95 \pm 0.01$ |
| Fetal | $0.77 \pm 0.14$ |

### C.12 Correlation of weights through training

**Goal.**  We wish to assess the correlation of weights through model training.

**Experiment.**  To provide this assessment we follow Jin et al [62], where weight correlation is defined as the average cosine similarity between weight vectors of neurons. We conduct experiment to assess the correlation of model weights over epochs $e \in \{1, 2...E\}$. i.e. we assess the average correlation of model weights $(e_1, e_2), (e_2, e_3)$ and so on.

**Takeaway.**  We note that [62], shows that LOW correlation is less than 0.2. Our validation shows that the weights have LOW correlation, with an average weight correlation across the 4 datasets of $0.1 \pm 0.02$. Thus, even though our goal is different from a typical sampling setting (i.e. we just want to measure the dispersion of observed variables) - we still wish to highlight the low correlation of weights between epochs.

However, as a recommendation for practitioners if this is a concern: a practical solution is that the practitioner could decide to simply increase the intervals for which the dynamics are sampled, thereby decreasing this correlation even further.

## C.13   Data sculpting: Absolute number of samples

**Goal.**   When performing data sculpting as in Section 4.3 we desire to know the absolute number of samples removed in order to quantify the importance of the effect of removing such datapoints on model performance.

**Experiment.**   We present this analysis for two scenarios:

- Balanced SEER-CUTRACT: 1000 samples each
- Unbalanced SEER-CUTRACT: 10000 samples in SEER and 1000 in CUTRACT

**Takeaway.**   For the balanced experiment the full Ambiguous subgroup (i.e. proportion=1) has a total of 500 samples. Hence, when we sweep the x-axis representing the proportions of Ambiguous samples; $p = \{0, 0.2, 0.4, 0.6, 0.8, 1\}$, this is reflective of $\{0, 100, 200, 300, 400, 500\}$ absolute samples. This result shows that the number of samples that can be excluded is fairly substantial.

We compare this to the unbalanced experiment. The results are in Figure 35. It shows that a substantial number of Ambiguous samples can be removed to improve performance. However, in the case of (b) when there are a lot of samples, removing too many can also be harmful.

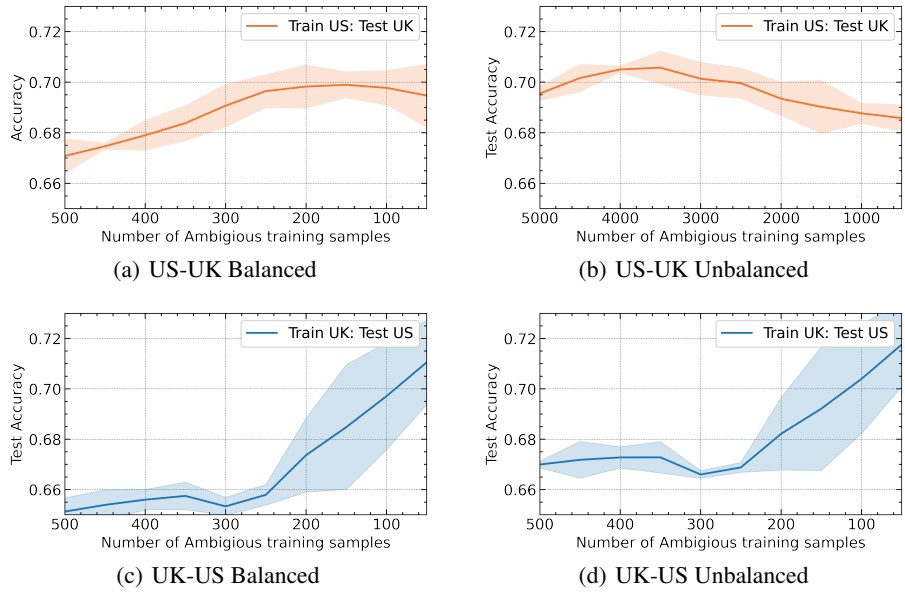

(a) US-UK Balanced            (b) US-UK Unbalanced

(c) UK-US Balanced            (d) UK-US Unbalanced

Figure 35: Data sculpting - absolute number of *Ambiguous* samples removed vs test accuracy