# OpenReview forum: "Data-IQ: Characterizing subgroups with heterogeneous outcomes in tabular data"
_NeurIPS.cc/2022/Conference — NeurIPS 2022 Accept_

### Official Review · Reviewer_2wVN · 2022-07-10

**Rating:** 3
**Confidence:** 3
**Soundness:** 2 fair
**Presentation:** 2 fair
**Contribution:** 2 fair

**Summary:**

The authors proposed a framework of data characterisation and data sculpting during model training based on the outcome and feature characteristics of the subgroups within a dataset. The authors separate the subgroups into the Easy, Ambiguous and Hard groups. Via a set a of experiments on different datasets, they showed that the the framework was able to identify the uncertainties due to the data characteristics, insights regarding the subgroup feature characteristics, and can potentially improve model generalisability by excluding Ambiguous group from training.

**Questions:**

1. To my understanding, the "Hard" group are sample with similar feature as the "Easy" group, but opposite outcomes. How is this different from the "Ambiguous" group, which includes samples that have similar features but different outcomes?
2. Regarding the data insights, are the findings clinical meaning full? Are their any evidences (related biomedical researches), can be referenced?
3. The improved in Table 2 is marginal. Also the comparison involves two factors: Data-IQ and modelling method Group-DOR. Could the authors show a baseline model + Data-IQ?

**Limitations:**

The authors discussed the limitations of the method in the section 5. However, the discussion was limited, maybe due to the length of the paper. I personally would be interested to understand how the framework may influence the inclusivity of the dataset used for model training.

**Strengths And Weaknesses:**

The authors provides a different perspective, which puts the characteristics of data in the centre, to analyse the factors that influences the performance of a model. They provide a tool, Data-IQ, to help the researcher understand data better before diving into modelling process. To me, it is in innovative and should be encouraged. However, what the authors propose might not be sufficiently mature and can be used as framework for assessing datasets and guide model training process by the community. Regarding the writing, the paper was trying to put too much ideas into the limited length, including variation vs robustness, subgroup characterisation, model generalisability, etc.. It would be much clearer if there could focus on one single point, e.g. data sculpting based on the identified subgroups.

---

> ### Author Response · Authors · 2022-08-02
> **Response to Reviewer 2wVN [Part 1/6]**
>
> Thank you for your thoughtful comments and suggestions.
>
> We give answers to each of the following in turn, as well as highlighting corresponding updates to the revised manuscript at the end of every point. We have uploaded the revised manuscript which incorporates the changes and suggestions:
>
> -----
>
> (A) Maturity of Data-IQ **[Part 2/6]**
>
> (B) Too many contributions of Data-IQ **[Part 3/6]**
>
> (C) Clinical meaningfulness of data insights **[Part 4/6]**
>
> **Clarifications**
>
> (D) Clarification of sub-group definitions **[Part 5/6]**
>
> (E) Clarification of Group-DRO experiment **[Part 5/6]**
>
> (F) Usage of Baseline model + Data-IQ **[Part 5/6]**
>
> (G) Influence of Data-IQ on data inclusivity **[Part 6/6]**
>
> -----
>
> ## Preface:
> We wish to highlight that the problem solved by Data-IQ is to assess and characterize the data itself into sub-groups. This motivates our usage of aleatoric uncertainty, as we seek to model the uncertainty inherent to the data. Identifying these sub-groups is a practically valuable problem, as improving accuracy and robustness often depends on the data's characteristics and quality [R1,R2,R3].  That said, the problem tacked by Data-IQ around data quality as mentioned in [R3,R4]  “is often undervalued as merely operational, yet failing to account for it can have immense practical harm” [R3].
>
> Consequently, our goal in Data-IQ is to build a systematic framework with four desired properties, motivated by the considerations of practitioners at various stages of the ML pipeline. In satisfying these properties, Data-IQ addresses the “dire need for an ML-aware data quality that is not only principled but also practical for a larger collection of ML models” [R2].
>
> `Response continues in part 2`

---

> > ### Author Response · Authors · 2022-08-02
> > **Response to Reviewer 2wVN [Part 2/6]**
> >
> >
> > `This is part 2 of the authors' response`
> >
> > -----
> >
> > # (A) Maturity of Data-IQ
> >
> >
> > ------
> > **TL;DR:**  We highlight Data-IQs maturity along 5 key dimensions.
> >
> > 1. Data-IQ has extensive experimental validation
> > 2. Data-IQ is principled in metrics & properties
> > 3. Data-IQ is flexible & can be used with many models
> > 4. Data-IQ is general purpose & can be extended to other modalities
> > 5. Data-IQ is a usable tool for the community
> >
> > ------
> > The maturity of a method can be assessed across a variety of dimensions. We highlight Data-IQ’s maturity across five different dimensions:
> >
> > 1. **Data-IQ has extensive experimental validation**: Data-IQ has extensive experiments on 4 real-world datasets, across 4 ML model types/classes, and across 3 modalities. Additionally, Data-IQ addresses a variety of use-cases across the ML pipeline, from principled data collection to reliable model deployment. We believe that the benefit of Data-IQ as showcased in multiple settings signals the method's maturity and highlights the applicability of Data-IQ to the community to tackle the important, yet overlooked problem of data quality and assessment.
> >
> >
> > 2. **Data-IQ is principled in metrics & properties**: Besides experimental validation, we have also shown in the paper, that Data-IQ and the metrics we use are more principled when compared to other approaches (**Section 3.3 and Appendix A**). This motivates the **conceptual shift** to model the inherent data uncertainty (aleatoric), instead of model uncertainty (epistemic) as is normally done.  **Section 3.3 & Figure 2** also highlights why the choice of principled metric is important. Furthermore, we have introduced 4 principled properties (**Section 1, L57-L64**) i.e. (P1) Robust data characterization, (P2) Principled data collection, (P3) Reliable model deployment and (P4) Plug & play. These properties are motivated by considerations across the ML pipeline and address the requirements needed for “ML-aware data quality that is principled and practical for a variety of ML models”. Formalizing these considerations adds to the rigor of Data-IQ and could be used as properties by future methods.
> >
> > 3. **Data-IQ is general & can be used with many models**: Unlike prior methods, which typically can only be used with neural networks (see A.1 for this discussion); Data-IQ can be used with *any* iteratively trained ML model, this is due to our formulation. We initially evaluated our method with neural networks vs XGBoost (**Appendix C.2**); but to demonstrate the wide applicability of Data-IQ, we have added LightGBM and CatBoost to the revision of Appendix C.2. This flexibility highlights Data-IQs maturity, as it allows practitioners who might not use neural networks to still engage with our method to assess their data.
> >
> > 4. **Data-IQ is general purpose & can be extended to other modalities**: We have thoroughly assessed the usefulness of Data-IQ on tabular data, which in itself is an often overlooked modality. However, as part of the revised paper, we have also extended this evaluation and show that Data-IQ is generally applicable beyond tabular data. We showcase Data-IQ’s generality on both image (computer vision tasks) and text (NLP tasks). We refer the reviewer to the revised paper **Appendix C.7 and C.8**. For a snapshot of the results see: **TEXT** [https://imgur.com/a/NoK9I0t], **IMAGES** [https://imgur.com/a/TcDq9eg]. The use of Data-IQ across three different modalities (tabular, image and text) is illustrative of Data-IQs maturity.
> >
> > 5. **Data-IQ is a usable tool for the community**: As discussed in **Appendix B.1.1**: Data-IQ has been implemented in a versatile object-oriented way and is mature from an integration perspective as it can plug into any framework including Pytorch, Tensorflow/Keras, Skorch and Scikit-learn style APIs. Besides being plug and play with different models, Data-IQ is equally “plug and play” from a framework usability and integration perspective. Note that for libraries including Pytorch, Tensorflow/Keras, Skorch, we can easily include Data-IQ simply by adding it to the training callbacks of the respective training loops. We will make the code publically available upon acceptance such that the Data-IQ tool can be used by the community.
> >
> > -----
> > `Response continues in part 3`

---

> > > ### Author Response · Authors · 2022-08-02
> > > **Response to Reviewer 2wVN [Part 3/6]**
> > >
> > >
> > > `This is part 3 of the authors' response`
> > >
> > > ------
> > > # (B) Too many contributions of Data-IQ
> > >
> > >
> > > ------
> > > **TL;DR:**  The extensive experiments covering different areas help to validate the 4 properties of useful sub-group discovery.
> > >
> > > ------
> > > We clarify that despite the numerous contributions, they all serve to address the same key problem. In Data-IQ, we tackle one key problem:  how do we characterize examples into sub-groups based on the data's inherent qualities (IQ); at both training and deployment time.  This permits Data-IQ to provide an ML-aware data quality that is principled and practical for a variety of ML models. For further discussion, see **Section 1, L50-69**.
> > >
> > > In satisfying this requirement, we have formalized 4 desired properties (or 4 paradigms, as noted by R-3GB7), motivated by the considerations of practitioners at various stages of the ML  pipeline. Thus, while we have 4 properties, all are necessary components that together serve to address the key problem of data quality assessment and subgroup characterization.
> > >
> > > We now cover each property (P1-P4) and clarify how each experimental contribution evaluates the property.
> > >
> > >
> > > *(P1) Robust data characterization* [**Section 4.1**].
> > > * Goal: the characterization of data examples should be robust, such that it is consistent across similar performing models, that have different architectures/parametrizations.
> > > * Experiment outcome: We observe that Data-IQ is the most consistent and robust to variation across different models compared to baselines. Hence, showing the Data-IQ provides the most stable characterization of the data.
> > >
> > > *(P2) Principled data collection* [**Section 4.2**]
> > > * Goal: the characterization should be informative and actionable, providing practitioners insights that enable both quantitative feature collection and selection between datasets.
> > > * Experimental outcome: (1)Principled feature acquisition: we show that Data-IQ’s characterization provides a principled approach using the aleatoric uncertainty to assess the benefit of acquiring a specific feature, (2) Principled dataset comparison: we show that Data-IQ is a useful tool in the hands of practitioners wishing to assess data quality, especially when selecting between different datasets.
> > >
> > > *(P3) Reliable model deployment*  [**Section 4.3**].
> > > * Goal: the characterization should enable reliable model usage, both by unmasking unreliable sub-groups or using the sub-groups to tailor the data for better performance.
> > > * Experiment outcome: (1) Data sculpting: we show the value of sculpting the training dataset, on the basis of the Data-IQ sub-groups, as a way to improve the reliability and performance of a deployed model, (2) Group-DRO: we show that Group-DRO in tabular settings is not the silver bullet to ensuring reliable model performance on sub-groups (especially Ambiguous sub-groups) and (3) sub-group-informed usage of uncertainty estimation: we show that typical usage of predictive uncertainty estimates to defer samples at deployment time, might not be reliable on the Ambiguous sub-group.
> > >
> > > (P4) Plug & Play [**Section 3.4 & Appendix C.2**].
> > > * Goal:  the characterization should be applicable to a variety of ML models widely used on tabular data, including neural networks, gradient boosting (and variants) etc.
> > > * Experimental outcome: we show that Data-IQ can be used flexibly used with neural networks, XGBoost, LightGBM and CatBoost, and at the same time provide consistent data characterization.
> > >
> > > Thus, we respectfully highlight that each property P1-P4 when working together is a necessity to address the “dire need for an ML-aware data quality that is not only principled but also practical for a larger collection of ML models”.
> > >
> > > We ask the reviewer to consider that P1-P4 are desired properties in addressing the key problem of data quality assessment and characterization. The breadth and comprehensive experimental validation of Data-IQ on a wide variety of settings and datasets then serve to demonstrate how these properties are satisfied by Data-IQ.
> > >
> > > ------
> > > `Response continues in part 4`

---

> > > > ### Author Response · Authors · 2022-08-02
> > > > **Response to Reviewer 2wVN [Part 4/6]**
> > > >
> > > > `This is part 4 of the authors' response`
> > > >
> > > > ------
> > > > # (C) Clinical meaningfulness of data insights
> > > >
> > > > ------
> > > > **TL;DR:**  We highlight the clinical utility of the data insights on the basis of discussions with two clinicians and how the data insights match other Biomedical literature.
> > > >
> > > > ------
> > > > We address the clinical meaningfulness of the data insights in a two-fold manner:
> > > > (1) References from the biomedical literature supporting the data insights and (2) Discussions with two clinicians about the clinical utility of the data insights. We reference the prostate cancer data insights experiment for brevity.
> > > >
> > > > 1. **References from the biomedical literature supporting the data insights**
> > > >
> > > > The data insights plot in Figure 5 highlights insights about prostate cancer patients. Below, we provide examples where our data insights match the findings from other biomedical studies.
> > > >
> > > > * **Intuitive sub-groups identified**: As shown by Albertsen [R2] men with higher stage/severity disease were considerably more likely to die of prostate cancer and vice versa for lower severity. Further, increased age also significantly affected the probability of dying from prostate cancer. While this is unsurprising, it matches the EASY sub-group of examples identified by Data-IQ where patients which a higher cancer stage, higher PSA (severity score) and higher age had a higher probability of mortality and vice versa for lower severity.
> > > >
> > > > * **Heterogeneity identified**: As discussed by Gartlehner et al [R4] clinical populations can have heterogeneous or variable outcomes beyond what would be expected. The HARD sub-group represents this as 1-2% of the actual data and are those patients which are opposite of the expected intuitive outcome. i.e. sick patients living and healthy patients dying.
> > > >
> > > > * **Link to heterogeneity matches data insights**: As shown by Hall et al [R6,] specific types of metastatic prostate cancers are sensitive to hormone therapy in terms of outcome. Their findings match our HARD examples with stage 4 cancer and treated with hormone therapy BUT without mortality.
> > > >
> > > > * **Therapies**: Lu-Yao et al [R3] studied which types of prostate cancer patients would be the best candidates for hormonal therapy. They found that hormonal therapy is best suited for poorly differentiated cancers. i.e. the grade is higher. We can see in our identified sub-groups that indeed those that did receive hormonal therapy had higher grades than those that didn't. Which links the data insights to this real-world finding.
> > > >
> > > > 2. **Discussions with two clinicians about the clinical utility of the data insights**
> > > >
> > > > We have also **engaged with two clinicians** to assess the clinical utility of the data insights. From a clinical perspective, they highlighted that most cancer outcomes can be linked to existing and well-known clinical factors. For example, high PSA (i.e. severity) is linked to greater mortality. That being said, there are still patients that behave in different ways. This is noted as the current SOTA clinical methods Prostate PREDICT and MSKCC achieve accuracies of around +-0.8 [R6]. This means there are sub-groups of underperformance.
> > > >
> > > > The clinicians both noted that it would be valuable to have a method to identify cohorts or subsets of patients for clinicians to focus on. Specifically, highlighting cohorts for which collecting extra factors may be useful - such that the limited clinical resources are used in a targeted and efficient manner. This type of identification is further backed up by Pensen & Albertsen [R1], who highlight the value of identifying types of patients to clinicians to guide management and clinical decision-making.
> > > >
> > > > This is exactly what Data-IQ provides as a tool wherein clinicians can target their focus to the AMBIGUOUS sub-group, whilst knowing that the EASY sub-group has high performance. Hence, the data insights experiment aims to illustrate how these three sub-groups can be highlighted and surfaced to clinicians using the radial plots such as Figure 5.
> > > >
> > > > This mirrors frameworks such as CONSORT-AI [R7] which define reporting guidelines for AI-based medical applications. Methods such as Data-IQ could be used as a tool to answer the reporting guideline questions about how the quality of the input data was assessed.
> > > >
> > > > ------
> > > > `Response continues in part 5`

---

> > > > > ### Author Response · Authors · 2022-08-02
> > > > > **Response to Reviewer 2wVN [Part 5/6]**
> > > > >
> > > > >
> > > > > `This is part 5 of the authors' response`
> > > > >
> > > > > ------
> > > > >
> > > > > # Clarifications
> > > > >
> > > > >
> > > > > ------
> > > > >
> > > > > # (C) Clarification on sub-group definitions
> > > > >
> > > > >
> > > > > ------
> > > > >
> > > > > We wish to clarify the definitions of the different sub-groups. We highlight the definitions of the three sub-groups Easy, Ambiguous and Hard in **Section 1, L40-49** and the analysis of differences between sub-groups in representation space in **Appendix C.1**.
> > > > >
> > > > > To further clarify, let us consider the task of patient mortality prediction, which we have studied in the paper. It is intuitive that sicker patients more often have a mortality event, and the severity of the disease will be reflected in the measured features or covariates of the patient.
> > > > >
> > > > > **EASY**: This relationship where sicker patients more typically have morality events is, thus, easy to learn for any model and will be predicted correctly with high confidence.
> > > > >
> > > > > **HARD**: In clinical cases, sometimes patients have heterogeneous outcomes, i.e. survival despite their poor prognosis [R4]. This heterogeneity could result from randomness, making it practically impossible for a model to learn. These examples will be predicted incorrectly yet with high confidence (or equivalently have low confidence for the correct class).
> > > > >
> > > > > **AMBIGUOUS**: In tabular data, there are also examples with inherent ambiguity where the predicted probability for the correct class remains low. They appear where the current features are insufficient to distinguish the example correctly, regardless of the model used [14, 15]. Thus, while they might have similar features, these patients have different outcomes. The real issue with Ambiguous is that the current features are insufficient to distinguish the example correctly.
> > > > > We experimentally highlight this difference in **Section 4.2: feature acquisition** and **Appendix C.1: where we analyze the difference in representation space**.
> > > > >
> > > > >
> > > > >
> > > > >
> > > > >
> > > > > ------
> > > > >
> > > > > # (E) Clarification of Group-DRO experiment
> > > > >
> > > > > ------
> > > > > **TL;DR:**  We clarify the purpose of the Group-DRO experiment is to illustrate that Group-DRO might not be a silver bullet in all settings to improve sub-group performance; which is challenges commonly held notions. Furthermore, we analyze why this is the case in the tabular setting compared to computer vision and provide recommendations on how to improve performance by collecting better features.
> > > > >
> > > > > ------
> > > > > We wish to clarify the purpose of the Group-DRO experiment, especially around the marginal improvement using Data-IQ. We agree with this point and acknowledge this in **Section 4.3, L382-387**: "Nevertheless, while using Data-IQ can boost performance, it is evident that simply applying Group-DRO is not a silver bullet to equalize sub-group performance, given the sometimes small improvement."
> > > > >
> > > > > For clarification, the experiment assesses whether Group-DRO is always the silver bullet to improve sub-group performance. The reason is that the literature in other settings (e.g. computer vision), typically assumes methods such as Group-DRO will improve performance and robustness when applied. The results of our experiment on tabular data challenge this common notion.
> > > > >
> > > > > We have provided a rationale for this difference (**Section 4.3, L387-390**): “tabular setting is different, from the spurious correlation setting in computer vision where Group-DRO typically shines”. And have provided a suggestion to practitioners (**L390-393**) that “based on our feature acquisition results (Sec 4.2), that in tabular settings, practitioners would be better served to acquire better features to improve performance and reduce ambiguity.”
> > > > >
> > > > > Hence, while performance gains might be marginal, the experiment's goal is simply to highlight that methods such as Group-DRO are not a silver bullet to improve deployed model performance and should not be blindly applied in all settings. Rather, alternative approaches such as feature acquisition should be explored.
> > > > >
> > > > >
> > > > > ------
> > > > >
> > > > > # (F) Baseline model + Data-IQ
> > > > >
> > > > > ------
> > > > > **TL;DR:**
> > > > > The experiment in **Section 4.3: Less is more: data sculpting based on sub-groups**, showcases the usage of how the sub-groups identified by Data-IQ can be used to improve the performance of a baseline model.
> > > > >
> > > > > ------
> > > > > We wish to clarify the usage of Data-IQ with a baseline model and highlight the experiment in **Section 4.3: Less is more: data sculpting based on sub-groups** which address this consideration. We train a baseline predictive model on cancer patient data. Specifically, we train a model on US data (SEER) and assess performance on patients in the UK (CUTRACT) and vice versa train another model for the opposite direction. In both cases, we use Data-IQ to characterize data into the sub-groups - Easy, Ambiguous and Hard. Thereafter, **Figure 7** shows that by sculpting the dataset using the sub-groups identified by Data-IQ (specifically Ambiguous samples); that the performance of the baseline model can be improved.
> > > > >
> > > > > ------
> > > > > `Response continues in part 6`

---

> > > > > > ### Author Response · Authors · 2022-08-02
> > > > > > **Response to Reviewer 2wVN [Part 6/6]**
> > > > > >
> > > > > > `This is part 6 of the authors' response`
> > > > > >
> > > > > > ------
> > > > > > # (G) Influence of Data-IQ on data inclusivity
> > > > > >
> > > > > >
> > > > > > ------
> > > > > > **TL;DR:** Data-IQ enables the assessment of dataset quality at a sample level. Coupled with the practical remedies proposed in the paper (to reduce ambiguity) - Data-IQ could also assist data owners to better audit and understand data limitations, as well as, quantify if extra features may help to improve the dataset. Thus, Data-IQ could be leveraged as an assessment tool when trying to improve inclusivity on underperforming sub-groups.
> > > > > >
> > > > > > ------
> > > > > >
> > > > > > We fully agree that understanding the inclusivity of samples in a dataset is important. Especially, when thinking about fairness and reliability. Data-IQ can be used to highlight sub-groups of under-performance in a dataset. Hence, we believe that this could be used by data owners to better understand and audit their datasets on the basis of the identified sub-groups.
> > > > > >
> > > > > > Furthermore, in the case of Ambiguous samples, we have provided analysis to show: (1) how we can improve performance on these samples (by obtaining better features) and (2) how Data-IQ could also quantify these changes with respect to reduction in sample ambiguity. The benefit is that for example if these samples correspond to a specific group, it means that Data-IQ could be used to pinpoint samples for targeted improvement.
> > > > > >
> > > > > > We believe that Data-IQ could be used as a tool to help data owners to understand their data when trying to improve the inclusivity of their datasets.
> > > > > >
> > > > > > **UPDATES:** We will include this discussion at the end of the paper as part of the **extra camera-ready page**.
> > > > > >
> > > > > > ------
> > > > > > `End of authors' response`
> > > > > >
> > > > > > ------
> > > > > > # REFERENCES
> > > > > >
> > > > > > ------
> > > > > > [R1] Penson, David F., and Peter C. Albertsen. "Lessons learnt about early prostate cancer from large scale databases: population-based pearls of wisdom." Surgical oncology 11, no. 1-2 (2002): 3-11.
> > > > > >
> > > > > >
> > > > > > [R2] Albertsen, Peter C., James A. Hanley, Donald F. Gleason, and Michael J. Barry. "Competing risk analysis of men aged 55 to 74 years at diagnosis managed conservatively for clinically localized prostate cancer." Jama 280, no. 11 (1998): 975-980.
> > > > > >
> > > > > >
> > > > > > [R3] Lu-Yao, Grace, Dirk F. Moore, John U. Oleynick, Robert S. DiPaola, and Siu-Long Yao. "Population based study of hormonal therapy and survival in men with metastatic prostate cancer." The Journal of urology 177, no. 2 (2007): 535-539.
> > > > > >
> > > > > > [R4] Gerald Gartlehner, Suzanne L West, Alyssa J Mansfield, Charles Poole, Elizabeth Tant, Linda J Lux, and Kathleen N Lohr. Clinical heterogeneity in systematic reviews and health technology assessments: synthesis of guidance documents and the literature. International Journal of Technology Assessment in Health Care, 28(1):36–43, 2012.
> > > > > >
> > > > > > [R5] Hall, Mary E., Heather L. Huelster, Amy N. Luckenbaugh, Aaron A. Laviana, Kirk A. Keegan, Zachary Klaassen, Kelvin A. Moses, and Christopher JD Wallis. "Metastatic hormone-sensitive prostate cancer: current perspective on the evolving therapeutic landscape." OncoTargets and therapy 13 (2020): 3571.
> > > > > >
> > > > > > [R6] Lee, Changhee, Alexander Light, Ahmed Alaa, David Thurtle, Mihaela van der Schaar, and Vincent J. Gnanapragasam. "Application of a novel machine learning framework for predicting non-metastatic prostate cancer-specific mortality in men using the Surveillance, Epidemiology, and End Results (SEER) database." The Lancet Digital Health 3, no. 3 (2021): e158-e165.
> > > > > >
> > > > > > [R7] Liu, Xiaoxuan, Samantha Cruz Rivera, David Moher, Melanie J. Calvert, and Alastair K. Denniston. "Reporting guidelines for clinical trial reports for interventions involving artificial intelligence: the CONSORT-AI extension." bmj 370 (2020).

---

> ### Author Response · Authors · 2022-08-05
> **Dear Reviewer 2wVN**
>
> Dear Reviewer 2wVN
>
> We are sincerely grateful for your time and energy in the review process.
>
> We hope that our responses and appendix/manuscript updates have been helpful. Please let us know of any leftover concerns and if there was anything else we could do to address any further questions or comments :)
>
> Thank you!
>
> Paper6611 Authors

---

> > ### Author Response · Authors · 2022-08-09
> > **Author follow-up**
> >
> > Dear Reviewer 2wVN
> >
> > Thank you again for your time and expertise during the review process!
> >
> > If there were any leftover concerns, we would sincerely appreciate the opportunity to clarify them---before the discussion period for authors ends. We believe our responses have addressed in detail the full set of questions you had raised, along with corresponding appendix/manuscript updates.
> >
> > We would appreciate if the reviewer kindly let us know if there were any further questions in the very limited time remaining. We are eager to do our utmost to address them!
> >
> > Thank you,
> >
> > Paper6611 Authors

---

### Official Review · Reviewer_WioB · 2022-07-10

**Rating:** 7
**Confidence:** 3
**Soundness:** 3 good
**Presentation:** 4 excellent
**Contribution:** 3 good

**Summary:**

The paper proposes a framework to identify examples in the datasets that a classifier tends to distinguish correctly, wrongly, and randomly.  The paper decomposes the accuracy of a given example into a summation of epistemic uncertainty and aleatoric uncertainty. The paper then uses aleatoric uncertainty to classify examples into the three aforementioned categories, as opposed to existing work that uses epistemic uncertainty as the criterion.

**Questions:**

* In equation (2), is the estimate based on the empirical process a good estimate for the two uncertainty quantities given that the samples in the empirical process might be correlated with each other rather than iid?

**Limitations:**

The paper discusses the limitation of the proposed method. Finding the attributes that are responsible for the hardness of the data points seems to be particularly interesting and relevant for the interpretability of machine learning algorithms.

**Strengths And Weaknesses:**

Strengths:
* the paper deals with a significant problem towards a better understanding of how difficult a data point is to classifiers in terms of classify it accurately.
* The proposed approach that decomposes the accuracy of a given example into a summation of epistemic uncertainty and aleatoric uncertainty is clean and informative of the difficulty of classifying a given example.
* The presentation of the paper is clear. Experiments are illustrative.

Weakness:
* Given the generality of the proposed method, the paper may consider evaluating the proposed method beyond tabular data.

---

> ### Author Response · Authors · 2022-08-02
> **Response to Reviewer WioB [Part 1/3]**
>
> Thank you for your thoughtful comments and suggestions!
>
> We give answers to each of the following in turn, as well as highlighting corresponding updates to the revised manuscript at the end of every point. We have uploaded the revised manuscript, which incorporates the changes and suggestions:
>
> (A) Usage of Data-IQ with other modalities beyond tabular data **[Part 2/3]**
>
> (B) Sampling per epoch & empirical distribution **[Part 3/3]**
>
> `Response continues in part 2`

---

> > ### Author Response · Authors · 2022-08-02
> > **Response to Reviewer WioB [Part 2/3]**
> >
> >
> > `This is part 2 of the authors' response`
> >
> > -----
> >
> > # (A) Usage of Data-IQ with other modalities beyond tabular data
> >
> > ------
> > **TL;DR:** To showcase the generality of Data-IQ we provide an assessment on two additional modalities: TEXT and IMAGES with the analysis included in the revised manuscript, **Appendix C.7 and C.8** respectively.
> >
> > ------
> >
> > Thank you for the suggestion! To highlight the general usefulness of Data-IQ, beyond the tabular data, we have included two additional experiments on two new modalities: Image (computer vision tasks) and Text (NLP tasks) data.
> >
> > **Image data**:
> >
> > We have conducted an experiment on the **CelebA dataset**, where the task is to predict the celebrity’s gender based on their image using a CNN. We have included the results in the revised supplementary in **Appendix C.8**
> >
> > We provide a snapshot of our results here: https://imgur.com/a/TcDq9eg
> >
> > *Summary of sub-groups*:
> > * EASY: images of the celebrities which are easy to classify into the correct gender.
> > * AMBIGUOUS: captures specific traits which are Ambiguous. This could reflect issues such as biases/spurious factors. Example type of “Ambiguity” or issues identified with Data-IQ that might harm model performance:  females with short hair, blur in the images, obscured images due to angle of the image or objects such as hats.
> > * HARD: images that were hard to classify as they had the incorrect label for the gender. I.e. mislabelling of the data.
> >
> > **Text data**:
> >
> > We have conducted an experiment on the **IMDb 50k dataset**, where the task is to predict the sentiment of a text review of a movie using a bidirectional RNN. We have included the results in the revised supplementary in **Appendix C.7**
> >
> > We provide a snapshot of our results here: https://imgur.com/a/NoK9I0t
> >
> > *Summary of sub-groups*:
> > * EASY: reviews with a large amount of the correct sentiment, making it easy to classify
> > * AMBIGUOUS: longer reviews in length. Contain both positive and negative comments, making it linguistically ambiguous. In addition, the reviews have many sentences and comments that simply describe the plot of the movie/series without any sentiment. i.e. much neutral text
> > * HARD: short reviews with very positive/negative comments – yet with the opposite label, which reflects mislabelling of the data.
> >
> > Both results show the practical usefulness of Data-IQ on two further data modalities.
> >
> > **UPDATES**:
> > We have included the results for the new modalities in the revised manuscript. The TEXT experiment is in **Appendix C.7** and the IMAGE experiment in **Appendix C.8**.
> >
> > ----------
> >
> > `Response continues in part 3`

---

> > > ### Author Response · Authors · 2022-08-02
> > > **Response to Reviewer WioB [Part 3/3]**
> > >
> > >
> > > `This is part 3 of the authors' response`
> > >
> > > -----
> > > # (B) Sampling per epoch & empirical distribution
> > >
> > > -----
> > > Our method Data-IQ is similar in spirit to Data Maps [R1] in that the goal is to summarize the variability of the sample dynamics over the course of training (across epochs).
> > > Consequently, the empirical distribution is simply the observed parameters during training. Hence, we clarify that we are not trying to sample the parameters as would be done in a Bayesian setting. Thus, we highlight that our goal is different, as we simply want to measure the dispersion of observed variables.
> > >
> > > That said, we have also conducted an experiment to assess the correlation of model weights over epochs $e \in \{1,2...E\}$. i.e. we assess the average correlation of model weights $(e_1, e_2), (e_2, e_3)$ and so on. We follow Jin et al [R2], where weight correlation is defined as the average cosine similarity between weight vectors of neurons.
> > >
> > > The paper [R2] shows that LOW correlation is less than 0.2. Our validation shows that the weights have **low** correlation, with an average weight correlation for models across the 4 datasets of **0.1 +- 0.02**. Thus, even though our goal is different from a typical sampling setting (i.e. we just want to measure the dispersion of observed variables) - we still wish to highlight the low correlation of weights between epochs.
> > >
> > > That being said, if this is still a concern, a practical solution is that the practitioner could decide to simply increase the intervals between which the training dynamics are captured, thereby decreasing this correlation even further.
> > >
> > > **UPDATES**: We have included the discussion and experiment in the revised manuscript, **Appendix C.12**.
> > >
> > > ------
> > > `End of author response`
> > >
> > > -----
> > > # REFERENCES
> > >
> > > -----
> > > [R1] Swabha Swayamdipta, Roy Schwartz, Nicholas Lourie, Yizhong Wang, Hannaneh Hajishirzi, Noah A Smith, and Yejin Choi. Dataset cartography: Mapping and diagnosing datasets with training dynamics. In Proceedings of the 2020 Conference on Empirical Methods in Natural Language Processing (EMNLP), pages 9275–9293, 2020.
> > >
> > > [R2] Jin, Gaojie, Xinping Yi, Liang Zhang, Lijun Zhang, Sven Schewe, and Xiaowei Huang. "How does weight correlation affect generalisation ability of deep neural networks?." Advances in Neural Information Processing Systems 33 (2020): 21346-21356.

---

> ### Author Response · Authors · 2022-08-05
> **Dear Reviewer WioB**
>
> Dear Reviewer WioB
>
> We are sincerely grateful for your time and energy in the review process.
>
> We hope that our responses and appendix/manuscript updates have been helpful. Please let us know of any leftover concerns and if there was anything else we could do to address any further questions or comments :)
>
> Thank you!
>
> Paper6611 Authors

---

> > ### Author Response · Authors · 2022-08-09
> > **Author Follow-up**
> >
> > Dear Reviewer WioB
> >
> > Thank you again for your time and expertise during the review process!
> >
> > If there were any leftover concerns, we would sincerely appreciate the opportunity to clarify them---before the discussion period for authors ends. We believe our responses have addressed in detail the full set of questions you had raised, along with corresponding appendix/manuscript updates.
> >
> > We would appreciate if the reviewer kindly let us know if there were any further questions. We are eager to do our utmost to address them!
> >
> > Thank you,
> >
> > Paper6611 Authors

---

### Official Review · Reviewer_3GB7 · 2022-07-11

**Rating:** 7
**Confidence:** 4
**Soundness:** 4 excellent
**Presentation:** 3 good
**Contribution:** 4 excellent

**Summary:**

The authors present an approach that given a model can identify 3 general sub-groups in the dataset that differentiates the model performance across the dataset. In particular they analyzed tabular data to find such sub-groups that are useful for meaningful data-driven AI and satisfies requirements from an MLOps perspective. They conducted experiments and demonstrated various aspects of the proposed method. Overall, their proposed method aims at informing reliable model usage.

**Questions:**

Some questions for the authors:
- while the authors rely on the training dataset performance, its not evident that their results and/or formulation is dependent on it. For example, the results in the supplementary section shows how the $v_{al}$ stabilizes after a few epochs. One can also draw corollary from the steady state dynamics for state space model and so on. Equation 1 also approximates the aleatoric uncertainty as an expectation. One can argue that for converged models, the steady state value may be enough. In contrast would a burn-in period have an impact on this estimate?
- It seems that the entire formulation is true while comparing models of similar performance. How does this formulation play into Eqns (1) and (2).
- Section 4.3: How many ambiguous datapoints were in each dataset? The absolute number is important while analyzing the importance of the effect of removing such datapoints on model performance

**Strengths And Weaknesses:**

The are several interesting aspects of this paper

- the proposed method of identifying the sub-groups using the training performance of the model using the data uncertainty aspect is while simple, is intuitive and more importantly the presented analyses supports the importance of this formulation.
- The identified 4 paradigms for a useful sub-group discovery is interesting, especially from a MLops perspective. The paper is masterfully presented that describes and motivates each of these paradigm and analyzes how the proposed method addresses each of these requirements. Especially, the  plug and play aspect of this method is very appealing. The results presented in the appendix comparing the classical GBT models is also interesting (perhaps could be included in the main paper).
- The insights about the importance of the `Ambiguous` groups is perhaps the most interesting aspect of the paper. Pertinently the novel usage of this identified sub-group for aspects such as identifying the most important features to comparing the `value` of a dataset is interesting.

While the paper itself is quite interesting, there are a few aspects to improve upon:
- the Figures are very low resolution and hinders readability. While acknowledging the space limitations, some of the sections/paragraph could be moved to supplementary section. For example, Figure 4 and the corresponding paragraph comparing agains DataMaps is quite intuitive and may not need to be part of the main paper. The y-axis is closely related to the metric for $v_{al}$ and the shape is quite expected. Similarly, Fig 8 can be argued to be intuitive and a direct consequence of the selected measure.
- Table 1 and the corresponding paragraph may need further context. The discussion ignores the context around the fidelity of the synthetic data to actual ground truth. E.g. guarding against trivially Easy data
- Finally, while the paper is quite well written, the actual method to calculate the Easy, Ambiguous, and Hard groups is not presented properly - it should be present at least as an algorithm in the supplementary section.

---

> ### Author Response · Authors · 2022-08-02
> **Response to Reviewer 3GB7 [Part 1/3]**
>
> Thank you for your thoughtful comments and suggestions.
>
> We give answers to each of the following in turn, as well as highlighting corresponding updates to the revised manuscript at the end of every point. We have uploaded the revised manuscript which incorporates the changes and suggestions:
>
> (A) Shift Neural Network vs XGBoost comparison to the main paper **Part[2/3]**
>
> (B) Increase figure resolution **Part[2/3]**
>
> (C) Add method algorithm to supplementary **Part[2/3]**
>
> (D) Clarify the synthetic data experiment context **Part[2/3]**
>
> (E) Effect of burn-in period **Part[3/3]**
>
> (F) Clarify the Link between formulation & Eq 1 and 2 **Part[3/3]**
>
> (G) Clarify the absolute number of Ambiguous samples in Section 4.3 **Part[3/3]**
>
> `Response continues in part 2`

---

> > ### Author Response · Authors · 2022-08-02
> > **Response to Reviewer 3GB7 [Part 2/3]**
> >
> > `This is part 2 of the authors' response`
> >
> > -----
> >
> > # (A) Shift Neural Network vs XGBoost comparison to the main paper
> >
> > ------
> > **TL;DR:** We will shift the comparison across model types to the main paper in the additional camera ready page. We highlight the new results which include two additional models (LightGBM and CatBoost) to highlight generality, as well as, the new quantitative experiment which highlights Data-IQ’s consistency of characterization across model types. These will be included as part of the revised discussion.
> >
> > ------
> >
> > Thank you for the suggestion! We agree that the comparison between neural networks and classical GBT models is an interesting and important comparison. We will shift the results from the supplementary material into the additional camera-ready page.
> >
> > While we initially evaluated our method with neural networks vs XGBoost (**Appendix C.2**); but to demonstrate the wide applicability of our method, we have also included an additional two iteratively trained models: LightGBM and CatBoost. We have updated the figures in **Appendix C.2** to reflect this, but will shift the discussion to the main paper in the additional camera-ready page.
> >
> > These updated results include the Data-IQ vs Data Maps plots, showing Data-IQ provides a most consistent characterization across models compared to Data Maps. We have also added another quantitative result where we assess the Spearman correlation of the metrics across model combinations (including the two new models), similar to Section 4.1. We provide a snapshot of these results below.
> >
> >
> > Table 1. Correlation over all model combinations (neural nets, XGBoost, LightGBM and CatBoost) for Data-IQ vs Data Maps. We see Data-IQ is more robust to these variations.
> >
> >
> >  Dataset  | (Ours) Data-IQ | Data Maps    |
> > |----------|----------------|--------------|
> > | Covid    |  0.85 +- 0.04  | 0.57 +- 0.07 |
> > | Support  |  0.83 +- 0.03  | 0.48 +- 0.09 |
> > | Prostate |  0.80 +- 0.07  | 0.42 +- 0.29 |
> > | Fetal    |  0.72 +- 0.09  | 0.45 +- 0.06 |
> >
> >
> > **UPDATES:** We have updated **Appendix C.2** with the new results, but will shift the discussion to the main paper in the additional camera-ready page.
> >
> > ------
> > # (B) Increase figure resolution:
> >
> > ------
> > Thank you for pointing this out! We have re-generated the Figures as high-resolution vector images in PDF form. We have updated the images in the revised version uploaded.
> >
> > **Updates:** Figures in the revised manuscript have been updated to the high-res PDF form.
> >
> > ------
> > # (C) Add method algorithm to supplementary
> >
> > ------
> >
> > Thank you for the suggestion! On the basis of the suggestion, we have included an algorithm as part of the supplementary, **Appendix B** in the revised paper.
> >
> > **UPDATES**: Appendix B now includes an algorithm of Data-IQ.
> >
> > ------
> > # (D) Clarify the synthetic data experiment context
> >
> > ------
> > We wish to clarify the context around Table 1 and the surrounding paragraph. A common way to assess the **fidelity** of synthetic data generation methods is to **train on the synthetic data and test on real data** [R1,R2]. The synthetic data which has the best fidelity is the one that produces the best model performance when evaluated on the real data (test set). This assessment guards against trivial training samples, as the model trained on that data subsequently won’t generalize when assess with the real test data.
> >
> > This is the approach we take in the experiment. The reason is that we have assumed the setup of comparing two synthetic data vendors providing synthetic training datasets, yet we do not have access to the "REAL" training data, only synthetic versions. However, do have access to  "REAL" test data. Hence, train on synthetic, test on real fits this setting well.
> >
> > To clarify the setup, we use Data-IQ to characterize the synthetic datasets and the percentage of Easy examples for each synthetic version as a quality measure. The accuracy performance rankings refer to the test performance on real data, which measures fidelity. The model achieving rank 1 implies the best synthetic data, as it produces a model which achieves the best test performance on real data. We then make the link that the synthetic datasets with the best quality, as measured by Data-IQ also produce the best test performance on real data (i.e. Rank 1).
> >
> > **UPDATES:** We have updated the explanation in **Section 4.2** of the main paper to clarify these points and included the additional references.
> >
> > -------
> > `Response continues in part 3`

---

> > > ### Author Response · Authors · 2022-08-02
> > > **Response to Reviewer 3GB7 [Part 3/3]**
> > >
> > >
> > > `This is part 3 of the authors' response`
> > >
> > > ------
> > > # (E) Effect of burn-in period
> > >
> > >
> > > ------
> > > To assess the impact of a burn-in period, we vary the epoch from which we start to capture and compute the metrics. For one variant of the model, we compute the metrics from the beginning of training (No burn-in). Whilst, for a second variant of the model, we compute the metrics across epochs, only once the model has converged and reached a steady state. This is done by tracking the loss.
> > >
> > > We then compute the correlation of the average Data-IQ and Data Maps per-sample metrics, i.e. aleatoric and epistemic respectively between the No burn-in vs Burn-in metrics.
> > >
> > > Note, we test one model with early stopping and one without and show the results below.
> > >
> > > Corr(No burn-in, Burn-in)
> > > 1. Early stopping:
> > > Data-IQ (Aleatoric uncertainty): 0.96
> > > Data Maps (Epistemic uncertainty): 0.58
> > >
> > > 2. No early stopping:
> > > Data-IQ (Aleatoric uncertainty): 0.970
> > > Data Maps (Epistemic uncertainty):  0.73
> > >
> > > What does this result show?
> > > * Data-IQ is much less sensitive whether there is a burn-in period or not, meaning the estimates and findings will be similar and more stable.
> > > * That said, we highlight that we still require the training dynamic to delineate the different groups. The reason is our formulation relies on BOTH aleatoric uncertainty and predictive confidence. We show the evolution of predictive confidence (see https://imgur.com/a/clhZ6Pt). This shows that the different types of samples EASY, AMBIGUOUS and HARD evolve differently through training. By analyzing the training dynamic, we can correctly characterize the difference, especially between EASY and HARD on the basis of the confidence. We have outlined this necessity in  **Section 3.3, L180-183**.
> > >
> > >
> > >
> > > ------
> > > # (F) Clarify the Link between formulation & Eq 1 and 2
> > >
> > > ------
> > > Our formulation looks at comparing similar performing models. The reason for similar performing models is that naturally, a weak model (e.g. linear) cannot be compared to a high-performing model (e.g. neural network) - as the differences would result from the difference in model capacity.
> > >
> > > This links to Eq.1 & 2 in that if the models perform similarly, this means we have reduced variation due to differences in model performance. Then our primary metric of aleatoric uncertainty is comparable and indeed captures examples for which the model cannot predict the appropriate label with high confidence.
> > >
> > > This is precisely data uncertainty. i.e. we are able to compare the variability due to the inherent inability to predict with certainty (linked to the data).
> > >
> > > ------
> > > # (G) Clarify the absolute number of Ambiguous samples in Section 4.3
> > >
> > > ------
> > > To clarify the effect of data sculpting, we outline the absolute number of data points of Ambiguous samples. Note, in the experiment in Section 4.3, we balance the number of samples in both SEER (USA) and CUTRACT (UK), with 1000 samples in each dataset. Hence, the full Ambiguous sub-group (i.e. proportion=1) has a total of 500 samples. When, we sweep the x-axis which is proportions of Ambiguous  samples; p = {0, 0.2, 0.4, 0.6, 0.8, 1}, this is reflective of {0, 100, 200, 300, 400, 500} absolute samples. This shows that the **number of samples that can be excluded is fairly substantial**.
> > >
> > > **UPDATES:** We have included the additional figure with the absolute numbers in the revised supplementary, **Appendix C.13**. We have also conducted an additional experiment, where we don't balance SEER (USA) and CUTRACT (UK).  The datasets then have 10000 and 1000 samples respectively. These results are also included in the revised **Appendix C.13**
> > >
> > > ------
> > > `End of author response`
> > >
> > > ------
> > > # References:
> > >
> > > ------
> > > [R1] Esteban, Cristóbal, Stephanie L. Hyland, and Gunnar Rätsch. "Real-valued (medical) time series generation with recurrent conditional gans." arXiv preprint arXiv:1706.02633 (2017).
> > >
> > > [R2] Platzer, Michael, and Thomas Reutterer. "Holdout-based empirical assessment of mixed-type synthetic data." Frontiers in big Data (2021): 43.

---

> ### Author Response · Authors · 2022-08-05
> **Dear Reviewer 3GB7**
>
> Dear Reviewer 3GB7
>
> We are sincerely grateful for your time and energy in the review process.
>
> We hope that our responses and appendix/manuscript updates have been helpful. Please let us know of any leftover concerns and if there was anything else we could do to address any further questions or comments :)
>
> Thank you!
>
> Paper6611 Authors

---

> ### Comment · Reviewer_3GB7 · 2022-08-07
> **Thanks for the response**
>
> thanks for the detailed response. Overall the response makes sense, especially the burn-in results are most appreciated.

---

> > ### Author Response · Authors · 2022-08-09
> > **Thank you for the reply**
> >
> > Dear Reviewer 3GB7
> >
> > Thank you for your reply & positive feedback. We appreciate the invaluable feedback on our paper.
> >
> > Please let us know if there are any further questions; we are happy to engage :)
> >
> > Regards
> >
> > Paper6611 Authors

---

### Official Review · Reviewer_YLpH · 2022-07-12

**Rating:** 6
**Confidence:** 4
**Soundness:** 3 good
**Presentation:** 3 good
**Contribution:** 3 good

**Summary:**

The authors make a strong case that population level metrics for model performance may not be representative of model performance on subgroups. The authors proposed `DataIQ` a framework to phenotype and subgroup patients based on model uncertainity on an individual level. The authors apply their proposed technique on multiple real world healthcare datasets and demonstrate ability of the model to 1) identify robust data 2) Data collection 3) Model deployment.

**Questions:**

- Have the authors considered other potential sources of incorporating randomness for the aleatoric uncertainty they are trying to capture over the distribution of $v$, for example perhaps the use of a bayesian prior on $v$ or dropout masks?


**Limitations:**

As indicated above the distribution of $v$ is intimately tied to the model choice and the optimization approach. Even when the model class is the same, the hyper parameter choices for the optimizer like step size, momentum, might lead to vastly different distributions in the weights.

Since the authors propose to use the estimated groups as 'protected groups', it would be welcome to see more extensive discussion of how these choices effect recovery of the estimated groups.

**Strengths And Weaknesses:**

Overall I like this paper, even though there are a few weaknesses.

# Strengths

- The paper describes provides several reasons to model model uncertainty and conducts several ablation experiments to justify each use case.
- The multiple ablation studies, including the change in estimated uncertainty with addition of features (Figure 6)  as well as domain shift (Figure 7) are welcome.
- The comparison of output of the model along a with Group DRO objective to improve subgroup level performance adds to the completeness of an already exhaustive set of experiments.

# Weaknesses

- The way the paper proposes to estimate the variance around the model outcomes is by assuming the model parameters at the end of each epoch to be sampled from a distribution. While interesting such a choice is not characteristic of an empirical distribution in the strictest sense. In this case the model parameters are going to be highly correlated which would break the assumption of sampling from an IID distribution.

---

> ### Author Response · Authors · 2022-08-02
> **Response to Reviewer YLpH [Part 1/3]**
>
> Thank you for your thoughtful comments and suggestions.
>
> We give answers to each of the following in turn, as well as highlighting corresponding updates to the revised manuscript at the end of every point. We have uploaded the revised manuscript which incorporates the changes and suggestions:
>
> (A) Clarifying our sampling per epoch & empirical distribution **Part[2/3]**
>
> (B) Adding other sources of randomness **Part[2/3]**
>
> (C) Effect of different hyperparameters **Part[3/3]**
>
> (D) Effect of choices on recovery of estimated groups **Part[3/3]**
>
> `Response continues in part 2`

---

> > ### Author Response · Authors · 2022-08-02
> > **Response to Reviewer YLpH [Part 2/3]**
> >
> > `This is part 2 of the authors' response`
> >
> > -----
> >
> >
> > # (A) Clarifying our sampling per epoch & empirical distribution
> >
> > -----
> > Our method Data-IQ is similar in spirit to Data Maps [R1] in that the goal is to summarize the variability of the sample dynamics over the course of training (across epochs).
> > Consequently, the empirical distribution is simply the observed parameters during training. Hence, we clarify that we are not trying to sample the parameters as would be done in a Bayesian setting. Thus, we highlight that our goal is different, as we simply want to measure the dispersion of observed variables.
> >
> > That said, we have also conducted an experiment to assess the correlation of model weights over epochs $e \in \{1,2...E\}$. i.e. we assess the average correlation of model weights $(e_1, e_2), (e_2, e_3)$ and so on. We follow Jin et al [R2], where weight correlation is defined as the average cosine similarity between weight vectors of neurons.
> >
> > The paper [R2] shows that LOW correlation is less than 0.2. Our validation shows that the weights have **low** correlation, with an average weight correlation for models across the 4 datasets of **0.1 +- 0.02**. Thus, even though our goal is different from a typical sampling setting (i.e. we just want to measure the dispersion of observed variables) - we still wish to highlight the low correlation of weights between epochs.
> >
> > That being said, if this is still a concern, a practical solution is that the practitioner could decide to simply increase the intervals between which the training dynamics are captured, thereby decreasing this correlation even further.
> >
> > **UPDATES**: We have included the discussion and experiment in the revised manuscript, **Appendix C.12**.
> >
> >
> > -----
> > # (B) Adding other sources of randomness
> >
> > ------
> > **TL;DR:** We assess the inclusion of dropout as an extra source of randomness. We find that Data-IQ remains the method most robust to variation, even with this additional randomness.
> >
> > ------
> > Thank you for the suggestion! We have included dropout as an extra source of randomness. We have tested our original model parameterizations from Section 4.1, but additionally trained an 3 more models with the same architecture, yet with the inclusion of different dropout masks with dropout probability: p=0.1, p=0.2, p=0.3.
> >
> > We then assess the Spearman correlation of the metrics across all model combinations, as was done in the experiment on robustness to variation from Section 4.1. The higher correlation, implies better robustness to variations. We find that Data-IQ remains the method most robust to variation, even with this additional randomness. We highlight the results on all four datasets below.
> >
> >
> > | Dataset  | (Ours) Data-IQ Correlation | Data Maps Correlation |
> > |----------|----------------------------|-----------------------|
> > | Covid    |        0.95 +- 0.03        |      0.72 +- 0.15     |
> > | Support  |        0.94 +- 0.01        |      0.68 +- 0.11     |
> > | Prostate |        0.97 +- 0.01        |      0.96 +- 0.01     |
> > | Fetal    |        0.85 +- 0.10        |      0.69 +- 0.15     |
> >
> > **UPDATES:** We include the experiment and a discussion as part of the revised manuscript in **Appendix C.10**.
> >
> > ------
> > `Response continues in part 3`

---

> > > ### Author Response · Authors · 2022-08-02
> > > **Response to Reviewer YLpH [Part 3/3]**
> > >
> > > `This is part 3 of the authors' response`
> > >
> > > ------
> > > # (C) Effect of different hyperparameters
> > >
> > > ------
> > > **TL;DR:**: The experiment in **Section 4.1**, assesses consistency under different model parameterizations and finds Data-IQ is most consistent and robust to variation across these different parameterizations vs baselines.
> > >
> > > ------
> > >
> > > We assess the impact of different model parameterizations (i.e. hyperparameters). Specifically, we assess the effect of different parameterizations on the robustness of Data-IQ, in **Section 4.1**. There we compare the consistency of the different characterization metrics for different parameterizations. We have specifically assessed the number of layers, hidden units, and type of optimizer used (SGD, Adam). We have highlighted these different parameterizations in **Appendix B**.
> > >
> > > As outlined in **Section 4.1, L263-L269**, when comparing the Spearman rank correlation between the metrics for all different model parameterizations, Data-IQ is shown to be the most consistent and robust to variation having the highest score on all datasets. This result is showcased in **Figure 3**. Thus, even though model parameterizations might differ, we highlight that Data-IQ is most robust to these variations when compared to the baseline methods.
> > >
> > > Furthermore, the aforementioned results with the inclusion of dropout also highlight Data-IQ’s robustness to variation
> > >
> > > ------
> > > # (D) Effect of choices on recovery of estimated groups
> > >
> > > ------
> > > **TL;DR:** We assess the effect of choices on the recovery of estimated groups. We show that there are large overlaps of the recovered groups across model parameterizations and that the sub-group assignment using Data-IQ is stable.
> > >
> > > ------
> > > Thank you for the suggestion! We assess how the model choices/parameterizations might affect the recovery of the different sub-groups.
> > >
> > > We desire that if a sample is categorized as EASY with one parameterization, that similarly it should be categorized as EASY for another parameterization. Similarly, we want their sub-group characterization to stay the same across different model parameterizations for the other sub-groups. We conduct an experiment to assess this property of stability of group assignment across parameterizations.
> > >
> > > We use the same sets of model parameterizations as assessed in Section 4.1 (Robustness to variation experiment). We then compute the overlap of sample IDs for each sub-group (EASY, AMBIGUOUS, HARD) between the different models. This can be thought of as accuracy or stability of sample assignment to a specific sub-group. We present the results across all four datasets below, which suggest that the recovery of sub-groups is stable, with more than 90% of samples being assigned to the same sub-group between different models. This further motivates modeling the inherent data uncertainty (aleatoric).
> > >
> > > Table 1. Assessment of stability in group assignment across different parameterizations. The score reflects the average overlap at the sample level
> > > | Dataset  | (Ours) Data-IQ Overlap/Acc |
> > > |----------|----------------------------|
> > > | Covid    |        0.92 +- 0.03        |
> > > | Support  |        0.92 +- 0.007       |
> > > | Prostate |        0.95 +- 0.004       |
> > > | Fetal    |        0.77 +- 0.14        |
> > >
> > >
> > > **UPDATES:** We include this experiment and discussion in the revised manuscript, as **Appendix C.11**.
> > >
> > > ------
> > > # REFERENCES
> > >
> > > ------
> > > [R1] Swabha Swayamdipta, Roy Schwartz, Nicholas Lourie, Yizhong Wang, Hannaneh Hajishirzi, Noah A Smith, and Yejin Choi. Dataset cartography: Mapping and diagnosing datasets with training dynamics. In Proceedings of the 2020 Conference on Empirical Methods in Natural Language Processing (EMNLP), pages 9275–9293, 2020.
> > >
> > > [R2] Jin, Gaojie, Xinping Yi, Liang Zhang, Lijun Zhang, Sven Schewe, and Xiaowei Huang. "How does weight correlation affect generalisation ability of deep neural networks?." Advances in Neural Information Processing Systems 33 (2020): 21346-21356.

---

> ### Author Response · Authors · 2022-08-05
> **Dear Reviewer YLpH**
>
> Dear Reviewer YLpH
>
> We are sincerely grateful for your time and energy in the review process.
>
> We hope that our responses and appendix/manuscript updates have been helpful. Please let us know of any leftover concerns and if there was anything else we could do to address any further questions or comments :)
>
> Thank you!
>
> Paper6611 Authors

---

> > ### Author Response · Authors · 2022-08-09
> > **Author Follow-up**
> >
> > Dear Reviewer YLpH
> >
> > Thank you again for your time and expertise during the review process!
> >
> > If there were any leftover concerns, we would sincerely appreciate the opportunity to clarify them---before the discussion period for authors ends. We believe our responses have addressed in detail the full set of questions you had raised, along with corresponding appendix/manuscript updates.
> >
> > We would appreciate if the reviewer kindly let us know if there were any further questions. We are eager to do our utmost to address them!
> >
> > Thank you,
> >
> > Paper6611 Authors

---

### Official Review · Reviewer_6nr2 · 2022-07-13

**Rating:** 6
**Confidence:** 3
**Soundness:** 3 good
**Presentation:** 4 excellent
**Contribution:** 3 good

**Summary:**

This paper proposed Data-IQ, a framework compatible with any machine learning (classification) models whose training is conducted in stages (iterations/epochs), to assess the quality of data samples in tabular format by categorizing them into Easy, Ambiguous, Hard groups based on the aleatoric uncertainty of the data samples. The paper demonstrated the utility of Data-IQ in guiding feature acquisition, comparing training datasets and improving model generalization through experiments and analysis on multiple real-world datasets.

**Questions:**

1. The aleatoric uncertainty is quantified as the average variance over all training epochs. I am wondering how the distribution of the variance of a single data sample over the training epochs looks like? If the distribution of one training example is skewed, e.g. the variance is high at the beginning but it becomes much lower as the model parameters are refined during training, and the variance of another training example remains at a medium level during training, these two training examples might end up with similar average uncertainty but the training dynamics are very different. In this case, would the average variance still be a good metric to evaluate the uncertainty of an individual data sample?


2. The evaluation are mainly conducted on neural networks, which demonstrates that Data-IQ is robust to different parametrizations of the same model type. Would it be possible to compare the results across different model types? Without comparison across model types, e.g. neural networks v.s. gradient boosting, the claim that Data-IQ is robust to model variation is less convincing as the aleatoric uncertainty might still be pertinent to the model class (though not to different parameters under the same model type).

**Limitations:**

The authors have addressed the limitations and I think another potential limitation is how robust Data-IQ will be across different model types as stated in Q2 in **Questions**.

**Strengths And Weaknesses:**

**Originality:** the idea of using prediction outcomes to quantify the uncertainty of data samples is not new, e.g. in some active learning approaches the next queried data point would be the sample with predicted probability close to 0.5 as it is the most uncertain. The novelty of Data-IQ lies in that it quantifies the aleatoric uncertainty of data samples by evaluating the variance of the prediction outcome which intends to capture the inherent uncertainty of the data samples rather than the uncertainty with regard to the model parameters. Therefore, Data-IQ is capable of providing guidance to improve data quality in new directions with regard to the dataset itself, e.g. new feature acquisition.

**Quality:** the evaluation of the variance of the prediction outcome is straightforward, by noting that the classification outcome is a Bernoulli random variable, the variance is $p(x, \theta_e)(1 - p(x, \theta_e))$ in which $p(x, \theta_e)$ is the predicted probability using the model parameters $\theta_e$ at training epoch $e$. Then the aleatoric uncertainty is quantified as the average variance over all training epochs and categorization of samples is based on thresholds chosen by heuristics. The analysis of the categorization on real-world datasets is quite comprehensive, e.g. how to use Data-IQ to assess the quality of a new feature and how the Ambiguous samples will impact model generalization etc., but with regard to methodological perspectives, the technical novelty / contribution is not significant.

**Clarity:** the paper is well written and easy to follow. All the figures and tables are very clear.

**Significance:**  I think the research topic in this paper is of high practical value as the assessment of data quality (uncertainty) is very important for training high-quality machine learning models and obtaining reliable predictions on unseen data. However, as stated in
**Quality**, the methodological / theoretical novelty of the method is not very high.

---

> ### Author Response · Authors · 2022-08-02
> **Response to Reviewer 6nr2 [Part 1/5]**
>
> Thank you for your thoughtful comments and suggestions.
>
> We give answers to each of the following in turn, as well as highlighting corresponding updates to the revised manuscript at the end of every point. We have uploaded the revised manuscript which incorporates the changes and suggestions:
>
> (A) Novelty and contribution of Data-IQ **[Part 2/5]**
>
> (B) Comparison of Data-IQ’s usage of aleatoric uncertainty to active learning metrics **[Part 3/5]**
>
> (C) Comparison of Data-IQ across different model types, neural networks, XGBoost etc **[Part 4/5]**
>
> (D) Effect of averaging over the metric **[Part 5/5]**
>
> -----
>
> ## Preface:
> We wish to highlight that the problem solved by Data-IQ is to assess and characterize the data itself into sub-groups. This motivates our usage of aleatoric uncertainty, as we seek to model the uncertainty inherent to the data. Identifying these sub-groups is a practically valuable problem, as improving accuracy and robustness often depends on the data's characteristics and quality [R1,R2,R3].  That said, the problem tacked by Data-IQ around data quality as mentioned in [R3,R4]  “is often undervalued as merely operational, yet failing to account for it can have immense practical harm” [R3].
>
> Consequently, our goal in Data-IQ is to build a systematic framework with four desired properties, motivated by the considerations of practitioners at various stages of the ML pipeline. In satisfying these properties, Data-IQ addresses the “dire need for an ML-aware data quality that is not only principled, but also practical for a larger collection of ML models” [R2].
>
> At this point, we clarify two points:
> 1. We wish to clarify the problem solved by Data-IQ fundamentally differs from active learning. The goal of active learning-based uncertainty sampling is to improve the model by selecting which samples to best label next. This is different from Data-IQ, which assesses and characterizes the data itself into sub-groups.
> 2. We wish to highlight the difference between our metric, aleatoric uncertainty which models the uncertainty inherent to the data, and contrast it with the more commonly used metric of epistemic uncertainty which assesses model uncertainty.
>
> * **Aleatoric (data) uncertainty**: ​​ Variability due to inherent inability to predict with certainty (linked to the **data**)
>
> $$v_{\mathrm{al}}=\mathbb{E_{\vartheta}}[\mathbb{V}_{Y|X, \vartheta}[Y|X,\vartheta]]$$
>
> *  **Epistemic (model) uncertainty**: captures the variability due to change in predictions
> via parameters (linked to the **model**)
> $$v_{\mathrm{ep}}=\mathbb{V_{\vartheta}}[\mathbb{E}_{Y|X, \vartheta}[Y|X,\vartheta]]$$
>
> `Response continues in part 2`

---

> > ### Author Response · Authors · 2022-08-02
> > **Response to Reviewer 6nr2 [Part 2/5]**
> >
> > -----
> > # (A) Novelty & contribution of Data-IQ
> > ------
> > **TL;DR:** Data-IQ’s novelty & contribution is along the following dimensions:
> > 1. Data-IQ introduces a novel usage of aleatoric uncertainty, which is principally motivated.
> > 2. Data-IQ’s formulation provides applicability to other model types beyond neural networks.
> > 3. Data-IQ defines new paradigms of useful sub-group discovery.
> > 4. Data-IQ introduces novel use-cases of the subgroups.
> > 5. Data-IQ's metric vs similarly intuitive SOTA metrics.
> > 6. Data-IQ can be used across modalities (tabular, image, text)
> > ------
> > While the aleatoric uncertainty metric in Data-IQ might seem intuitive, we highlight Data-IQ's novelty and contributions along the following dimensions:
> > 1. **Data-IQ introduces a novel usage of aleatoric uncertainty, which is principally motivated.**: Data-IQ is the first to characterize data samples using aleatoric uncertainty. We motivate its usage as more principled to prior methods in **Sec. 3.3** , as aleatoric uncertainty captures the uncertainty inherent to the data. This differs from uncertainty tied to a specific model (epistemic uncertainty). *Fig. 2** then shows why the correct type of uncertainty matters.  This represents a  **conceptual shift** by highlighting the principled usage of aleatoric (data) uncertainty, instead of the commonly used epistemic (model) uncertainty.
> >
> > Further, we show that aleatoric uncertainty is more robust to variation across models/parameterizations, with Data-IQ experimentally outperforming the baselines. This was noted by *R-3GB7*, who highlights that the  "data uncertainty aspect is while simple, is intuitive and more importantly the presented analyses supports the importance of this formulation."
> >
> > 2. **Data-IQ’s formulation provides applicability to other model types beyond neural networks.**:
> > Another novelty of Data-IQ is the "plug & play" formulation, where Data-IQ can be used with *any* iteratively trained ML model: neural networks, XGBoost, LightGBM etc. For more, see the comment on *Comparison across different model types*. This flexibility is enabled by our formulation outlined in **Sec. 3.4**. Moreover, Data-IQ addresses the limitation of methods such as [R2] and [R3], which only apply to neural networks. This allows Data-IQ to provide utility in high-stakes tabular settings, e.g. healthcare/finance, which often use non-neural methods. Hence, in these settings, the “plug-and-play” ability provides increased utility vs other methods.
> > 3. **Data-IQ defines new paradigms of useful sub-group discovery**: Data-IQ defines and motivates properties necessary for principled data quality assessment and sub-group characterization (**Sec 1, L57-64**), i.e. (P1) Robust data characterization, (P2) Principled data collection,(P3) Reliable model deployment, (P4) Plug & play. As noted by *R-3GB7*, the "4 paradigms for a useful sub-group discovery is interesting, especially from a MLops perspective".  Thus, we believe that P1-P4 could be a useful contribution to the community by formalizing desired properties in this nascent area.
> > 4. **Data-IQ introduces novel use-cases of the sub-groups:** A less obvious novelty of Data-IQ is how the identified sub-groups can be used to enable important tasks across the ML pipeline including feature acquisition, data selection, data sculpting and reliable model deployment. Specifically, Data-IQ is the first paper to make the connections between sub-groups and how they enable these use-cases, which we then experimentally validate.
> > 5. **Data-IQ's metric vs similarly intuitive SOTA metrics:** The metrics used by the baselines could be regarded as similarly intuitive as Data-IQ. We highlight the key differences of our metric and why based on the four properties (P1-P4) that modeling the inherent data (aleatoric) uncertainty is more useful. This comparison is part of the extended related work in **Table 3 - Appendix A**. We summarize the metrics used by the baseline methods below, highlighting their simplicity.
> >
> > | Method         | Metric                   |
> > |----------------|--------------------------|
> > | Data Maps [R1] | Epistemic uncertainty    |
> > | AUM [R2]       | Difference of logits     |
> > | GradNd [R3]    | Gradient Norm            |
> > | JTT [R4]       | Upsample training errors |
> >
> >
> > While, Data-IQ might have a similarly intuitive metric, we show both principally/theoretically (**Sec. 3.3 & Appendix A**) and experimentally (**Sec. 4.1**) why our metric is indeed a useful contribution.
> >
> > 6. **Data-IQ can be used across modalities (tabular, image, text)**: Prior works have primarily focused on one modality, either images [R6,R7,R8] OR text [R5]. Tabular data is often overlooked, with Data-IQ addressing this gap. That said, we have added further experiments on text and image data to highlight the generality of Data-IQ. For more see **Appendix C.7 & C.8**. For a snapshot, see **TEXT** [https://imgur.com/a/NoK9I0t], **IMAGES** [https://imgur.com/a/TcDq9eg].
> >
> > `Response continues in part 3`

---

> > > ### Author Response · Authors · 2022-08-02
> > > **Response to Reviewer 6nr2 [Part 3/5]**
> > >
> > > `This is part 3 of the authors' response`
> > >
> > > # (B) Comparison of aleatoric uncertainty to active learning metrics
> > >
> > > ------
> > > **TL;DR:** Active learning samples points for which the model is uncertain as samples to label next. This is epistemic uncertainty (model uncertainty), which differs from the Data-IQ metric of aleatoric uncertainty (inherent data uncertainty). Data-IQ also differs in scope from active learning.
> > >
> > > ------
> > >
> > > We wish to compare and contrast Data-IQ and show that it is different from active learning both in metric and scope.
> > >
> > > 1. **Uncertainty metrics from active learning are different to aleatoric uncertainty (our metric).**
> > >
> > > We highlight that while a common approach in active learning is Uncertainty sampling, this selects instances for which a specific model is uncertain [R9], as samples to label next. This active learning type of uncertainty is **Epistemic uncertainty** which is model uncertainty. This is different from Data-IQ’s metric of aleatoric uncertainty, which is uncertainty inherent to the data itself, i.e. data uncertainty.
> > >
> > > In more depth, Maximum Entropy Sampling (MES) or Uncertainty Sampling aims to query samples with the maximum predictive entropy [R10,R11,R12]. This assesses a specific model's uncertainty, which is different from our data uncertainty (aleatoric) metric. The other paradigm of Bayesian Active Learning by Disagreement (BALD) aims to query points that maximize the mutual information between the observation and the model parameters [R13,R14]. This again reflects the model uncertainty rather than the aleatoric (data) uncertainty that is used by Data-IQ. It has further been shown by [R15] who use epistemic uncertainty for active learning that for "uncertainty sampling, the usefulness of an instance is better reflected by its epistemic uncertainty." These points highlight the difference that uncertainty-based selection in active learning assesses the model’s uncertainty (epistemic), which is different from the inherent data uncertainty (aleatoric) used in Data-IQ.
> > >
> > > 2. **Scope of Data-IQ is different to active learning:**
> > >
> > > In active learning, the goal is to select samples for which the model is uncertain in order to select the best samples to label next. Whereas, Data-IQ's scope is fundamentally different from active learning. In contrast, Data-IQ characterizes/stratifies sub-groups in an already given dataset, providing ML-aware data quality assessment.
> > >
> > > `Response continues in part 4`

---

> > > > ### Author Response · Authors · 2022-08-02
> > > > **Response to Reviewer 6nr2 [Part 4/5]**
> > > >
> > > > `This is part 4 of the authors' response`
> > > >
> > > > ------
> > > > # (C) Comparison across different model types
> > > >
> > > > ------
> > > > **TL;DR:** We have initially evaluated our method with neural networks vs XGBoost (**see Appendix C.2**); but to further demonstrate the wide applicability of our method, we have also added a comparison with LightGBM and CatBoost to the revised version. The results show the consistent and stable characterization of Data-IQ compared to Data Maps.
> > > >
> > > > ------
> > > > We agree that it is important to compare Data-IQ across different model types, for example e.g. neural networks v.s. gradient boosting. We address this comparison in  **Appendix C.2 - Data-IQ: Neural Networks vs XGBoost**.
> > > >
> > > > As we discuss in  **Section 4.1, L260-262**, the majority of baseline methods are constrained to neural networks, as a result, we mainly compare Data-IQ to Data Maps. The results show that Data-IQ provides a more consistent and stable characterization across these two model classes compared to Data Maps.
> > > >
> > > > We have initially evaluated our method with neural networks vs XGBoost (**Appendix C.2**); but to demonstrate the wide applicability of our method, we have also added LightGBM and CatBoost  (two widely used methods on tabular data [R16]).
> > > >
> > > > To reflect these additions, we have updated the figures in **Appendix C.2** to include the results of the two new additional model types (LightGBM, CatBoost). We see a similar result with Data-IQ being more consistent across these model types when compared to Data Maps.
> > > >
> > > > To further augment this result, we quantitatively assess the consistency across model types. Similar to Section 4.1, we have compared the Spearman rank correlation between the metrics of all model combinations (neural networks, XGBoost, LightGBM, CatBoost). Note that all four models are trained to perform similarly on a held-out dataset.
> > > >
> > > > Below, we present a table showing the average and standard deviation. We see that Data-IQ has higher scores, highlighting the better consistency compared to Data Maps.
> > > >
> > > >
> > > > **UPDATES**: We update **Appendix C.2* to include the new results.
> > > >
> > > > Table 1. Correlation between all model combinations for Data-IQ vs Data Maps. We see Data-IQ is more robust to the model variations, providing a more stable characterization across models.
> > > > | Dataset  | (Ours) Data-IQ | Data Maps    |
> > > > |----------|----------------|--------------|
> > > > | Covid    |  0.85 +- 0.04  | 0.57 +- 0.07 |
> > > > | Support  |  0.83 +- 0.03  | 0.48 +- 0.09 |
> > > > | Prostate |  0.80 +- 0.07  | 0.42 +- 0.29 |
> > > > | Fetal    |  0.72 +- 0.09  | 0.45 +- 0.06 |
> > > >
> > > >
> > > > Table 2. Performance of the different model types on each dataset - where the models have similar performance.
> > > > | Dataset  | Neural Net Acc | XGBoost Acc | LightGBM Acc | CatBoost Acc |
> > > > |----------|----------------|-------------|--------------|--------------|
> > > > | Covid    |      0.72      |     0.73    |     0.74     |     0.72     |
> > > > | Support  |      0.74      |     0.74    |     0.74     |     0.74     |
> > > > | Prostate |      0.84      |     0.85    |     0.85     |     0.85     |
> > > > | Fetal    |      0.88      |     0.91    |     0.89     |     0.90     |
> > > >
> > > > `Response continues in part 5`

---

> > > > > ### Author Response · Authors · 2022-08-02
> > > > > **Response to Reviewer 6nr2 [Part 5/5]**
> > > > >
> > > > > `This is part 5 of the authors' response`
> > > > >
> > > > > ------
> > > > > # (D) Effect of averaging over the metric
> > > > > ------
> > > > > **TL;DR:**  We experimentally assess samples with different training dynamics (high variance then flatten vs remaining consistent). We show that we can still distinguish samples with different training dynamics even when averaging over the uncertainty metric. This helps to validate the usage of the average metric to evaluate the uncertainty of individual data samples.
> > > > >
> > > > > ------
> > > > > Thank you for the suggestion! We have added an experiment to assess the case where an example has high variance at the beginning of training and flattens with time, compared to an example that remains at a medium level throughout training. These represent the EASY (high variance in the beginning and then flattening) and AMBIGUOUS  (medium level throughout) sub-groups respectively. The question is, by taking an average will the uncertainties be similar, despite the training dynamics being different?
> > > > >
> > > > > We conduct an experiment to assess this by sampling 100 examples from each of these groups and plot the following three metrics per sample:
> > > > > 1. Training dynamic (i.e. the aleatoric uncertainty over training epochs).
> > > > > 2. The distribution over aleatoric uncertainty values across all training steps.
> > > > > 3. The “average” aleatoric uncertainty.
> > > > >
> > > > > The result (see https://imgur.com/a/D2AxSx4), shows that we can still distinguish samples with different training dynamics even when averaging over the uncertainty metric. This helps to validate the usage of the average metric to evaluate the uncertainty of individual data samples.
> > > > >
> > > > > **UPDATES**:We have included this result in the revised manuscript as **Appendix C.9**
> > > > >
> > > > > ------
> > > > > `End of author response`
> > > > >
> > > > > ------
> > > > > # References:
> > > > > ------
> > > > >
> > > > > [R1] Abhinav Jain, Hima Patel, Lokesh Nagalapatti, Nitin Gupta, Sameep Mehta, Shanmukha Guttula, Shashank Mujumdar, Shazia Afzal, Ruhi Sharma Mittal, and Vitobha Munigala. Overview and importance of data quality for machine learning tasks. ACM SIGKDD International Conference on Knowledge Discovery & Data Mining, 2020.
> > > > >
> > > > > [R2] Cedric Renggli, Luka Rimanic, Nezihe Merve Gürel, Bojan Karlas, Wentao Wu, and Ce Zhang. A data quality-driven view of mlops. IEEE Data Engineering Bulletin, 2021.
> > > > >
> > > > > [R3] Nithya Sambasivan, Shivani Kapania, Hannah Highfill, Diana Akrong, Praveen Paritosh, and Lora M Aroyo. “everyone wants to do the model work, not the data work”: Data cascades in high-stakes ai. In proceedings of the 2021 CHI Conference on Human Factors in Computing Systems, pages 1–15, 2021.
> > > > >
> > > > > [R4] Kiri Wagstaff. Machine learning that matters. arXiv preprint arXiv:1206.4656, 2012.
> > > > >
> > > > > [R5] Swabha Swayamdipta, Roy Schwartz, Nicholas Lourie, Yizhong Wang, Hannaneh Hajishirzi, Noah A Smith, and Yejin Choi. Dataset cartography: Mapping and diagnosing datasets with training dynamics. 2020 Conference on Empirical Methods in Natural Language Processing (EMNLP), pages 9275–9293, 2020.
> > > > >
> > > > > [R6] Geoff Pleiss, Tianyi Zhang, Ethan Elenberg, and Kilian Q Weinberger. Identifying mislabeled data using the area under the margin ranking. Advances in Neural Information Processing Systems, 33, 2020.
> > > > >
> > > > > [R7] Mansheej Paul, Surya Ganguli, and Gintare Karolina Dziugaite. Deep learning on a data diet: Finding important examples early in training. Advances in Neural Information Processing Systems, 34, 2021.
> > > > >
> > > > > [R8] Evan Z Liu, Behzad Haghgoo, Annie S Chen, Aditi Raghunathan, Pang Wei Koh, Shiori Sagawa, Percy Liang, and Chelsea Finn. Just train twice: Improving group robustness without training group information. In International Conference on Machine Learning,PMLR, 2021.
> > > > >
> > > > > [R9] Sharma, Manali, and Mustafa Bilgic. "Evidence-based uncertainty sampling for active learning." Data Mining and Knowledge Discovery 31, 2017): 164-202.
> > > > >
> > > > > [R10] Lewis, David D., and William A. Gale. "A sequential algorithm for training text classifiers." In SIGIR’94, pp. 3-12. Springer, London, 1994.
> > > > >
> > > > > [R11] Paola Sebastiani and Henry P Wynn. Maximum entropy sampling and optimal bayesian experimental design. Journal of the Royal Statistical Society, 145–157, 2000.
> > > > >
> > > > > [R12] Stephen Mussmann and Percy Liang. On the relationship between data efficiency and error for uncertainty sampling, 35th International Conference on Machine Learning, PMLR.
> > > > >
> > > > > [R13] Neil Houlsby, Ferenc Huszar, Zoubin Ghahramani, and Mate Lengyel. Bayesian active learning for classification and preference learning. arXiv preprint arXiv:1112.5745, 2011.
> > > > >
> > > > > [R14] Andreas Kirsch, Joost van Amersfoort, and Yarin Gal. Batchbald: Efficient and diverse batch acquisition for deep bayesian active learning. In Advances in Neural Information Processing Systems, pp. 7024–7035, 2019.
> > > > >
> > > > > [R15] Nguyen, Vu-Linh, Mohammad Hossein Shaker, and Eyke Hüllermeier. "How to measure uncertainty in uncertainty sampling for active learning." Machine Learning 111, no. 1 (2022): 89-122.
> > > > >
> > > > > [R16] ​​Shwartz-Ziv, Ravid, and Amitai Armon. "Tabular data: Deep learning is not all you need." Information Fusion 81 (2022): 84-90.

---

> ### Author Response · Authors · 2022-08-05
> **Dear Reviewer 6nr2**
>
> Dear Reviewer 6nr2
>
> We are sincerely grateful for your time and energy in the review process.
>
> We hope that our responses and appendix/manuscript updates have been helpful. Please let us know of any leftover concerns and if there was anything else we could do to address any further questions or comments :)
>
> Thank you!
>
> Paper6611 Authors

---

> > ### Author Response · Authors · 2022-08-09
> > **Author Follow-up**
> >
> > Dear Reviewer 6nr2
> >
> > Thank you again for your time and expertise during the review process!
> >
> > If there were any leftover concerns, we would sincerely appreciate the opportunity to clarify them---before the discussion period for authors ends. We believe our responses have addressed in detail the full set of questions you had raised, along with corresponding appendix/manuscript updates.
> >
> > We would appreciate if the reviewer kindly let us know if there were any further questions in the very limited time remaining. We are eager to do our utmost to address them!
> >
> > Thank you,
> >
> > Paper6611 Authors

---

> > > ### Comment · Reviewer_6nr2 · 2022-08-09
> > > **Thank you for the reply**
> > >
> > > Dear authors of paper 6611:
> > >
> > > Thank you very much for the detailed reply! The results of Data-IQ across different model types and data modalities make it more convincing, and the analysis of the training dynamics clarifies my concern. The point that I was trying to make by bringing up active learning in my review is not to compare your method with active learning (they are for different purposes) but that the uncertainty quantification is not entirely new (e.g. according to your aleatoric uncertainty formula, a data point with p=0.5 will result in the highest variance p(1-p)). Your formal characterization and comprehensive case study of the usage of the aleatoric uncertainty in data quality assessment is solid contribution and will definitely help raise awareness of this topic in the machine learning community.
> > >
> > > I thus raise my rating to 6. Thank you again for the great work!

---

### Author Response · Authors · 2022-08-02
**Response Overview**

We thank the reviewers for their insightful and positive feedback!

We are encouraged that they found Data-IQ to tackle a “significant problem” (**R - WioB**) of “high practical value” (**R - 6nr2**) by characterizing data sub-groups and that Data-IQ presents a “novel usage of this identified sub-group” (**R - 3GB7**). Further, they found Data-IQ as a tool which puts “data in the centre” (**R - 2wVN**) to be “innovative and should be encouraged” (**R - 2wVN**), with the “plug and play aspect of this method [being] very appealing”(**R - 3GB7**).

We are glad they deemed the 4 properties/paradigms (P1-P4) that Data-IQ formalizes for “useful sub-group discovery [as] interesting, especially from an MLops perspective”  (**R - 3GB7**). Whilst, the metric might seem “intuitive” (**R - 3BG7**), its use is supported by the “comprehensive” (**R - 6nr2**) and “exhaustive set of experiments” (**R - YLpH**) whose “analyses supports the importance of this formulation”  (**R - 3GB7**).

We address specific questions and concerns below and have incorporated all feedback by highlighting updates to the revised manuscript & supplementary which we have uploaded.

On the basis of our clarifications and updates, we hope we have addressed the reviewers' concerns.
Thank you for your kind consideration!

---

> ### Author Response · Authors · 2022-08-08
> **Dear Reviewers**
>
> Thank you again to the reviewers for their generous comments and suggestions.
>
> Besides the responses provided on a point by point basis, we have also updated the manuscript/supplementary to improve the paper. We wish to summarize the *earlier* updates to the manuscript/supplementary uploaded.
>
> **In addition, we provide new experimental results below, which we hope will be helpful.**
>
> -------
>
> > **New experiments**
>
> 1. Data-IQ produces a bell-shape on both text and image data modalities– similar to tabular data. As discussed in the main paper, this is beneficial as it standardizes Data-IQ’s interpretation. See https://imgur.com/a/YUZpijE
> 2. Data-IQ produces more stable/consistent characterizations when comparing characterizations with 4 different model types/classes. See https://imgur.com/a/MOdIWpW
>
> -------
> > **Summary of updates to manuscript & supplementary**
>
> **Additional modalities & models**
> * Additional model types comparison (increased from neural networks vs XGBoost, to comparing with 4 model types); see updated Appendix C.2
> * Additional data modalities comparison (i.e. TEXT [https://imgur.com/a/NoK9I0t], IMAGES [https://imgur.com/a/TcDq9eg]); see Appendix C.7 and C.8
>
> **Additional assessment of Data-IQ's properties**
> * Assessing the effect of averaging the uncertainty metric in distinguishing samples. *Outcome:* we can distinguish samples with different training dynamics even when averaging;  see Appendix C.9
> * Assessing the effect of including additional sources of randomness. *Outcome:*  Data-IQ is still stable/consistent with the additional randomness; see Appendix C.10
> * Assessing the effect of hyper-parameters on the recovery of estimated groups . *Outcome:*  Data-IQ shows stability in sub-group assignment across different parameterizations. ; see Appendix C.11
>
> **Clarifications**
>
> * Additional algorithm added to clarify Data-IQ; see Appendix B
> * Context clarified for synthetic data experiment; see Section 4.2
> * Correlation of weights through training ; see Appendix C.12
> * Data sculpting: Absolute number of samples ; see Appendix C.13
> * Figures updated to high-resolution
> * Extra camera ready page (once available): discussion on data inclusivity & comparison across model types.
>
> ------
> *With our responses and paper updates, we hope that we have addressed the reviewers' concerns. Please let us know if there were any further questions or comments. We are eager to do our utmost to address them!*
>
> Thank you for your kind consideration :)
>
> Paper6611 Authors

---

### Meta-Review · Area_Chair_jP1M · 2022-08-30

**Recommendation:** Accept
**Confidence:** Certain

**Metareview:**

This easy-to-follow paper has been thoroughly evaluated by five competent reviewers. Four of them rated the work as acceptable (two full and two weak accepts), while one recommended a rejection. In my opinion, the reviewer with the negative assessment has not raised fundamental issues that would disqualify this paper from being considered for NeurIPS. The authors provided extensive clarifications to all the reviewers, including the one with a negative opinion. That reviewer did not engage in discussion with the authors. I recommend accepting this paper without reservations

**Award:**

No

---

### Decision · Program_Chairs · 2022-09-14

Accept